# Conserved role of hnRNPL in alternative splicing of epigenetic modifiers enables B cell activation

Poorani Ganesh Subramani [ID] [1,2], Jennifer Fraszczak[1], Anne Helness[1], Jennifer L Estall [ID] [1,2,3,4], Tarik Möröy [ID] [1,2,3,5] & Javier M Di Noia [ID] [1,2,3,4,5 ✉]

## Abstract

**The multifunctional RNA-binding protein hnRNPL is implicated in antibody class switching but its broader function in B cells is unknown. Here, we show that hnRNPL is essential for B cell activation, germinal center formation, and antibody responses. Upon activation, hnRNPL-deficient B cells show proliferation defects and increased apoptosis. Comparative analysis of RNA-seq data from activated B cells and another eight hnRNPL-depleted cell types reveals common effects on MYC and E2F transcriptional programs required for proliferation. Notably, while individual gene expression changes are cell type specific, several alternative splicing events affecting histone modifiers like KDM6A and SIRT1, are conserved across cell types. Moreover, hnRNPL-deficient B cells show global changes in H3K27me3 and H3K9ac. Epigenetic dysregulation after hnRNPL loss could underlie differential gene expression and upregulation of lncRNAs, and explain common and cell type-specific phenotypes, such as dysfunctional mitochondria and ROS overproduction in mouse B cells. Thus, hnRNPL is essential for the resting-to-activated B cell transition by regulating transcriptional programs and metabolism, at least in part through the alternative splicing of several histone modifiers.**

**Keywords** RNA Binding Protein; Alternative Splicing; B Cell Activation; Antibody Response; hnRNP
**Subject Categories** Chromatin, Transcription & Genomics; Immunology; RNA Biology

## Introduction

Upon exposure to cognate antigen, resting B cells become activated, as a prerequisite to either differentiate into plasma cells or to enter the germinal center, where they undergo affinity-based selection to produce more effective antibodies. The activation of resting B cells entails a switch from a quiescent, low energy spending state to a rapidly growing and proliferating state with much higher levels of transcription (Sadras et al, 2021; Nie et al, 2012; Kieffer-Kwon et al, 2017). To enact this change, B cells undergo rapid transcriptional and metabolic reprogramming, upregulating global transcription as well as glycolysis and oxidative phosphorylation for increased energy and biosynthetic demands (Sadras et al, 2021; Nie et al, 2012; Kieffer-Kwon et al, 2017). The mechanisms that coordinate transcriptional changes during the activation of B cells are thus critical for antibody responses.

RNA-binding proteins can modulate gene expression by affecting transcription, as well as by regulating the splicing and/ or stability of the (pre)-mRNA transcripts, and thus have consequential cellular roles, including in B and other immune cells (Díaz-Muñoz and Turner, 2018; Turner and Díaz-Muñoz, 2018; Marasco and Kornblihtt, 2023). Splicing factors are emerging as important regulators of B cell biology, especially in activated B cells, with the consequent impact on the germinal center reaction (Qureshi et al, 2023; Monzón-Casanova et al, 2018; Diaz-Muñoz et al, 2015; Huang et al, 2023; Osma-Garcia et al, 2021, 2023). hnRNPL belongs to the heterogeneous nuclear ribonucleoprotein (hnRNP) family of RNA-binding proteins, composed of ~20 major, and some less abundant, members; many involved in splicing (Geuens et al, 2016; Han et al, 2010; Marasco and Kornblihtt, 2023). hnRNPL is best known as an alternative splicing factor, largely regulating exon inclusion and exclusion, whereby it can define the usage of protein isoforms (Cole et al, 2015; McClory et al, 2018; Motta-Mena et al, 2010). Other functions have also been reported for hnRNPL. Thus, hnRNPL can directly stabilize some transcripts with long 3'UTRs by protecting them from nonsense-mediated decay (NMD) (Kishor et al, 2019). hnRNPL can also bind to several lncRNAs that can modulate gene expression (Gu et al, 2020), and could thus indirectly affect gene expression. A more direct role of hnRNPL in regulating transcription has been suggested in some contexts, via its interaction with the mediator subunit MED23 (Huang et al, 2012) or the pTEFb kinase (Giraud et al, 2014). Thus, hnRNPL can be multifunctional and more work is required to understand its mechanisms of action.

In line with its multifunctionality, several biological effects have been described for hnRNPL in different cellular contexts. In most cell types, hnRNPL prevents apoptosis. This can reflect a role in

[1]Institut de Recherches Cliniques de Montréal, 110 avenue des Pins Ouest, Montréal, QC H2W 1R7, Canada. [2]Department of Medicine, Division of Experimental Medicine, McGill University, 1001 Boulevard Decarie, Montreal, QC H4A 3J1, Canada. [3]Molecular Biology Programs, Université de Montréal, C.P. 6128, succ. Centre-ville, Montréal, QC H3C 3J7, Canada. [4]Department of Medicine, Université de Montréal, C.P. 6128, succ. Centre-ville, Montréal, QC H3C 3J7, Canada. [5]Département de microbiologie, infectiologie et immunologie, Université de Montréal, 2900 Boul Edouard-Montpetit, Montréal, QC H3T 1J4, Canada. ✉E-mail: javier.di.noia@ircm.qc.ca

preventing p53 activation. as observed in mouse fetal liver and embryonic stem cells, or some human cell lines (Gaudreau et al, 2016a; Li et al, 2015; Seo et al, 2017; Siebring-Van Olst et al, 2017), or preserving protein levels of antiapoptotic BCL2, as reported for human renal Wilms tumors and prostate cancer cell lines (Luo et al, 2019; Zhou et al, 2017). Deletion of hnRNPL in T cells causes aberrant CD45 splicing, increased proliferation, and defects in cell migration (Gaudreau et al, 2012a). Overexpression of hnRNPL in B cell lymphoma promotes survival by protecting the oncogenic IgH-BCL2 fusion transcripts from NMD (Kishor et al, 2019). In the CH12 mouse B cell lymphoma line, hnRNPL was shown to be required for efficient antibody isotype switching from IgM to IgA (Hu et al, 2015). However, hnRNPL has not been studied in normal B cells, and its role during physiological class switch recombination (CSR) remains to be tested. As a cofactor of SETD2, the methyltransferase catalyzing H3K36me3, hnRNPL contributes to maintaining H3K36me3 levels (Bhattacharya et al, 2021a; Yuan et al, 2009a). While reduced Setd2 levels in B cells confers a growth advantage and predisposes to B cell lymphoma, Setd2 is dispensable for CSR (Leung et al, 2022; Begum et al, 2012), suggesting Setd2-independent roles for hnRNPL. Other functions of hnRNPL in B cell biology and its relevance for the antibody response in vivo have not been studied.

The hnRNP family members exist as paralogous pairs that can have redundant functions (Geuens et al, 2016). About 50% of splicing events seem to be regulated by more than one hnRNP (Huelga et al, 2012). Accordingly, hnRNPL and its paralog hnRNPLL recognize CA-rich sequences and have overlapping as well as unique functions (Smith et al, 2013; Fei et al, 2017a). hnRNPLL has an important role in plasma cell differentiation but not in B cell activation (Chang et al, 2015; Yabas et al, 2021). The RNA recognition motif of hnRNPL is also similar to the one in PTBP1 (hnRNPI) (Geuens et al, 2016), which is induced and regulates alternative splicing in germinal center B cells (Monzón-Casanova et al, 2018). These similarities raise the question of to what extent hnRNPL, hnRNPLL, and PTBP1 might be redundant in B cells in vivo.

Here, we find that hnRNPL is essential during B cell activation. The loss of hnRNPL in B cells hinders the germinal center reaction and antibody responses in mice. hnRNPL-deficient B cells undergo cell cycle arrest and apoptosis and have additional intrinsic defects in antibody class switching. A comparative analyses of gene expression and splicing changes in mouse B cells and eight other cell types upon hnRNPL depletion identifies cell type-specific and conserved mechanisms regulated by hnRNPL. Regulating the expression of MYC and E2F transcriptional programs, as well as limiting lncRNA levels, are conserved functions of hnRNPL. Despite little overlap in the specific gene changes among hnRNPL-deficient cell types, we identify conserved hnRNPL-controlled exon exclusion events in genes encoding histone modifiers like KDM6A, NSD2, and SIRT1. These alternative splicing events have functional consequences, and we verify global changes in two histone marks in hnRNPL-deficient B cells, which likely contribute to deregulating gene expression and to other phenotypes. We further show that hnRNPL is needed to maintain mitochondrial function and prevent the accumulation of ROS in resting and activated B cells. Collectively, we identify both B-cell specific, as well as conserved roles for hnRNPL that provide insight into the mechanisms by which it regulates cell biology.

## Results

### hnRNPL is required for the antibody response

We first determined that *Hnrnpl* was expressed in the mouse peripheral B cell subpopulations (Fig. 1A). In contrast, its paralog *Hnrnpll* was not expressed in the quiescent follicular (FO), marginal zone (MZ), or proliferative germinal center (GC) B cells, and was only expressed in plasmablast and plasma cells, albeit at 3–4-fold lower levels than *Hnrnpl* (Fig. 1A). *Ptbp1* expression was substantially higher than *Hnrnpl* in all B cell stages, albeit both genes were relatively similarly expressed in B cells activated ex vivo with T cell mimicking stimuli (Fig. 1A).

To assess the relevance of hnRNPL in mature B cells, we deleted hnRNPL using the *CD21-cre* system that deletes mostly in peripheral B cells (Kraus et al, 2004). We noticed that *CD21-cre Hnrnpl^{F/F}* mice were smaller than controls (Fig. EV1A), suggesting that *hnRNPL* was important for additional tissues in which CD21-cre excises during development (Schmidt-Supprian and Rajewsky, 2007). *CD21-cre Hnrnpl^{F/F}* mice had smaller spleens (Fig. EV1A) and fewer splenocytes (Fig. 1B), correlating with fewer total B and GC B cells (Fig. EV1B,C). The proportions of total B and T (Fig. 1C), as well as newly formed (NF), FO, and GC B cells were not significantly affected, while MZ B cell frequency was significantly reduced (Fig. EV1D). After immunization with the T cell-dependent antigen NP-OVA, *CD21-cre Hnrnpl^{F/F}* mice produced ~8-fold less anti-NP IgG1 than *CD21-cre* controls (Fig. 1D).

### An activated B cell-intrinsic role for hnRNPL supporting antibody responses

To identify B cell autonomous phenotypes of hnRNPL deficiency, we performed mixed bone marrow (BM) chimera experiments to generate mice in which only peripheral B cells have deleted *hnRNPL*. BM from *µMT* mice, which are incapable of producing B cells (Kitamura et al, 1991), was mixed in a 1:1 ratio with BM from either *CD21-cre Hnrnpl^{F/+}* (controls) or *CD21-cre Hnrnpl^{F/F}* mice and transferred intravenously into lethally irradiated WT mice (Fig. 1E). The body and spleen weight, as well as splenocyte numbers in chimeric mice with *CD21-cre Hnrnpl^{F/F}* BM, were similar to the controls (Fig. EV1E). However, the frequency of total B cells was lower with a concurrent increase in the proportion of T cells, due primarily to a reduction in B cell numbers, while T cell numbers were not affected (Fig. 1F). The proportion of MZ B cells was again significantly reduced, but NF, FO and GC B cells were unaffected (Fig. 1G). However, upon immunization with NP-OVA, chimeric mice reconstituted with *CD21-cre Hnrnpl^{F/F}* BM cells showed a similar ~8-fold defect in antibody response (Fig. 1H), indicating a cell-intrinsic defect caused by hnRNPL deficiency in FO B cells when responding to antigenic challenge.

To better understand the role of hnRNPL in activated B cells, we isolated resting B cells from *CD21-cre Hnrnpl^{F/F}* mice and activated them with LPS + IL-4. Compared to controls, hnRNPL-deficient B cells showed a clear defect in expansion 3 days after stimulation (Fig. 1I), which contrasted with the apparently normal formation of GC observed in vivo. However, while hnRNPL protein levels were reduced in the resting *CD21-cre Hnrnpl^{F/F}* B cells indicating efficient *Hnrnpl* excision, the activated B cells from these mice paradoxically showed WT levels of hnRNPL 4 days after stimulation (Fig. 1J).

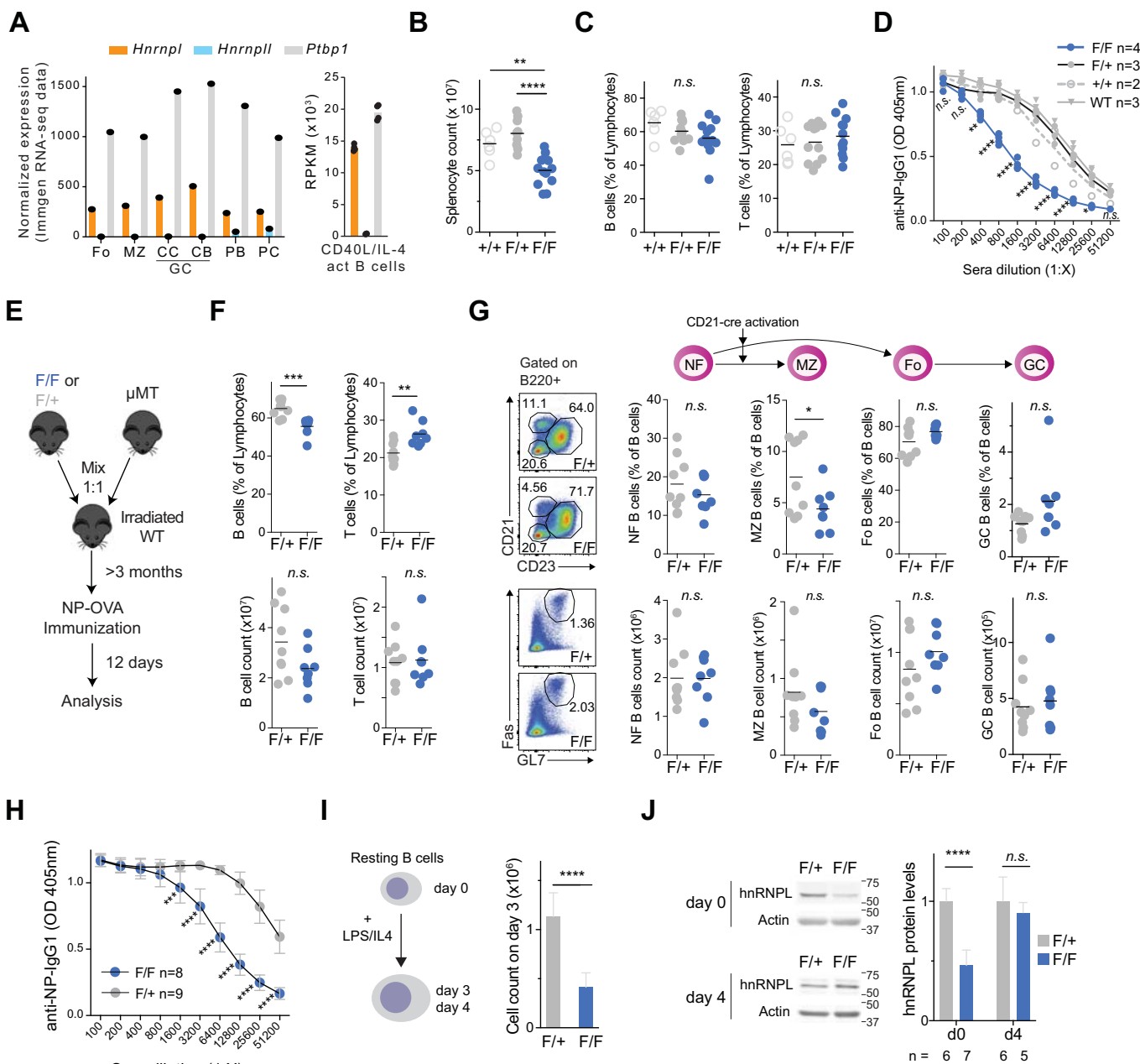

**Figure 1. hnRNPL loss in peripheral B cells leads to a cell-intrinsic defect in antibody response.**

(A) Expression levels of *Hnrnpl*, *Hnrnpll*, and *Ptbp1* in mature B cell subpopulations. Follicular (Fo), marginal zone (MZ), centrocyte (CC), and centroblast (CB) germinal center (GC) B cells, plasmablasts (PB), and plasma cells (PC) from Immgen (GSE127267), as well as in in vitro activated B cells (mean of four samples, GSE120309). (B) Splenocyte counts (B) in *CD21-cre* (+/+) (n = 6), *CD21-cre Hnrnpl^{F/+}* (F/+) (n = 9) and *CD21-cre Hnrnpl^{F/F}* (F/F) (n = 12) mice 11 days post-immunization with NP-OVA. (C) Splenic B and T cell proportions for the mice in (B) (n = 6 +/+, n = 12 F/+, n = 12 F/F). (D) Serum anti-NP IgG1 titers for n mice per group, as indicated. Genotypes labeled as in (B), with WT mice being *Hnrnpl^{F/+}* mice. (E) Scheme of mixed bone marrow (BM) chimera experiment. Mice were reconstituted with BM cells from μMT mice and either *CD21-cre Hnrnpl^{F/+}* (F/+) or *CD21-cre Hnrnpl^{F/F}* (F/F) mice in a 1:1 ratio. (F) Proportion and number of splenic B and T cells in mice made as in (E), 12 days after immunization with NP-OVA, (n = 9 F/+, n = 8 F/F). (G) Subpopulation of newly formed (NF), MZ, FO and GC B-cell subpopulations, for the mice in (F). (H) Anti-NP IgG1 titers in the serum of mice from (F) 11 days after immunizations. (I) Ex vivo activation of splenic B cells purified from *CD21-cre Hnrnpl^{F/+}* (F/+) and *CD21-cre Hnrnpl^{F/F}* (F/F) mice and cell counts after 3 days (n = 9 mice per group). (J) Representative western blot and quantification of hnRNPL protein levels in B cells, resting (d0) or 4 days post-activation (d4), as in (I). Data information: In (B–D, F, G), data were presented for individual mice with lines indicating mean values. (H–J) Data were presented as mean ± SD. Differences in group means were considered significant when P value p < 0.05, as analyzed by: (B, C), one-way ANOVA with post hoc Tukey's multiple comparison test; (D, H) Two-way ANOVA test with Sidak's multiple comparison test; (F, G, I, J), unpaired two-tailed *t*-test with Welch's correction (*p < 0.05, **p < 0.01, ***p < 0.001, ****p < 0.0001). Data compiled from (B, C) four experiments, (D, F–H) 2 experiments, (I) three experiments, (J) indicated the number of replicates, each a different mouse.

This indicated that cells which escaped Cre-mediated excision outgrew the *Hnrnpl*-deleted cell population over time, implying a major disadvantage for hnRNPL-deficient B cells in culture following activation.

We conclude that hnRNPL has a non-redundant B cell-intrinsic role in maintaining MZ B cell homeostasis and a major post-activation role in FO B cells, likely supporting cell fitness, which is required for antibody responses.

## MZ and GC B cells are hypersensitive to hnRNPL loss in vivo

To monitor B cell fitness in vivo, we performed competitive mixed BM chimera experiments. To track hnRNPL excision in B cells, we introduced a *Rosa^{mT/mG}* cre-reporter allele into *CD21-cre Hnrnpl^{F/F}* mice. This allele at the *Rosa26* locus encodes constitutive expression of membrane-targeted tdTomato (mT). Upon cre recombinase expression, the mT cassette is excised, allowing for the expression of membrane GFP (mG) from a downstream cassette (Muzumdar et al, 2007), thus labeling cells with Cre activity and their progeny with GFP (Fig. 2A). BM cells from CD45.1+ WT mice were mixed with BM from either CD45.2+ *Rosa^{mT/mG} CD21-cre Hnrnpl^{F/F}* or the control *Rosa^{mT/mG} CD21-cre Hnrnpl^{F/+}* mice in 1:1 ratio and intravenously injected into irradiated WT recipient mice (Figs. 2B and EV1F). Following reconstitution and immunization, the number of total B cells and the overall proportions of total B, as well as NF, FO, MZ, and GC B cells were similar in both groups of mice (Fig. EV1G,H).

While both groups of reconstituted mice showed similar proportions of CD45.1 and CD45.2 mature B cell subsets, GFP+ cells were significantly underrepresented among B cells of the animals that received BM from *Rosa^{mT/mG} CD21-cre Hnrnpl^{F/F}* mice, demonstrating that hnRNPL was intrinsically required for mature B cell homeostasis (Fig. 2C). Further examining splenic B cell subpopulations showed the onset of Cre recombinase activity after the NF B cells, as expected for *CD21-cre* (Schmidt-Supprian and Rajewsky, 2007), and no significant defect in hnRNPL-deficient FO B cells (Fig. 2D). In contrast, hnRNPL-deficient (GFP+) MZ and GC B cells were severely underrepresented compared to hnRNPL WT cells in the chimeric mice reconstituted with WT plus *Rosa^{mT/mG} CD21-cre Hnrnpl^{F/F}* mixed BM (Fig. 2D). The defect was stronger in GC B cells, with even the control *Hnrnpl^{F/+}* cells being outcompeted by WT cells in the control chimeras (Fig. 2D). Considering that MZ B cells exist in a primed or pre-activated state (Lopes-Carvalho et al, 2005) and GC B cells originate from activated FO cells, these results together with the counterselection of activated hnRNPL-deficient B cells observed ex vivo (Fig. 1J) indicate that hnRNPL is critically required during B cell activation.

## hnRNPL is necessary for isotype switching to IgG1

Isotype switching occurs relatively early after B cell activation, and hnRNPL depletion reduces isotype switching to IgA in the CH12 mouse B cell line (Hu et al, 2015). To confirm this observation in primary B cells, which could be contributing to the antibody response defect, we activated B cells from the *Rosa^{mT/mG}* cre-reporter mice ex vivo with LPS and IL-4, which allowed us to monitor isotype switching to IgG1 specifically in hnRNPL-deficient B cells.

Since CSR is a stochastic process linked to cell division (Hodgkin et al, 1996), we also used a tracking dye that allows enumerating cell generations after activation. hnRNPL loss caused a severe defect in B cell proliferation (Fig. 3A). Nonetheless, comparing cells that had undergone the same number of generations confirmed that hnRNPL loss reduced antibody class switching to IgG1 (Fig. 3A).

CSR is initiated by DNA deamination by activation-induced deaminase (AID), which requires transcription of *IgH* Switch (S) regions, producing sterile germline transcripts (GLT) (Methot and Di Noia, 2017). RNA-seq data (see next section) showed that while the donor Sμ GLT was unchanged, the acceptor Sγ1 GLT levels were reduced 2-fold in hnRNPL-deficient activated B cells (Fig. 3B). This contrasts with the CH12 B cell line, in which the acceptor Sα GLT levels were not affected (Hu et al, 2015). However, Sα is constitutively transcribed in CH12 cells (Zhang et al, 2019), while acceptor GLTs in primary B cells are induced by cytokine stimuli, suggesting a potential reason for this discrepancy. To determine whether hnRNPL might be acting directly or indirectly at the *IgH* during CSR, we determined its occupancy by ChIP. hnRNPL localized primarily to the Sμ GLT region in both resting and activated WT B cells (Fig. 3C). At the Sγ1 region, hnRNPL was detectable only in activated B cells, when this region is transcribed, but at very low levels (Fig. 3C). We conclude that hnRNPL associates to the *IgH*, largely at the Sμ region, and is required for CSR in primary B cells. The reduced Sγ1 GLT levels in hnRNPL-deficient B cells provides another way in which hnRNPL contributes to CSR in primary B cells, which is not mutually exclusive with the previously proposed role in the DNA end-joining step of CSR (Hu et al, 2015). This result also raised the possibility that hnRNPL might regulate the expression of other genes in B cells (see below).

## hnRNPL loss causes apoptosis and cell cycle arrest of activated B cells

To characterize the proliferation and survival defects upon activation in hnRNPL-deficient B cells, we isolated splenic B cells from the *Rosa^{mT/mG} CD21-cre Hnrnpl^{F/F}* and *Rosa^{mT/mG} CD21-cre Hnrnpl^{F/+}* control mice and stimulated them with LPS/IL-4. The proportion of GFP+ *Rosa^{mT/mG} CD21-cre Hnrnpl^{F/F}* B cells (hereafter hnRNPL-deficient B cells) dropped starting at day 2 post-activation, and by day 4, only tdTomato+ cells (i.e., cells without hnRNPL excision) remained (Fig. 4A, EV2A). Accordingly, hnRNPL-deficient B cells showed ~2-fold more apoptosis than control cells following activation, as measured by Annexin V staining (Fig. 4B), compared to heterozygous control cells. hnRNPL-deficient B cells were also smaller and less granular, indicating defective blasting following activation (Fig. EV2B). The cell division dye staining confirmed that live hnRNPL-deficient B cells displayed a severe proliferation defect compared to control cells (Fig. 4C), as observed in CSR assays. This was caused by a G1 cell-cycle arrest with failure to enter S-phase (Fig. 4D). We confirmed the effect on cell proliferation by using the induced germinal center B cells (iGB) cell culture system, in which B cells grow exponentially following activation by CD40L and BAFF displayed on the surface of feeder cells plus IL-4 (Nojima et al, 2011). Cell counts, cell proliferation tracing, and cell cycle assays mirrored the results from the LPS/IL-4-activated B cells (Fig. 4E–G). We conclude that hnRNPL protects B cells from

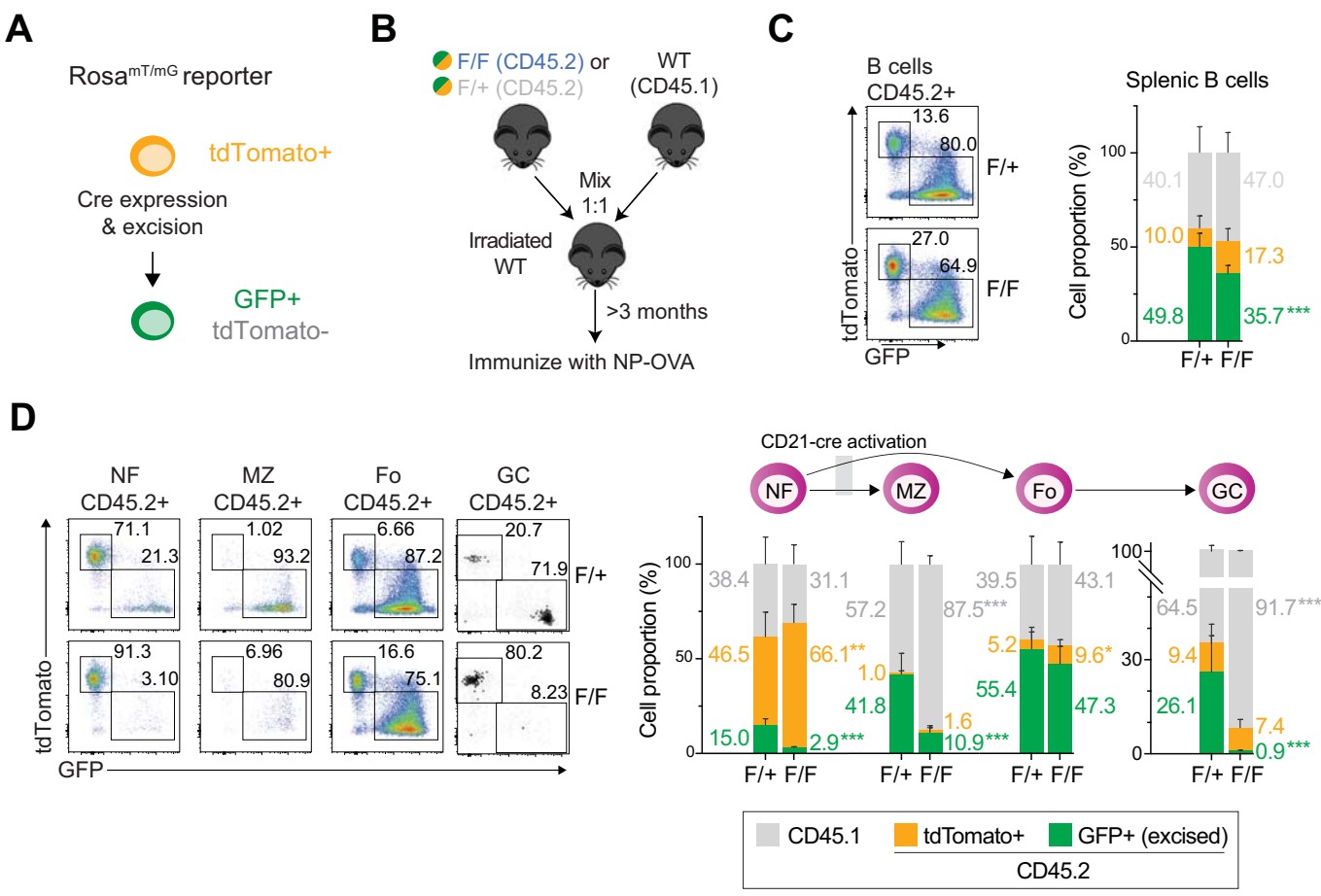

**Figure 2. MZ and GC B cells are hypersensitive to hnRNPL loss.**

(A) Scheme of Rosa^mT/mG reporter mouse model. (B) Scheme of competitive BM chimera experiments. Irradiated WT mice were reconstituted with CD45.1 WT BM mixed 1:1 with BM from either CD45.2 *Rosa^mT/mG CD21-cre Hnrnpl^F/+* (F/+) or *Rosa^mT/mG CD21-cre Hnrnpl^F/F* (F/F). (C) The proportion of splenic B cells in the recipient mice originating from BM cells of CD45.1 WT (gray) or CD45.2 that had evidence of Cre activity (GFP+, green) or not (tdTomato+, orange), with representative flow cytometry plot of the latter analysis, (n = 8 mice each group). (D) As in (C) for Splenic NF, MZ, FO, and GC B cell populations in the same mice. Data information: In (C, D) data were presented as stacked mean ± SD. Data were compiled from two experiments. For clarity, only significant (p < 0.05) differences by unpaired, two-tailed Mann–Whitney test are indicated (*p < 0.05, **p < 0.01, ***p < 0.001).

apoptosis and is required for cell cycle progression through the G1 checkpoint upon B cell activation.

## hnRNPL sustains transcriptional programs of proliferation and dampens p53 response in B cells

To identify early and potentially causal changes that might explain defects in hnRNPL-deficient B cells, we performed RNA-sequencing on sorted GFP+ cells from *Rosa^mT/mG CD21-cre HnrnplF/F* and *Rosa^mT/mG CD21-cre hnRNPL^+/+* controls 1 day after LPS/IL-4 activation, prior to apoptosis (Fig. 4A,B). Differential gene expression analysis identified 1499 up- and 817 downregulated genes in hnRNPL-deficient cells (adjusted p value cut-off of 0.1 and ≥1.5-fold-change) (Fig. 4H).

Functional annotation of differentially expressed genes by Gene set enrichment analyses (GSEA) using the Hallmark gene sets (Liberzon et al, 2015) and KEGG pathways gene ontology terms reflected cell cycle arrest, highlighting a decrease in MYC, E2F, and the mTORC1 signaling signatures, as well as increased p53 pathway

in hnRNPL-deficient B cells (Figs. 4I and EV2C,D). The combination of these changes could explain proliferation defects in these cells. Indeed, MYC and some E2F transcription factors, as well as mTORC1 signaling, play key roles in cell growth, G1 to S cell cycle progression, and proliferation in activated and germinal center B cells (Nie et al, 2012; Patterson et al, 2021; Hsia et al, 2002; Lam et al, 1998). Following activation, B cells increase the expression of *Myc*, *E2f1*, *E2f3*, and Cyclin D2 (*Ccnd2*) and decrease the expression of *Foxo3* to support cell cycle progression (Nie et al, 2012; Hsia et al, 2002; Lam et al, 1998; Hinman et al, 2009; Yusuf et al, 2004; Pae et al, 2021). hnRNPL-deficient B cells failed to enact these transcriptional changes (Fig. 4J) and showed increased FoxO signaling signatures (Fig. EV2C). Moreover, although p53 (*Trp53*) mRNA levels did not increase, *Cdkn1a*, which is a p53 target encoding the negative cell cycle regulator and CDK inhibitor p21 (Engeland, 2018), and Linc-p21 (*Tp53cor1*), which positively regulates p21 expression (Groff et al, 2016), were increased more than 10-fold (Fig. 4J). Upregulation of *Cdkn1a* in response to activation in the hnRNPL-deficient B cells was confirmed by

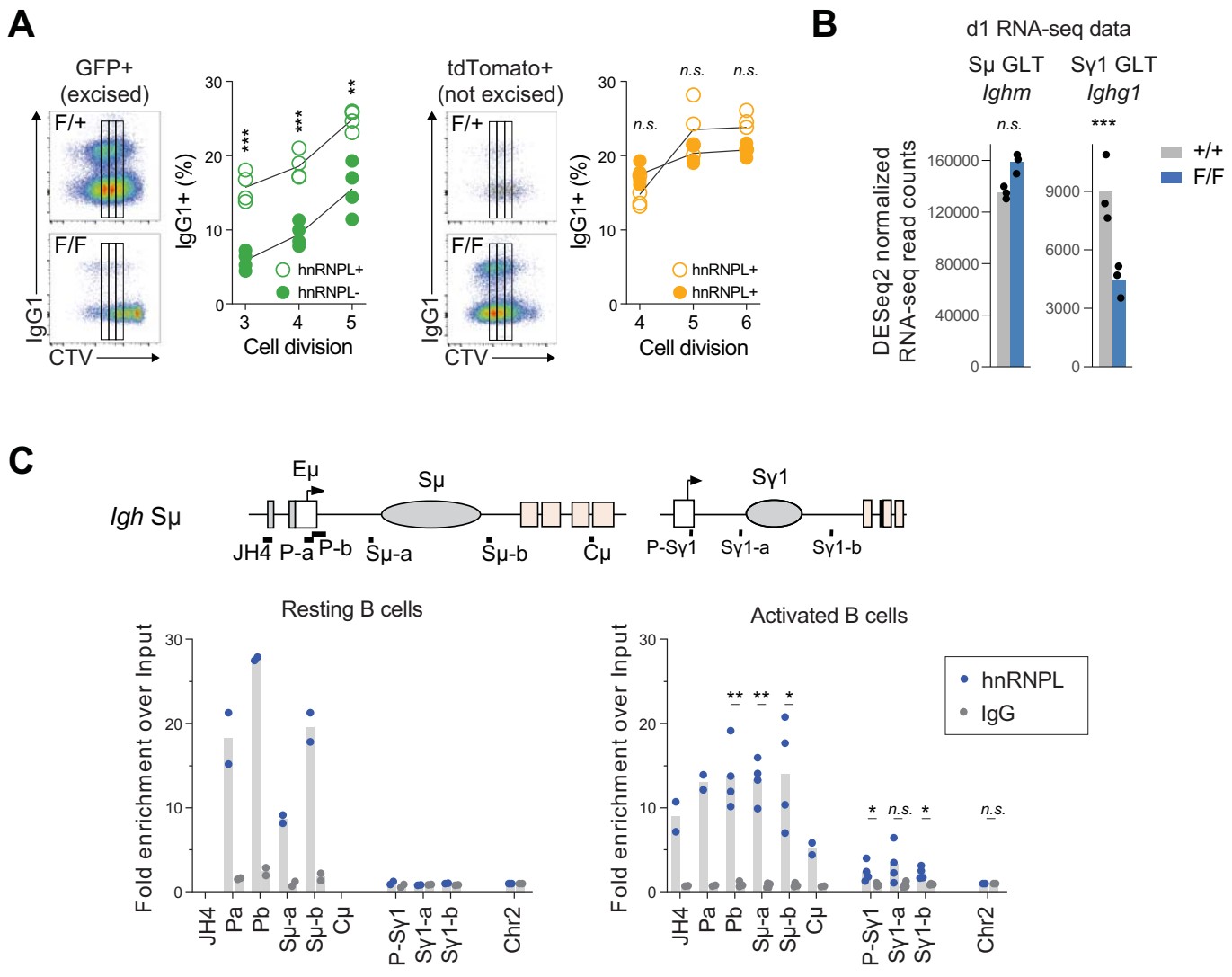

**Figure 3. hnRNPL supports class switch recombination.**

(A) Representative flow cytometry plot of isotype switched IgG1+ B cells as a function of cell division, tracked by dilution of CTV staining in GFP+ (green; hnRNPL-excised) and tdTomato+ (orange; non-excised) B cells from *Rosa^mT/mG CD21-cre Hnrnpl^{F/+}* (F/+) or *Rosa^mT/mG CD21-cre Hnrnpl^{F/F}* (F/F), 4 days after activation with LPS/IL-4. The plots compile proportions of IgG1+ cells per cell division gate as shown in the representative plots ($n = 4$ mice each group). (B) Normalized read counts from RNA-seq of Sμ and Sγ1 germline transcripts (GLT) in WT (+/+) and hnRNPL-deficient (F/F) B cells 1 day after activation with LPS/IL-4 ($n = 3$ mice each group). (C) hnRNPL occupancy by ChIP-qPCR at indicated amplicons of the *IgH* locus in WT splenic B cells resting (d0) ($n = 2$ mice each group) or activated with LPS/IL-4 for 2 days (d2), ($n = 4$ mice each group). Data information: In (A–C), data were presented with individual mice values with mean shown as lines or bars. Results were compiled from two experiments, except resting B cells in (C), one experiment. Statistical significance ($p < 0.05$) by (A) two-way ANOVA with Sidak's multiple comparison test, (B) DESeq *P*adj value indicated, (C) two-tailed paired *t*-test (*$p < 0.05$, **$p < 0.01$, ***$p < 0.001$). Statistical test was not performed for amplicons with only two biological replicates.

RT-qPCR (Fig. EV2E). We also verified that apoptosis was a response to activation by measuring several pro- and antiapoptotic genes in resting versus activated *CD21-cre Hnrnpl^{F/F}* cells (Fig. EV2E). In line with the p53 signature upregulation and apoptosis phenotype, p53 protein levels were increased in hnRNPL-deficient resting and activated B cells (Fig. 4K).

We conclude that alterations in the expression of key transcription factors and cell cycle regulators, including incomplete activation of the MYC and E2F transcriptional programs together with p53 activation, can at least partly explain reduced proliferation and apoptosis of hnRNPL-deficient B cells upon activation.

## A conserved role for hnRNPL in MYC and E2F transcriptional programs

Taking advantage of the availability of multiple RNA-seq datasets in hnRNPL-deficient cells, we sought to identify conserved functions of hnRNPL. Thus, we compared our RNA-seq data from activated B cells to datasets from 8 other different hnRNPL-deficient cell types: one from primary human keratinocytes (Li et al, 2021a) (Data ref: Li et al, 2021b), five from different human cell lines (ENCODE Project Consortium, 2012a; Davis et al, 2018; Fei et al, 2017a; McCarthy et al, 2021a; Bhattacharya et al, 2021a) (Data ref: Bhattacharya et al, 2021 for HEK293, McCarthy et al, 2021b for

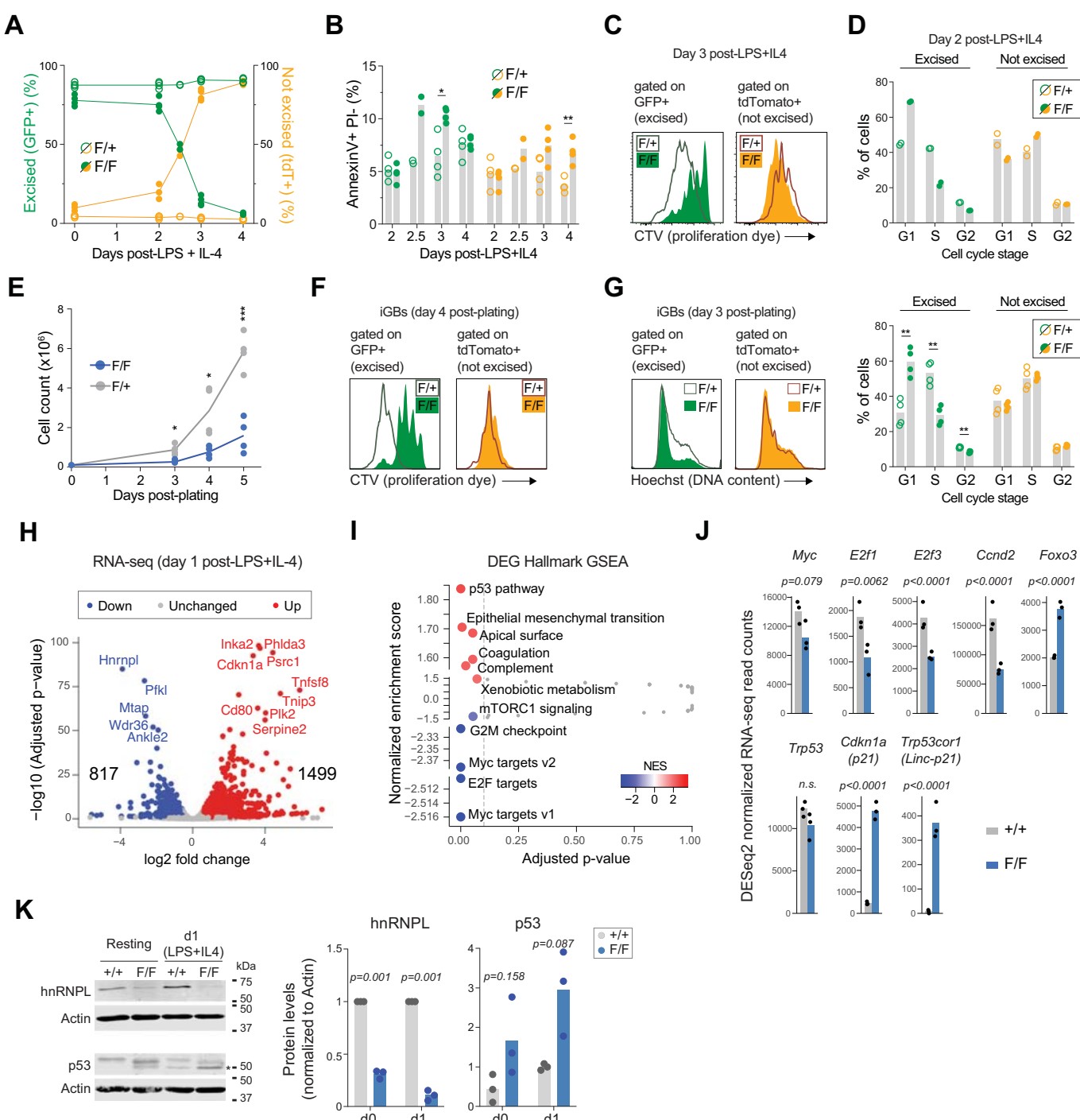

BJ, Fei et al, 2017b for LNCaP, ENCODE Project Consortium, 2012b for K562 and HepG2) and two from mouse tissues (Gaudreau et al, 2012a, 2016a) (Data ref: Gaudreau et al, 2016b for Fetal liver, Gaudreau et al, 2012b for thymocytes). In all cell types, *Hnrnpl* expression was higher than its homolog *Hnrnpll*, with the highest ratio in mouse splenic B cells (Fig. EV3A). In most cases, *Hnrnpll* expression did not increase to compensate for *Hnrnpl* loss (Fig. EV3A).

To compare the biological processes affected by hnRNPL depletion in each cell type, we performed GSEA enrichment analysis for the Hallmark gene signatures, as done for B cells. hnRNPL-deficient thymocytes exhibited much fewer differentially expressed genes than any other cell type (see Fig. EV3D), resulting in no Hallmark signatures significantly enriched in these cells; but all other cell types showed enrichments (Fig. 5A). Remarkably, the "MYC targets v1" and "E2F targets" gene sets were downregulated in 6 of the 8 informative datasets and, notably, increased in HEK293 cells (Fig. 5A). For the most part, these alterations could not be attributed to downregulation of MYC or E2Fs transcripts (Fig. EV3B), except perhaps partly in the hnRNPL-deficient B cells,

**Figure 4. hnRNPL loss causes apoptosis and cell cycle arrest of activated B cells.**

(A) The proportions of GFP$^+$ (green; *Hnrnpl*-excised) and tdTomato$^+$ (orange; non-excised) B cells from *Rosa$^{mT/mG}$ CD21-cre Hnrnpl$^{F/+}$* (F/+) or *Rosa$^{mT/mG}$ CD21-cre Hnrnpl$^{F/F}$* (F/F) in culture over time after ex vivo activation with LPS/IL-4, ($n = 4$ mice each group, except at day 2.5 when $n = 2$ mice each group). (B) Apoptosis estimated by Annexin V flow cytometry staining of GFP$^+$ and tdTomato$^+$ B cells from F/+ and F/F mice, cultured as in (A), ($n = 4$ mice each group, except at day 2.5 when $n = 2$ mice each group). (C) Representative flow cytometry plots to estimate proliferation by dilution of CellTrace Violet (CTV) dye of GFP$^+$ and tdTomato$^+$ B cells from F/+ and F/F mice, cultured as in (A). (D) Cell cycle phases quantified from flow cytometry staining of DNA content of GFP$^+$ and tdTomato$^+$ B cells from F/+ and F/F mice ($n = 2$ mice each group), cultured as in (A). (E) Growth curves of induced germinal center B (iGB) cells derived from F/+ and F/F mice splenic B cells cultured on feeder cells expressing CD40L and BAFF, supplemented with IL-4, ($n = 4$ mice in each group). GFP$^+$ and tdTomato$^+$ cells were not discriminated in this experiment. (F) Representative flow cytometry plots showing CTV dilution of GFP$^+$ and tdTomato$^+$ F/+ and F/F iGBs, cultured as in (E). (G) Cell cycle profile of cells in (F) and proportion of cells in each cell cycle stage ($n = 4$ mice each group). (H) RNA-seq volcano plot showing differential gene expression in hnRNPL-deficient (GFP$^+$ cells from *Rosa$^{mT/mG}$ CD21-cre Hnrnpl$^{F/F}$*) versus WT (GFP$^+$ cells from *Rosa$^{mT/mG}$ CD21-cre*) splenic B cells activated ex vivo for 1 day with LPS/IL-4. (I) Plot of normalized enrichment scores (NES) for GSEA Hallmark terms in hnRNPL-deficient over WT cells. Significantly enriched terms are defined and colored by NES. (J) Normalized read counts from RNA-seq of selected genes in WT (+/+) and hnRNPL-deficient (F/F) activated B cells. (K) Representative western blot of hnRNPL and p53 protein levels in B cells, purified from *CD21-cre* (+/+) and *CD21-cre Hnrnpl$^{F/F}$* (F/F) mice, resting (d0) or activated with LPS/IL-4 for 24 h (d1), ($n = 3$ mice per group). The asterisk indicates the p53 band, assigned based on controls run in parallel. Data information: In (A, B, D, E, G, J) data were presented as individual mice (biological replicates) values with means shown by lines or bars, (K) data are presented as mean normalized quantification. Results are compiled from (A, B, D, E, G, K) two experiments, (D) one experiment. Statistical significance ($p < 0.05$) by (B, E, G) unpaired, two-tailed $t$-test with Welch's correction (*$p < 0.05$; **$p < 0.01$; ***$p < 0.001$), (H) DEseq2, (I) GSEA, (J) DESeq2 *Padj* value indicated with $p < 0.1$ considered significant, (K) one sample $t$-test for hnRNPL and unpaired two-tailed $t$-test for p53.

in which *E2f1* and *E2f3* expression was reduced by ~50% (Fig. 4J). The "G2M checkpoint" gene set was also downregulated in the same cell types in which MYC and E2F targets were downregulated, consistent with a link between these programs. Most other gene sets were changed in three to five datasets (Fig. 5A), likely reflecting more context-dependent roles for hnRNPL. For example, "mTORC1 signaling" was downregulated in activated B cells and three other datasets. The "p53 pathway" was increased only in mouse fetal liver and activated B cells (with the caveat that p53 activation might be impaired in some of the immortalized human cell lines).

Another conserved feature across hnRNPL-deficient cells that emerged from our analysis was that a large proportion of the upregulated genes were long non-coding RNAs (lncRNAs), ranging from ~45% in K562 cells to ~20% in activated B cells, with ~3 times more lncRNAs being upregulated than downregulated in all datasets (Fig. 5B).

To explore the possibility that hnRNPL could directly affect transcription, we reanalyzed hnRNPL ChIP-seq data from K562 cells, in which hnRNPL has been suggested to associate with promoters and enhancers via interactions with nascent RNAs (Xiao et al, 2019). Accordingly, 87% of hnRNPL ChIP-seq peaks were located within annotated transcripts, and ~70% of those were within 2 kb of a transcription start site (TSS) (Fig. 5C). MYC target genes were over-represented in these hnRNPL-occupied genes, and, in fact, most hnRNPL-occupied genes were MYC targets in K562 cells (Fig. 5D,E). However, only a small fraction of hnRNPL-occupied genes were downregulated in hnRNPL-deficient K562 cells (Fig. 5E). Consistent with this, hnRNPL was associated with the *IgH* Sμ region in B cells, albeit hnRNPL loss did not affect the corresponding transcript levels (Fig. 3B,C). Similarly, there was no clear correlation between hnRNPL occupancy and the transcript level of a few additional B cell genes (*Pax5*, *Pim1*, *Apex1*, and *Myc*, selected for being AID "off-targets" (Casellas et al, 2016)) in resting or activated WT cells (Fig. EV3C), nor was their expression significantly changed in hnRNPL-deficient versus WT activated B cells (Dataset EV1). Thus, without excluding that it might happen in certain instances, our analysis does not support a general role of hnRNPL in directly regulating transcription via locus occupancy.

We then looked for common differentially expressed genes in hnRNPL-deficient cell types. However, most of the gene expression changes upon hnRNPL depletion were cell type specific (Fig. EV3D; Dataset EV2). Thus, there were only 28 species-specific changes, and only two transcripts (apart from *HNRNPL*) were differentially expressed in ≥6 datasets across species: *CUL7* and *ZBTB40* (Fig. EV3D). Interestingly, in line with the global lncRNA upregulation, six genes upregulated in all human datasets were lncRNAs (Fig. EV3D), including the transcriptional regulator *NEAT1* (Gu et al, 2022).

We conclude that hnRNPL influences the status of the MYC- and E2F-dependent transcriptional programs, usually but not always, by promoting their maintenance and that it globally dampens lncRNA expression. These effects are unrelated to any stable occupancy of the corresponding loci by hnRNPL. We also verify cell type-specific functions for hnRNPL.

## hnRNPL has a non-redundant role in regulating alternative splicing in activated B cells

Having ruled out a relationship between hnRNPL occupancy and gene expression changes in hnRNPL-deficient B cells, we analyzed alternative mRNA splicing.

We found 4733 differential splicing events in 2427 genes in hnRNPL-deficient versus control B cells (Fig. 6A; Dataset EV3) at ≥10% inclusion level difference with events supported by at least ten junction reads on average using rMATS (see Methods). Importantly, there was minimal overlap between the transcripts showing significantly altered splicing events and the differentially expressed genes between hnRNPL-deficient and WT B cells (Fig. 6B). We note that MXEs are rare events, and rMATS may overestimate them compared to estimates done by a different method in human cells (Hatje et al, 2017). Concordant with hnRNPL usually regulating alternative splicing (Cole et al, 2015; McClory et al, 2018; Motta-Mena et al, 2010), the most common type of event affected by hnRNPL loss in B cells was exon skipping (Fig. 6A), which is the most common type of alternative splicing event (i.e. alternative exon inclusion or exclusion). Functional annotation indicated that genes whose splicing was affected by hnRNPL loss were implicated in numerous biological processes and pathways. Autophagy, DNA recombination, and double-strand break repair were among the most enriched gene sets, the latter including several genes encoding for proteins participating in DNA

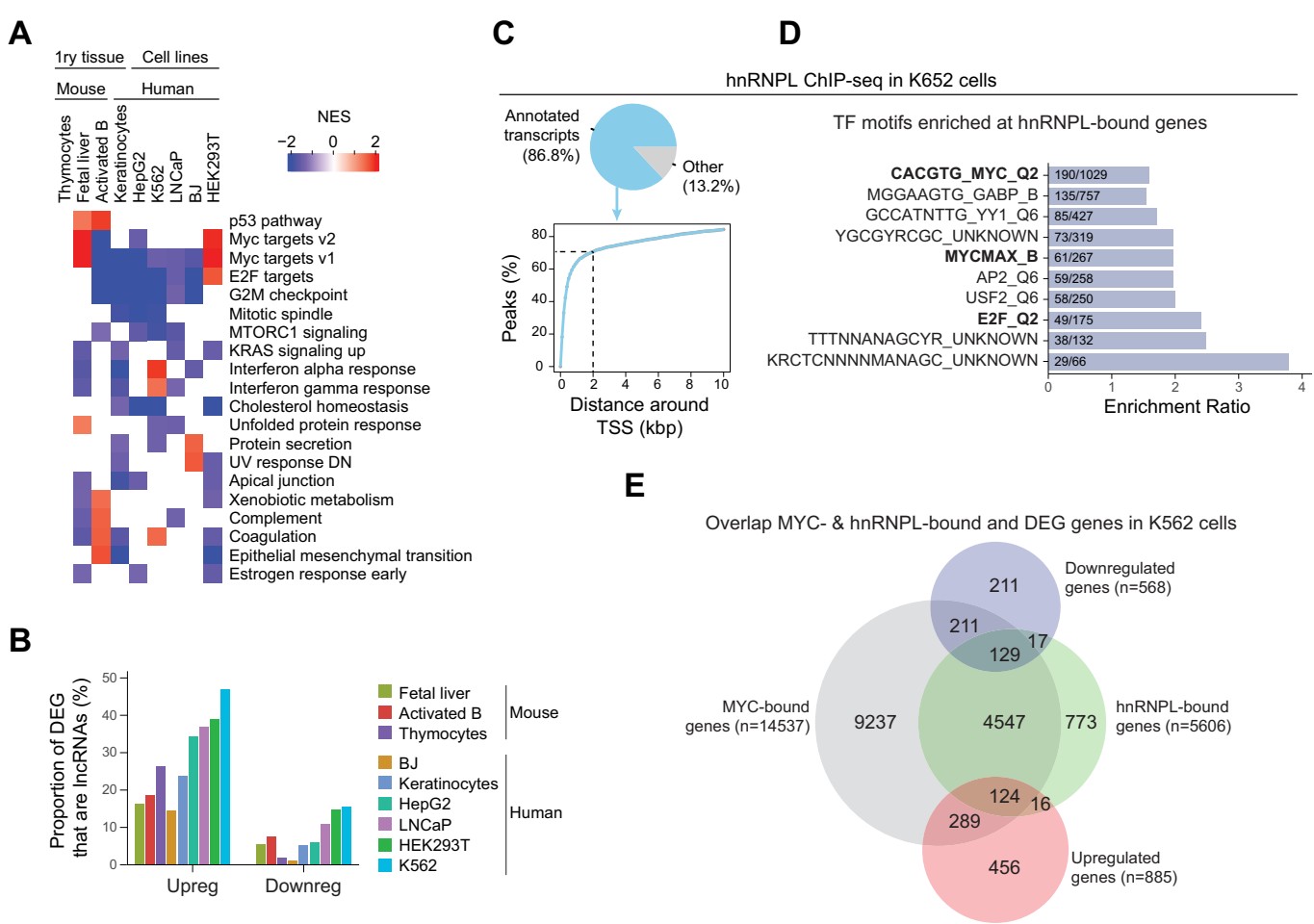

**Figure 5. Conserved roles for hnRNPL in mouse and human cells.**

(A) Comparison of Hallmark gene sets differentially enriched across cell types. (B) Proportion of lncRNAs in the up- and downregulated genes in hnRNPL-depleted cells. (C) Pie chart showing the proportion of hnRNPL ChIP-seq peaks at annotated transcripts and plot of the cumulative proportion of hnRNPL peaks as a function of the distance from the nearest transcription start site (TSS), in K562 cells. (D) Transcription factor (TF) motifs enriched at hnRNPL-bound gene promoters (hnRNPL peak ±2 kb from their TSS) in K562 cells. (E) Venn diagrams showing the overlap between MYC- or hnRNPL-occupied loci in K562 cells, and genes significantly upregulated or downregulated in hnRNPL-depleted versus WT K562 cells. Data information: In (A) data were presented as normalized enrichment scores calculated by the GSEA algorithm for gene sets significantly enriched ($Padj$ <0.05) in ≥3 datasets. The "p53 pathway" was also selected because it is relevant for B cells.

end-joining (*Prkdc*, *Atm*, *Shld2*, *Mre11*, etc.) (Fig. 6C; Dataset EV4). Notably, the most significantly enriched biological process was histone modification (Fig. 6C), which included a collection of 125 proteins with well-described roles in transcription regulation, such as transcription factors (*Bcl6*, *Pax5*, etc), lysine demethylases, histone deacetylases, sirtuins, and ubiquitin ligases, 87 of which contained significantly changed skipped exon events (Dataset EV4).

We conclude that hnRNPL plays a major role in regulating alternative splicing in activated B cells that cannot be compensated by other paralogs, which are either not expressed or unable to complement the defects, judging from the large phenotypic alterations noted. The splicing alterations in hnRNPL-deficient B cells seldomly affect the corresponding transcript levels directly, but a preference for regulating splicing of transcripts encoding for proteins that regulate transcription could at least partly explain the effect of hnRNPL on gene expression in B cells.

## hnRNPL has conserved roles in the splicing of transcriptional regulators

Given the large effect of hnRNPL deficiency on pre-mRNA splicing in B cells, which affected transcripts encoding factors capable of altering transcription, we then compared the role of hnRNPL in splicing across cell types to look for potential common mechanisms whereby hnRNPL could regulate gene expression. As in B cells, the most frequently altered type of event, and which affected the most genes, in all hnRNPL-deficient cell types was skipped exon (Dataset EV5). The RNA-seq datasets had different read length, depth, and number of replicates, which affect the sensitivity to detect splicing changes (Fig. EV4A) (see Methods). Nonetheless, functional annotation of the genes with splicing affected by hnRNPL loss across cell types showed that many of the top biological processes identified in B cells were also enriched in other cell types (Fig. 6D). Notably, the "Histone modification" and

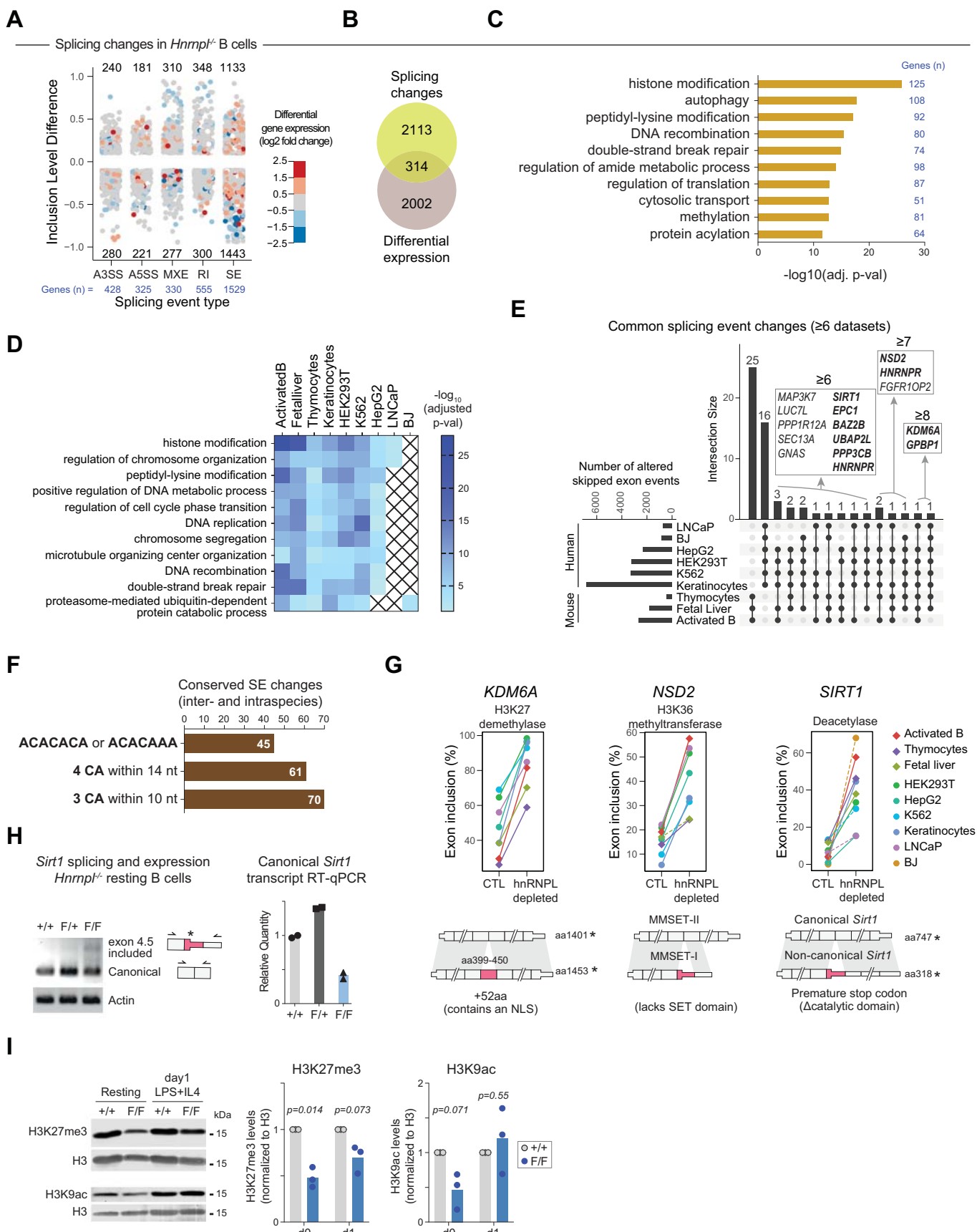

Figure 6. Conserved splicing events in hnRNPL-deficient cells.

(A) Splicing changes in hnRNPL-deficient (GFP$^+$ cells from *Rosa$^{mT/mG}$ CD21-cre Hnrnpl$^{F/F}$*) versus WT (GFP$^+$ cells from *Rosa$^{mT/mG}$ CD21-cre*) cells, activated ex vivo for 1 day with LPS/IL-4. Expression changes of the same genes are also indicated. (B) Venn diagram of the overlap between genes with significant splicing alterations and differentially expressed genes in hnRNPL-deficient versus WT control B cells. (C) Top ten enriched biological process GO terms among genes with significant splicing alterations in hnRNPL-deficient B cells. (D) Enrichment of the indicated biological processes among genes with significantly affected splicing for each cell type. X symbols indicate GOs that were not significantly enriched in the corresponding dataset. (E) Conservation of splicing events significantly changed upon hnRNPL depletion among the indicated cell types. All non-zero intersections of ≥6 datasets are shown. The inset shows the genes with conserved splicing changes in ≥6 mouse and human cell types. (F) The number of splicing events identified in (E) that contain each of the indicated hnRNPL-binding motifs. (G) Examples of conserved hnRNPL-regulated splicing events from (E), showing exon inclusion levels in the control (CTL) and hnRNPL-depleted cell types. The transcript schemes indicate the exons (in red, the exon more included in hnRNPL-deficient cells), positions of amino acids coded by the included exon, and stop codons in the respective human transcripts. (H) Electrophoresis gel and quantification of RT-PCR products for the *Sirt1* canonical and alternatively spliced non-canonical transcript, including the intermediate exon, in resting splenic B cells from *CD21-cre* (+/+), *CD21-cre Hnrnpl$^{F/+}$* (F/+) and *CD21-cre Hnrnpl$^{F/F}$* (F/F) mice. (I) Representative western blot and quantification of the global levels of selected histone marks in resting and activated (LPS/IL-4 for 24 h) mouse splenic B cells purified from the indicated mice (as in H). Data information: In (A, G) data were mean inclusion level difference between hnRNPL-deficient and control B cells, for transcripts with splicing events within ≥10% and ≤95% difference between genotypes, FDR <0.1 and average junction read count ≥10, as calculated by rMATS, except for (G) dotted lines that indicate events that were detected at FDR <0.2, but did not pass more stringent thresholds of FDR <0.1, Inclusion level difference ≥10% and ≤95% and average junction read count ≥10. In (C, D), data shows -log$_{10}$Padj value from Fisher's one-tailed test, corrected for multiple testing with the default gProfiler g:SCS algorithm. In (E, F) data shows numbers of splicing events. In (H), data represent relative levels of canonical Sirt1 mRNA quantified by RT-qPCR (n = 2 mice per genotype, biological replicates), (I) data were densitometry quantification of western blot signals for individual mice, normalized to the control mice (n = 3 mice per genotype, from two experiments). *P* values from one sample *t*-test.

"Regulation of chromosome organization" categories were significantly enriched in 8 out of 9 cell types (Fig. 6D), the exception being BJ fibroblasts that had the lowest read depth dataset and hence the lowest sensitivity for alternative splicing detection (Fig. EV4A).

We then looked for the conservation of specific splicing changes across all cell types. Considering the differing sensitivities of the datasets for detecting splicing changes, we defined as "conserved" splicing changes, those events equally changed in at least six out of the nine hnRNPL-deficient versus WT datasets. This analysis was restricted to the exon skipping or inclusion events because there is sufficient sequence conservation to compare them (see Methods for full description). Compared to differentially expressed genes (Fig. EV3D), there were more splicing changes shared between the datasets, with 41 species-specific events (Fig. 6E; Dataset EV5). In addition, 18 differential splicing events affecting 15 genes were shared in ≥6 datasets across species (Fig. 6E). As indirect evidence that all these conserved events were hnRNPL targets, we scored the presence of hnRNPL-binding motifs (Ray et al, 2013; Smith et al, 2013) within the skipped exon or the adjacent introns. All 70 events displayed at least one CA dinucleotide-rich motif that can be recognized by hnRNPL, and 45 out of the 70 had the canonical ACACACA or ACACAAA motif (Fig. 6F).

Notably, at least 9 of the 15 genes with conserved splicing changes encoded for proteins that can affect gene expression (bolded genes in the inset of Fig. 6E), including those encoding H3K27 demethylase *KDM6A* and transcriptional coactivator *GPBP1* (Hsu et al, 2003) in eight datasets, as well as H3K36 methyltransferase *NSD2*, splicing factor and transcriptional regulator *HNRNPR* (Ji et al, 2022), histone and non-histone protein deacetylase *SIRT1* (Wang et al, 2019; Gan et al, 2020), the NuA4/TIP60 complex subunit *EPC1*, and the H3K14ac histone reader BAZ2B that is part of chromatin remodeling complexes (Oppikofer et al, 2017) (Figs. 6E,G and EV4B; Dataset EV5).

Most of the conserved splicing events were increased inclusions of an intermediate exon upon hnRNPL depletion, indicating that hnRNPL normally promotes the exclusion of these exons. The consequence of each event was gene-specific, but they were all

known or could be predicted to have functional consequences in hnRNPL-deficient cells (Figs. 6G and EV4B). For example, the conserved hnRNPL-regulated splicing event affecting SIRT1 caused more inclusion of an intermediate exon that created a premature stop codon, disrupting the deacetylase domain (Figs. 6G and EV4C). The increase in this event coincided with reduced overall *SIRT1* transcript levels in activated B and six other cell types upon hnRNPL loss (Fig. EV4D). RT-qPCR showed that resting hnRNPL-deficient B cells also expressed less of the canonical *Sirt1* transcript than WT, the non-canonical *Sirt1* transcript with the exon inclusion being only detected as a weak amplification band in the hnRNPL-deficient B cells (Fig. 6H). These results suggested that the hnRNPL-regulated inclusion event might lead to NMD. We confirmed this by *Hnrnpl* knockdown in the CH12 B cell lymphoma line (Fig. EV4E). Reduced hnRNPL expression correlated with reduced canonical and increased non-canonical *Sirt1* transcript (Fig. EV4E). Transcription inhibition and chase showed that the non-canonical transcript displayed a shorter half-life, both in WT and hnRNPL-depleted CH12 cells (Fig. EV4F), as expected for a transcript-intrinsic trait. Other conserved alternative splicing events in hnRNPL-deficient cells created protein isoforms with known biological effects. For instance, the H3K36 methyltransferase NSD2 in hnRNPL-deficient cells showed increased inclusion of an alternative exon that biased the balance towards the MMSET-I form (Fig. 6G), which lacks the active methyltransferase domain (Lam et al, 2022). The increased exon inclusion in the *KDM6A* transcript, encoding an H3K27 demethylase (Fig. 6G), introduced a nuclear localization sequence that increases KDM6A nuclear abundance and, therefore, activity (Fotouhi et al, 2023). Expression levels of these other genes were mostly unchanged in hnRNPL-deficient cells (Fig. EV4D), consistent with the generation of alternative protein forms rather than NMD.

Based on the abundance of histone modifiers among the splicing events regulated by hnRNPL, we then asked if hnRNPL loss might result in global epigenetic alterations in B cells. Contrary to what might be expected from Sirt1 downregulation, H3K9ac was decreased in resting *Hnrnpl$^{-/-}$* B cells (Fig. 6I). However, the interpretation of this result is complicated because SIRT1 can have

indirect effects on histone modifications (Wang et al, 2019) and there are other H3K9 deacetylases and acetyltransferases in B cells. On the other hand, the increase in the transcript encoding the nuclear isoform of KDM6A did correlate with a global decrease in H3K27me3 levels in hnRNPL-deficient cells (Fig. 6I), which was consistent with the expected dominant effect from a higher nuclear level of this demethylase.

We conclude that hnRNPL has conserved roles in splicing that affect multiple enzymes involved in transcriptional regulation, including histone modifiers like SIRT1, NSD2, and KDM6A. Accordingly, hnRNPL deficiency leads to global epigenetic alterations in B cells, which could explain in part the non-overlapping effect on gene expression.

## Mitochondrial defects in hnRNPL-negative B cells

hnRNPL loss has been associated with mitochondrial defects in fetal liver cells (Li et al, 2015; Gaudreau et al, 2016a). SIRT1 is implicated in mitochondrial biogenesis and function, as well as mitophagy (Wan et al, 2022; Sun et al, 2022), partly by deacetylating PGC-1α and PGC-1β, transcriptional coactivators important for mitochondrial function (Kelly et al, 2009; Yi and Luo, 2010). GSEA analysis of DEG against a collection of metabolic and mitochondrial gene sets did not yield significant changes (not depicted). However, *Ppargc1b* (the gene encoding PGC-1β), which is the family member expressed in B cells, was downregulated by 50% and *Tfam*, encoding a transcription factor required for mitochondrial biogenesis (Wan et al, 2022), was significantly reduced in hnRNPL-deficient B cells (Fig. EV5A). Furthermore, the leading edge of the upregulated "Xenobiotic metabolism" Hallmark gene set (Fig. 4I) reflected the upregulation of genes encoding enzymes that respond to redox or mitochondrial stress, such as *Pink1, Cd36, Gstt2, Gstm4, Gstk1*, etc. (Fig. EV5B). Given these observations and the critical role of mitochondrial function during B cell activation (Akkaya et al, 2018; Boothby et al, 2022; Sadras et al, 2021; Waters et al, 2018), we analysed mitochondrial function in hnRNPL-deficient B cells.

To measure B cell mitochondrial function, we stained resting and LPS/IL-4-stimulated B cells with two mitochondrial probes: mitotracker APC that is a measure of activity through mitochondrial membrane potential (MMP), and mitotracker green (MTG), which is incorporated into all mitochondria regardless of membrane potential and can thus be used to measure total volume/mass. hnRNPL-deficient resting B cells clearly exhibited higher MMP and MTG staining compared to controls (Fig. 7A). These differences were still observable 1 day after activation but equalized by day 2, possibly due to the loss of the hnRNPL-deficient cells (Fig. 7A). At day 2, we observed more depolarized (i.e., MTG$^+$ MMP$^{low}$) mitochondria in hnRNPL-deficient B cells (Fig. 7A), potentially reflecting the onset of apoptosis and/or functional defects in mitochondria. We also verified a higher gene dose ratio of mitochondrially encoded *Nd1* to nuclear- encoded *Rpl35a* showing increased mitochondrial DNA content in hnRNPL-deficient versus WT B cells (Fig. 7B), consistent with the increased mitochondrial mass (MTG signal).

Despite evidence of more mitochondria, hnRNPL-deficient resting B cells had the same basal and maximal oxygen consumption rate (OCR) compared to control cells, although they displayed increased proton leak (Fig. 7C). Glycolysis was unaffected in these cells, as measured by extracellular acidification rate (ECAR), indicative of lactate production (Fig. 7C). However, one day after activation, hnRNPL-deficient cells showed reduced coupled, maximal, and spare respiratory capacity (Fig. 7D), indicating mitochondrial dysfunction. In contrast, ECAR was increased, leading to a reduced OCR/ECAR ratio (Fig. 7D,E), suggesting that activated hnRNPL-deficient B cells were trying to compensate for an energy deficit by increasing glycolysis.

Intracellular reactive oxygen species (ROS) were substantially higher in resting and activated hnRNPL-deficient B cells, with the largest differences in resting B cells (Fig. 7F), as well as in hnRNPL-depleted CH12 B cells (Fig. EV5C). Of note, ROS production preceded the observation of increased apoptosis or p53 activation in primary B cells (see Fig. 4B), suggesting it was not a consequence of compromised survival. The transcript levels of major ROS scavengers, electron transport chain components, and mitophagy regulators were unchanged or slightly (<1.5-fold) increased in hnRNPL-deficient B cells (Fig. EV5D), ruling out defects in their expression. While increased ROS could simply reflect the mitochondrial mass increase in hnRNPL-deficient B cells, when coupled with defects in oxidative phosphorylation observed shortly after LPS /IL-4 treatment, increased ROS may also indicate an inherent mitochondrial dysfunction that becomes accentuated by the process of B cell activation.

We conclude that hnRNPL is required for optimal mitochondrial function in B cells, limiting ROS accumulation and supporting optimal mitochondrial respiration required during activation.

## Discussion

B cell activation entails a rapid increase in cellular biomass and global transcription to sustain proliferation and function (Sadras et al, 2021; Nie et al, 2012; Kieffer-Kwon et al, 2017), which likely brings about requirements for increased splicing control. We show that hnRNPL is required for several aspects of the B cell activation program. Mice lacking hnRNPL in resting B cells have reduced MZ B cells, which exist in a primed state (Lopes-Carvalho et al, 2005), and fail to produce germinal center B cells. The hnRNPL-deficient B cells stimulated ex vivo fail to increase their size, progress past the G1 cell cycle phase, or proliferate, and undergo apoptosis. Gene expression and metabolic changes strongly suggest a few concurrent causes for the B cell activation defect via roles of hnRNPL in regulating alternative splicing events and, thereby, key transcriptional programs and mitochondrial function.

Using a comparative approach, we identified conserved functions for hnRNPL, notably influencing the transcriptional programs orchestrated by MYC and E2Fs in B and other cell types and dampening lncRNA expression.

In activated B and most other cell types, hnRNPL somehow enables MYC and E2F gene signatures. MYC is a major driver of B cell activation; it is upregulated soon after activation and contributes to globally increasing transcription and metabolism to sustain cell growth and proliferation (Tesi et al, 2019; Caro-Maldonado et al, 2014; Nie et al, 2012; Heinzel et al, 2017; Kieffer-Kwon et al, 2017). Interestingly. the study of B cells defective in splicing factors hnRNPF, HuR, PTBP1, TIA1 (Monzón-Casanova et al, 2018; Osma-Garcia et al, 2021, 2023; Huang et al, 2023), and now hnRNPL, suggest that either MYC expression or its

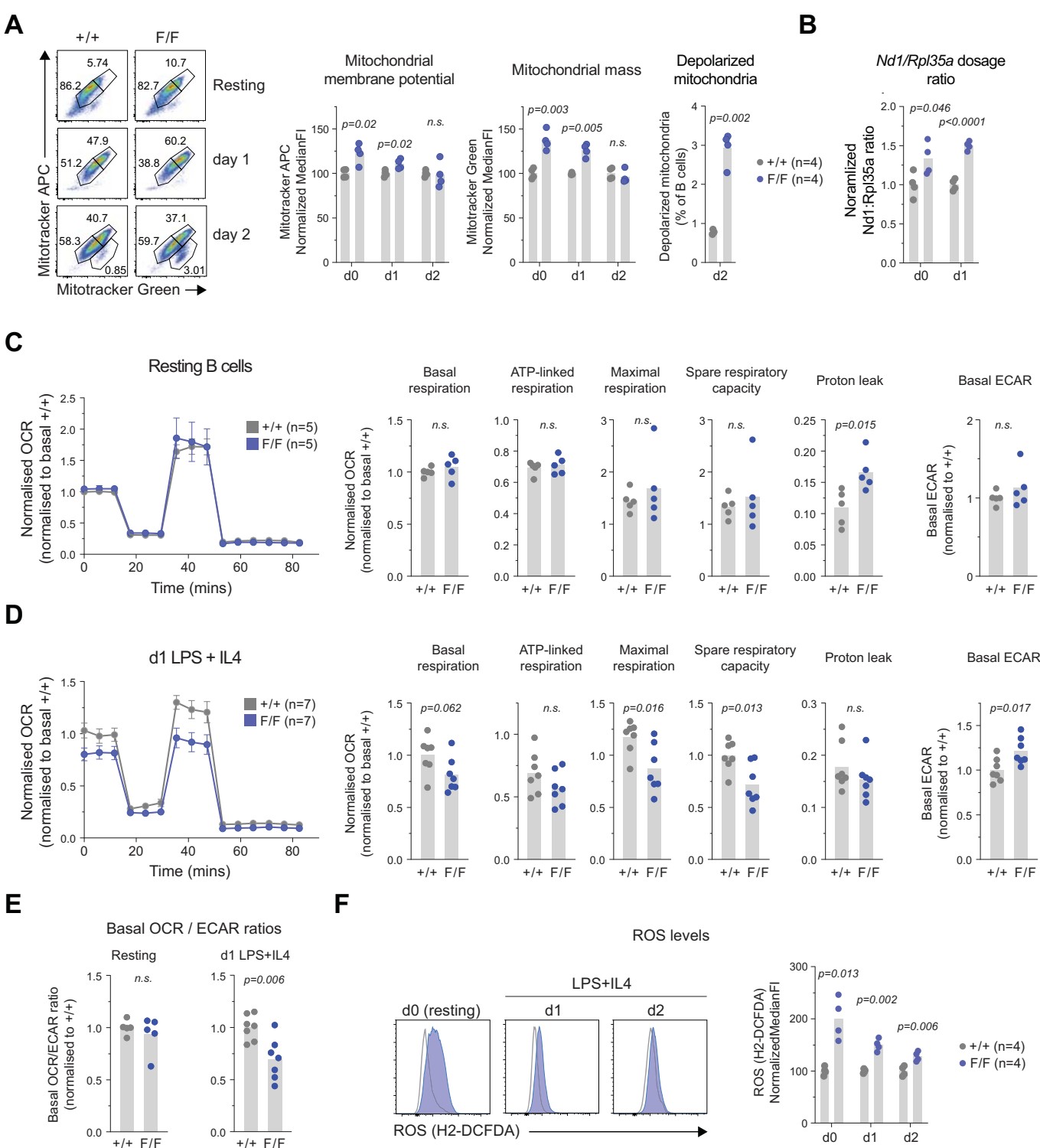

transcriptional program are especially sensitive to splicing defects in activated B cells. The activity of E2F transcription factors is also necessary for S-phase entry (Lam et al, 1998; Hsia et al, 2002; Nie et al, 2012; Hinman et al, 2009; Yusuf et al, 2004; Pae et al, 2021). Thus, suboptimal MYC and E2Fs programs can contribute to the activation defect in B cells, with additional impact of more B cell-specific effects. For instance, mTORC signaling enables anabolic

pathways essential for cell cycle progression and cell growth following B cell activation (Saxton and Sabatini, 2017; Patterson et al, 2021; Iwata et al, 2016) and is also reduced in hnRNPL-deficient B cells.

The number of genes affected in either their expression or splicing by hnRNPL deficiency imply that the cause of B cell activation and antibody response defects is likely pleiotropic. It is thus unlikely that

**Figure 7. Mitochondrial defects precede B cell activation defects in hnRNPL-deficient B cells.**

(A) Representative flow cytometry plots and quantification summarizing mitotracker APC and mitotracker green signal, as well as a proportion of B cells with depolarized mitochondria in purified resting (d0) and activated (LPS/IL-4) splenic B cells from *CD21-cre* (+/+), *CD21-cre Hnrnpl^F/+* (F/+) and *CD21-cre Hnrnpl^F/F* (F/F) mice at days 1 (d1) and 2 (d2). (n = 4 mice per group). (B) *Nd1* to *Rpl35a* gene dosage ratio calculated by RT-qPCR in cells from (A). (C) Normalized oxygen consumption rate (OCR) from Seahorse mitochondrial stress test (Mean ± SEM), and plots showing derived parameters of mitochondrial basal, ATP-linked, maximal, spare respiratory capacity, and mitochondrial proton leak, as well as ECAR measurement, in purified resting splenic B cells (n = 5 mice per group). (D) as in (C), but for B cells 1 day after activation with LPS/IL-4 (n = 7 mice per group). (E) Basal OCR to ECAR ratio for cells in (C) and (D). (F) Representative flow cytometry plots and quantification summarizing normalized median fluorescence intensity of ROS staining in resting and B cells activated with LPS/IL-4 (n = 4 mice per group). Data information: In (A–F), data were represented as individual mice values with mean shown as bars. Plots compile results from (A, B) two experiments, (C–E) three experiments, (F) two experiments. (A–F) statistical significance (p < 0.05) tested by unpaired two-tailed t-test with Welch's correction.

any single change in alternative splicing can provide the molecular link between hnRNPL and the MYC/E2F transcriptional programs. The splicing of E2Fs or MYC was not affected in any cell type, ruling out a direct effect. E2f1 and E2f3 expression was reduced in hnRNPL-deficient B cell and might contribute to the E2F signature defect, albeit other E2fs or Myc were not significantly changed. Moreover, the expression of E2Fs or MYC genes was largely unaffected in the other cell types. In addition, while their signatures are affected across cell lines, the specific differentially expressed genes did not overlap much between different cells. Finally, unlike in most other cell types, the MYC/E2F programs were not down but upregulated in HEK293 cells. Collectively, these observations suggest that the effect on MYC/E2F programs is conserved but the mechanisms behind it is context dependent. We posit that this can be at least partly explained as the indirect consequence of changes in the epigenetic landscape (see below).

Our data suggests that hnRNPL impacts transcription indirectly. It has been proposed that hnRNPL can modulate transcription via Mediator or pTEFb (Giraud et al, 2014; Huang et al, 2012). Large scale analyses in K562 and HEPG2 cell lines show association of hnRNPL with coding genes (Xiao et al, 2019) and there are anecdotal examples suggesting an effect, such as the occupancy of the *Trail2R* promoter by hnRNPL and differential expression of *Trail2R* In hnRNPL-null versus WT fetal liver cells (Gaudreau et al, 2016a). We confirmed the association of hnRNPL with genes by reanalyzing K562 ChIP-seq data and in selected B cell genes using ChIP. However, our analysis of the K562 data and the *IgH*, *Myc*, and other genes in B cells, fails to show a clear correlation between hnRNPL occupancy and gene expression changes upon its deletion. Thus, indirect mechanisms are more likely to explain the effect of hnRNPL on gene expression. hnRNPL can also affect the level of a subset of transcripts by direct binding, either by protecting from NMD, as described in human cell lines (Kishor et al, 2019), or with transcript-dependent effect on the abundance of different lncRNAs (Li et al, 2014; Klingenberg et al, 2018; Ruan et al, 2016). However, changes in lncRNA abundance could also be influenced by splicing changes in multiple factors involved in non-coding RNA processing ($Padj = 9.35 \times 10^{-9}$), including the exosome subunits Exosc1 and Exosc10 (see Dataset EV4), which limits lncRNA expression (Laffleur et al, 2021). Our data cannot confirm or exclude these possibilities. The upregulation of lncRNAs in hnRNPL-deficient cells, adds another layer of complexity, since lncRNA are well known regulators of gene expression (Li et al, 2014; Klingenberg et al, 2018; Ruan et al, 2016). Nonetheless, the best known and likely main function of hnRNPL is in alternative splicing by determining the inclusion or exclusion of exons in pre-mRNAs, which can alter transcript levels and protein function (Dominguez et al, 2018; Cole et al, 2015; McClory et al, 2018) offers a more likely possibility.

The regulation of alternative splicing of histone-modifying and other transcriptional regulators emerged in our analysis as a distinct conserved function of hnRNPL. This conservation extended even at the level of specific alternative splicing events, which, based on the presence of CA-rich motifs, could be directly recognized by hnRNPL. These conserved events affected transcripts for at least 9 factors with the capacity to change transcription or the epigenetic landscape and likely have functional consequences. The hnRNPL-regulated alternative splicing of *NSD2* pre-mRNA could change the ratio of isoforms to reduce its methyltransferase activity, which produces H3K36me2. In addition, the overall *Nsd2* expression is ~40% lower in hnRNPL-null than control B cells (Fig. EV5D). We did not find suitable antibodies to probe for H3K36me2 in B cells, but given the association of H3K36me2 with transcription regulation (Lam et al, 2022), it could impinge on gene expression. Furthermore, since Nsd2 contributes to the *IgH* Sγ1 expression, AID recruitment, and DNA repair in mice (Pei et al, 2013; Nguyen et al, 2017), reduced Nsd2 may contribute to the CSR defect in hnRNPL-deficient B cells. Interestingly, hnRNPL is a cofactor of SETD2, the methyltransferase catalyzing H3K36me3 onto H3K36me2. However, we can exclude a Setd2-dependent mechanism as a cause of hnRNPL-deficient B cell phenotypes because other splicing factors can compensate for hnRNPL loss in this context (Bhattacharya et al, 2021a) and, in contrast to *Hnrnpl* deletion, B cell-specific deletion of *Setd2* has little effect on GC formation (Leung et al, 2022). We did follow up on KDM6A, which was the most conserved event (8 of 9 datasets) and has a known functional consequence. Consistent with the enhanced nuclear localization and activity of the KDM6A isoform that is repressed by hnRNPL (Tran et al, 2020), we verified a global reduction in H3K27me3 in naïve and activated hnRNPL-deficient B cells. The loss of this repressive mark could underlie the upregulation of lncRNAs and other genes. SIRT1 is highly expressed in resting B cells and up to 2 days after activation (Gan et al, 2020), so we also pursued the conserved hnRNPL-repressed splicing event affecting SIRT1, which had not been characterized. We could demonstrate that the premature stop codon caused by this splicing event correlates with a lower half-life of the corresponding isoform transcript in a B cell line. This is consistent with NMD and can explain the reduced Sirt1 expression across hnRNPL-deficient cell types. Deacetylation by SIRT1 modulates the function of several histone and non-histone proteins (McBurney et al, 2013; Yang et al, 2022). As an example of a Sirt1 histone target we probed H3K9ac levels, but we observed a global reduction in hnRNPL-deficient resting B cells, rather than the increase that might be expected. The interpretation of this result is not straightforward because, unlike the change in KDM6A, reduced Sirt1 would not have a dominant phenotype and due to the presence of at least seven other histone deacetylases and five histone acetyltransferases that could affect this mark. Nonetheless, the global

H3K9ac decrease provides further evidence of epigenetic alteration in the absence of hnRNPL. SIRT1 can also deacetylate transcription factors, whereby it could also affect gene expression. Thus, deacetylation of p53 dampens its response (Yi and Luo, 2010), of the RB protein prevents E2F inhibition to enable cell cycle progression (Wong and Weber, 2007; Jablonska et al, 2016; Imperatore et al, 2017), and of FOXO1/3a prevents it from enforcing cell cycle arrest (Motta et al, 2004). Deacetylation of MYC can promote or dampen its activity depending on the context (Mao et al, 2011; Menssen et al, 2012; Yuan et al, 2009a). Thus, reduced levels of SIRT1 in hnRNPL-null B cells could contribute to the deregulation of MYC and E2F targets, as well as p53 activation, thus hampering cell cycle progression and survival. Collectively, these examples, and the evidence of global changes in transcription associated histone marks, provide a likely mechanism whereby hnRNPL could indirectly and widely affect gene expression by controlling alternative splicing of transcription regulators.

Activated hnRNPL-null B cells show strong indications of mitochondrial respiration defects, increased ECAR, and elevated ROS, which might be linked to mitochondrial dysfunction. ROS-induced oxidative damage could activate p53 and apoptosis. $Hnrnpl^{-/-}$ fetal liver cells also show upregulation of p53 target genes and apoptosis, albeit the latter is p53-independent (Gaudreau et al, 2016a), which we did not test in B cells. Nonetheless, the p53 activation in $Hnrnpl^{-/-}$ B cells likely contributes to the G1 cell cycle arrest by inducing p21 (Maillet and Pervaiz, 2012). Our data does not directly link the mitochondrial defects to Sirt1 or other differentially expressed ($Tfam$, $Ppargc1b$) or spliced gene in hnRNPL-null B cells. The defect could also be an indirect consequence of epigenetic changes, some upregulated lncRNA, or a combination of factors. Nonetheless, since mitochondrial

remodeling and respiration are critical for B cell activation (Waters et al, 2018), mitochondrial dysfunction likely contributes to the activation defect of hnRNPL-null B cells.

Our study has revealed an essential role of hnRNPL in B cell activation and physiology and, thereby, in the antibody response. hnRNPL is required for cell cycle progression, preventing cell death during activation, and enabling mitochondrial function. We provide evidence for conserved hnRNPL functions in murine and human cell types, regulating the activity of epigenetic modifiers by alternative splicing, and dampening lncRNA levels. Both functions can have direct and indirect effects on gene expression regulation, as exemplified by global levels of at least some epigenetic marks. This observation could explain the little overlap between genes differentially expressed and mis-spliced genes in the absence of hnRNPL. While hnRNPL has pleiotropic effects, it consistently affects the MYC and E2F transcriptional programs, which are necessary for cell cycle progression and essential for B and likely other cells. Thus, our work suggests common underlying principles of hnRNPL function.

## Methods

### Mice

The generation of the $Hnrnpl^{F/F}$ mice was previously described (Gaudreau et al, 2012a). C57BL6/J, $CD21$-$cre$ (Kraus et al, 2004) (RRID:IMSR_JAX:006368) and $Rosa^{mT/mG}$ reporter

**Reagents and tools table**

| Reagent/Resource | Reference or Source<br>**Source** (public): Stock center, company, other labs Reference: list relevant study if referring to previously published work; use "this study" if new. If neither applies: briefly explain. | Identifier or Catalog Number<br>*Provide catalog numbers, stock numbers, database IDs or accession numbers, RRIDs or other relevant identifiers.* |
|---|---|---|
| **Experimental Models** | | |
| *List cell lines, model organism strains, patient samples, isolated cell types etc. Indicate the species when appropriate.* | | |
| C57BL/6 J (M. musculus) | Jackson Lab | B6.129P2Gpr37tm1Dgen/J |
| CD21-cre (M. musculus) | Jackson Lab | RRID:IMSR_JAX:007676 |
| B6.129(Cg)- Gt(ROSA)26Sor<sup>tm4(ACTB-tdTomato,-EGFP)Luo</sup>/J (M. musculus) | Jackson Lab | RRID:IMSR_JAX:007676 |
| B6.SJL-Ptprc<sup>a</sup> Pepc<sup>b</sup>/BoyJ (M. musculus) | Jackson Lab | RRID:IMSR_JAX:002014 |
| Hnrnpl<sup>F/F</sup> (M musculus) | Gaudreau et al, 2012a | Condtional Hnrnpl allele |
| CH12-F3 cell line (M. musculus) | Nakamura et al, 1996 | Clone F3 from Dr Tasuku Honjo and maintained in our laboratory |
| HEK293T cell line (H. sapiens) | ATCC | RRID:CVCL_0063 |
| **Recombinant DNA** | | |
| *Indicate species for genes and proteins when appropriate* | | |
| EZ-tet-pLKO-blast | Addgene | Cat # 85973 |
| pMD2.G | Addgene | Cat # 12259 |
| psPAX2 | Addgene | Cat # 12260 |
| **Antibodies** | | |
| monoclonal rat anti-B220 (CD45R) Alexa Fluor 700 Flow Cytometry 1:100 | BioLegend | 103232 |

 

| Reagent/Resource | Reference or Source<br><br>**Source** (*public*): *Stock center, company, other labs* Reference: *list relevant study if referring to previously published work; use "this study" if new. If neither applies: briefly explain.* | Identifier or Catalog Number<br><br>*Provide catalog numbers, stock numbers, database IDs or accession numbers, RRIDs or other relevant identifiers.* |
|---|---|---|
| monoclonal rat anti-B220 (CD45R) APC Flow Cytometry 1:100 | BD | 553092 |
| monoclonal rat anti-B220 (CD45R) PercP-Cy5.5 Flow Cytometry 1:100 | BD | 552771 |
| monoclonal hamster anti-CD3e biotin Flow Cytometry 1:100 | BD | 553239 |
| monoclonal hamster anti-CD3e PE Flow Cytometry 1:100 | BD | 553063 |
| monoclonal rat anti-GL7 BV421 Flow Cytometry 1:100 | BD | 562967 |
| monoclonal rat anti-GL7 PerCP-efluor 710 Flow Cytometry 1:100 | eBioscience | 46-5902-80 |
| monoclonal hamster anti-Fas (CD95) BV421 Flow Cytometry 1:100 | BD | 562633 |
| monoclonal rat anti-IgD FITC Flow Cytometry 1:100 | BD | 553439 |
| monoclonal rat anti-IgM BV421 Flow Cytometry 1:100 | BD | 562595 |
| monoclonal rat anti-CD21/CD35 APC Flow Cytometry 1:100 | BD | 558658 |
| monoclonal rat anti-CD23 PE-Cy7 Flow Cytometry 1:100 | BD | 562825 |
| monoclonal mouse anti-CD45.1 APC Flow Cytometry 1:100 | BD | 558701 |
| monoclonal mouse anti-CD45.2 APC-Cy7 Flow Cytometry 1:100 | BD | 560694 |
| monoclonal mouse anti-CD45.2 BUV395 Flow Cytometry 1:100 | BD | 564616 |
| Annexin V APC Flow Cytometry 3:100 | BD | 550474 |
| monoclonal rat anti-IgG1 biotin Flow Cytometry 1:100 | BD | 553441 |
| monoclonal mouse anti-Biotin APC Flow Cytometry 1:100 | Miltenyi | 130-090-856 |
| Streptavidin BV605 Flow Cytometry 1:50 | BioLegend | 405229 |
| Viability dye Efluor 780 Flow Cytometry 1:100 | eBioscience | 65-0865-14 |
| monoclonal rabbit anti-H3K27me3 WB 1:2000 | Cell Signaling | 9733 |
| monoclonal mouse anti-H3 WB 1:5000 | Cell Signaling | 3638 |
| monoclonal mouse anti-p53 WB 1:1000 | Cell Signaling | 2524 |
| polyclonal abbit anti-Actin WB 1:2000 | Sigma | A2066 |
| polyclonal goat anti-mouse IgG IRDye 800 WB 1:10000 | LI-COR | 926-32210 |
| polyclonal goat anti-rabbit IgG IRDye 800 WB 1:10000 | LI-COR | 925-32211 |
| polyclonal donkey anti-rabbit IgG Alexa Fluor 680 WB 1:10000 | Invitrogen | A10043 |
| polyclonal rabbit anti-hnRNPL WB 1:2000 | Aviva | ARP40368_P050. Lot: QC9464-42964 |
| polyclonal rabbit anti-hnRNPL ChIP 5ug / ChIP | Aviva | ARP40368_P050. Lot: QC9464-42964 |
| polyclonal mouse anti-mouse IgG ChIP 5ug / ChIP | Santa cruz | sc-2025 |
| polyclonal rat anti-mouse IgG1 ELISA (coating) 2 µg/ml | BD | 553445 |
| polyclonal rat anti-mouse IgG1 biotin ELISA (detection) 1:3000 | BD | 553441 |
| monoclonal mouse anti-mouse IgG1κ ELISA (standard curve) | BD | 557273 |
| Streptavidin HRP ELISA 1:5000 | Thermo Scientific | N100 |
| **Oligonucleotides and sequence-based reagents** | | |
| RT-qPCR primers | | |
| p21 (Cdkn1a) Forward | This study | AGATCCACAGCGATATCCAGAC |
| p21 (Cdkn1a) Reverse | This study | ACCGAAGAGACAACGGCACACT |
| Puma (Bbc3) Forward | This study | ACGACCTCAACGCGCAGTACG |

| Reagent/Resource | Reference or Source<br><br>**Source** (*public*): *Stock center, company, other labs* Reference: *list relevant study if referring to previously published work; use "this study" if new.* If neither applies: *briefly explain.* | Identifier or Catalog Number<br><br>*Provide catalog numbers, stock numbers, database IDs*<br>*or accession numbers, RRIDs or other relevant identifiers.* |
|---|---|---|
| Puma (Bbc3) Reverse | This study | GAGGAGTCCCATG AAGAGATTG |
| Noxa (Pmaip1) Forward | This study | TCGCAAAAGAGCAGGATGAG |
| Noxa (Pmaip1) Reverse | This study | CACTTTGTCTCCAATCCTCCG |
| Bax Forward | This study | CAGGATGCGTCCACCAAGAA |
| Bax Reverse | This study | AGTCCGTGTCCACGTCAGCA |
| Bad Forward | This study | CAGCAGCCCAGAGTATGTTCC |
| Bad Reverse | This study | CGTCCCTGCTGATGAATGTTG |
| Bcl2 Forward | This study | GAGCGTCAACAGGGAGATGT |
| Bcl2 Reverse | This study | CTGGGGCCATATAGTTCCACAA |
| Mcl1 Forward | This study | CCCCTCCCCCATCCTAATCA |
| Mcl1 Reverse | This study | CAATCCCTGGTCACTGTCGG |
| BclXL (Bcl2l1) Forward | This study | AGTAAACTGGGGGTCGCATCG |
| BclXL (Bcl2l1) Reverse | This study | GCCATCCAACTTGCAATCCG |
| Actin Forward | This study | CTCTGGCTCCTAGCACCATGAAGA |
| Actin Reverse | This study | GTAAAACGCAGCTCAGTAACAGTCCG |
| Rps15 Forward | This study | GAAGTGGAGCAGAAGAAGA |
| Rps15 Reverse | This study | CTGCATCAGTTGCTCATAG |
| Nd1 Forward | This study | GTC CAT ACG GCA TCC TAC AAC CAT |
| Nd1 Reverse | This study | TGT GAG TGA TAG GGT GGG TGC AAT |
| Rpl35a Forward | This study | CGTGCCAAATTCCGAAGCAA |
| Rpl35a Reverse | This study | ATGGGTACAGCATCACACGG |
| Mfn1 Forward | This study | CCT ACT GCT CCT TCT AAC CCA |
| Mfn1 Reverse | This study | AGG GAC GCC AAT CCT GTG A |
| Drp1 Forward | This study | CAGGAATTGTTACGGTTCCCTAA |
| Drp1 Reverse | This study | CCTGAATTAACTTGTCCCGTGA |
| Tfam Forward | This study | GGTATGGAGAAGGAGGCCCGG |
| Tfam Reverse | This study | CGAATCATCCTTTGCCTCCTGGAAGC |
| Catalase Forward | This study | GCG GAT TCC TGA GAG AGT GGT AC |
| Catalase Reverse | This study | GCC TGA CTC TCC AGC GAC TGT GGA G |
| Sod1 Forward | This study | CGG ATG AAG AGA GGC ATG TT |
| Sod1 Reverse | This study | TGC TCT CCT GAG AGT GAG AT |
| Sod2 Forward | This study | GGC CAA GGG AGA TGT TAC AAC |
| Sod2 Reverse | This study | GCA ACT CTC CTT TGG GTT CTC |
| Gpx1 Forward | This study | CAG GAG AAT GGC AAG AAT GAA GAG |
| Gpx1 Reverse | This study | GGC ATT CCG CAG GAA GGT AAA GAG CGG |
| Ndufs1 Forward | This study | AGG ATA TGT TCG CAC AAC TGG |
| Ndufs1 Reverse | This study | TCA TGG TAA CAG AAT CGA GGG A |
| Ndufs5 Forward | This study | GAC ATA CAG AAA AAG CTG GGC A |
| Ndufs5 Reverse | This study | TCG CCT CAT CGT TTT GTA CCG |
| Uqcrc1 Forward | This study | TGC CTT AGA GAA GGA GGT AGA G |
| Uqcrc1 Reverse | This study | GAC AGT GCC TTG ATG AGG TAA G |
| CytC Forward | This study | GCAAGCATAAGACTGGACCAAA |
| CytC Reverse | This study | TGTTGGCATCTGTGTAAGAGAATC |
| Cox5b Forward | This study | ATG CTA CCT CCA AAG GCA GCT TC |
| Cox5b Reverse | This study | TGC AGC CCA CTA TTC TCT TGT TGC |

| Reagent/Resource | Reference or Source | Identifier or Catalog Number |
| --- | --- | --- |
| | **Source** (*public*): *Stock center, company, other labs* Reference: *list relevant study if referring to previously published work; use "this study" if new.* If neither applies: *briefly explain.* | *Provide catalog numbers, stock numbers, database IDs* *or accession numbers, RRIDs or other relevant identifiers.* |
| Cox6c Forward | This study | GAG TTG CCG CTG CCT ATA A |
| Cox6c Reverse | This study | CTG AAA GAT ACC AGC CTT CCT C |
| Atp5b Forward | This study | TAT GTG CCT GCT GAT GAC CTG ACT |
| Atp5b Reverse | This study | ATC CAC AGC TGG ATA GAT GCC AAA |
| Atp5e Forward | This study | GTG AAA GTC TCG AAG AAG GAG TAG |
| Atp5e Reverse | This study | CCA GGA GGT GAG GTT GAT TT |
| Sirt1 - canonical isoform Forward | This study | TATCTATGCTCGCCTTGCGG |
| Sirt1 - canonical isoform Reverse | This study | GTCCGGGATATATTTCCTTTGCAAAC |
| Sirt1 - isoform with "exon 4.5" Forward | This study | GTGCAGTGGAAGGAAAGCAATTTT |
| Sirt1 - isoform with "exon 4.5" Reverse | This study | TGACACAGAGACGGCTGGAA |
| Sirt1 end point PCR Forward | This study | TATCTATGCTCGCCTTGCGG |
| Sirt1 end point PCR Reverse | This study | TGACACAGAGACGGCTGGAA |
| ChIP-qPCR primers | | |
| JH4 Forward | Rouaud et al, 2013 J Ex Med | AGGGACTTTGGAGGCTCATT |
| JH4 Reverse | Rouaud et al, 2013 J Ex Med | CTCCAACTACAGCCCCAACT |
| Pa (Su - TSS) Forward | Methot et al, 2018 Nat. Commun. | CCACCTGGGTAATTTGCATTTC |
| Pa (Su - TSS) Reverse | Methot et al, 2018 Nat. Commun. | GGGAAACTAGAACTACTCAAGCTAA |
| Pb (Su - TSS) Forward | Methot et al, 2018 Nat. Commun. | AGCTTGAGTAGTTCTAGTTTCCC |
| Pb (Su - TSS) Reverse | Methot et al, 2018 Nat. Commun. | GAGACCAATAATCAGAGGGAAGAA |
| Sµ-a (Upstream of switch repeats) Forward | Cortizas et al, 2013 J. Immunol. | tagtaagcgaggctctaaaaagcac |
| Sµ-a (Upstream of switch repeats) Reverse | Cortizas et al, 2013 J. Immunol. | ACTCAGAGAAGCCCACCCAT |
| Sµ-b (Downstream of switch repeats) Forward | Cortizas et al, 2013 J. Immunol. | GGTTGGGAGACCATGAATTG |
| Sµ-b (Downstream of switch repeats) Reverse | Cortizas et al, 2013 J. Immunol. | TTCTTAGCTCAACCCAGTTTATCC |
| P-Sy1 (TSS) Forward | Methot et al, 2018 Nat. Commun. | GCTGCAAGAAGAGGCCATAC |
| P-Sy1 (TSS) Reverse | Methot et al, 2018 Nat. Commun. | CTCCTTCCCAATCTCCCGTG |
| Sy1-a (Upstream of switch repeats) Forward | Methot et al, 2018 Nat. Commun. | GAGGAGTGCAGGAAGTCTGG |
| Sy1-a (Upstream of switch repeats) Reverse | Methot et al, 2018 Nat. Commun. | CCTTGATGCCCTCCCTTT |
| Sy1-b (Downstream of switch repeats) Forward | Methot et al, 2018 Nat. Commun. | GGATGTCTAGGCTGGAGCTG |
| Sy1-b (Downstream of switch repeats) Reverse | Methot et al, 2018 Nat. Commun. | GAAGCTCAGGCCTGTTGCTG |
| Myc - intron 1 (AID hotspot) Forward | This study | GGACAGGGATGTGACCGATT |
| Myc - intron 1 (AID hotspot) Reverse | This study | GATACCCGCGGATCCCAAG |
| Pax5 - introm 1 (AID hotspot) Forward | This study | GATGGACGCCTGTGAGTCAA |
| Pax5 - introm 1 (AID hotspot) Reverse | This study | ATCAAAGAGCCCATCGACCG |
| Pim1 - intron 3 (AID hotspot) Forward | This study | CGCTGGGAGGATGAAAACCT |
| Pim1 - intron 3 (AID hotspot) Reverse | This study | CCCTCGCGCTTTGATCTACT |
| Apex pr (promoter) Forward | This study | GCTGTAACCGGCACTACCA |
| Apex pr (promoter) Reverse | This study | GGTTCTGGAGCGCGATGAT |
| p21_c Forward | This study | CCCTACGTCGCGTTTCAGA |
| p21_c Reverse | This study | CGAGAGCAAAGAAAGTGCTCTTA |
| p21_e Forward | This study | GAGCCTGAAGACTGTGATGGG |
| p21_e Reverse | This study | GCCATGAGCGCATCGCAA |
| Intergenic (chr2) Forward | This study | TGGGCATATCCCTGGAGCTT |
| Intergenic (chr2) Reverse | This study | GGCCATCCCACAGTCACAAC |
| shRNA oligonucleotides | | |
| Control Forward | SHC002 MISSION shRNA control plasmids | CTAGCCAACAAGATGAAGAGCACCAATA CTAGTTTGGTGCTCTTCATCTTGTTGTTTTTG |

| Reagent/Resource | Reference or Source<br><br>**Source** (*public*): Stock center, company, other labs Reference: *list relevant study if referring to previously published work; use "this study" if new.* If neither applies: *briefly explain.* | Identifier or Catalog Number<br><br>*Provide catalog numbers, stock numbers, database IDs or accession numbers, RRIDs or other relevant identifiers.* |
|---|---|---|
| Control Reverse | SHC002 MISSION shRNA control plasmids | AATTCAAAAACAACAAGATGAAGAGCACC AAACTAGTATTGGTGCTCTTCATCTTGTTGG |
| hnRNPL Forward | RNAi consortium database TRCN0000112039 | CTAGCCCTGAACCATTACCAGATGAATACTAG TTTCATCTGGTAATGGTTCAGGTTTTTG |
| hnRNPL Reverse | RNAi consortium database TRCN0000112039 | AATTCAAAAACCTGAACCATTACCAGATG AAACTAGTATTCATCTGGTAATGGTTCAGGG |
| **Chemicals, enzymes and other reagents** | | |
| *e.g. drugs, peptides, recombinant proteins, dyes etc.* | | |
| NP$_{18}$-OVA | Biosearch technologies | Cat # N-5051-10 |
| Imject Alum adjuvant | Thermo Fisher Scientific | Cat # 77161 |
| NP$_{26}$-BSA | Biosearch technologies | Cat # N-5050H-10 |
| Trans-IT LT-1 | Mirus Bio | Cat # MIR 2305 |
| Hexadimethrine bromide | Sigma | Cat # H9268 |
| Blasticidin | Thermo Fisher Scientific | Cat # R21001 |
| RPMI 1640 | Wisent | Cat # 350-000-CL |
| DMEM | Wisent | Cat # 319-005-CL |
| 2-mercaptoethanol | Bioshop | Cat # MER002 |
| HEPES | Wisent | Cat # 330-050-EL |
| Fetal bovine serum | Wisent | Cat # 080150 |
| Penicillin/streptomycin | Wisent | Cat # 450-201-EL |
| CellTrace™ Violet | Invitrogen | Cat # C34557 |
| First Strand cDNA Synthesis with ProtoScript™ M-MuLV Taq | New England BioLabs | Cat # E6300 |
| Random Primer Mix | New England BioLabs | Cat # S1330S |
| Disuccinimidyl glutarate | Thermo Fisher Scientific | Cat # H58208-ME |
| Dynabeads Protein G | Life Technologies | Cat # 10003D |
| Mitotracker Green FM dye | Thermo Fisher Scientific | Cat # M46750 |
| Mitotracker Deep Red | Thermo Fisher Scientific | Cat # M46753 |
| CM-H2DCFDA | Thermo Fisher Scientific | Cat # C6827 |
| DRB | Cayman chemicals | Cat 10010302-50 |
| Seahorse XF Cell Mito Stress Test Kit | Agilent | Cat # 103010-100 |
| RNeasy Mini kit | Qiagen | Cat # 74104 |
| NEBNext Poly(A) mRNA Magnetic Isolation Module | New England BioLabs | Cat # E7490 |
| KAPA RNA HyperPrep kit | Roche | Cat # 08098107702 |
| **Software** | | |
| *Include version where applicable* | | |
| FlowJo v10 | LLC (flowjo.com) | flow cytometry analysis |
| Prism v10.0 | Graphpad.com | statistical analysis |
| ImageStudiolite v5.5 | Li-cor (https://www.licor.com/bio/image-studio-lite/) | western blot quantifications |
| R version 4.2.2 | https://cran.r-project.org/ | |
| RStudio 2022.07.2 Build 576 | https://posit.co/ | |
| fastp version 0.23.1 | Chen et al, 2018 | |
| STAR version 2.7.9a | Dobin et al, 2013 | |
| Rsubread version 2.12.2 | Liao et al, 2014 | |
| DESeq2 version 1.38.2 | Anders & Huber, 2010 | |
| samtools version 1.16.1 | Danecek et al, 2021 | |

| Reagent/Resource | Reference or Source<br><br>**Source** (*public*): *Stock center, company, other labs* Reference: *list relevant study if referring to previously published work; use "this study" if new. If neither applies: briefly explain.* | Identifier or Catalog Number<br><br>*Provide catalog numbers, stock numbers, database IDs*<br>*or accession numbers, RRIDs or other relevant identifiers.* |
|---|---|---|
| deepTools version 3.5.0 | https://deeptools.readthedocs.io/en/ | |
| sra-toolkit version 3.0.0 | https://github.com/ncbi/sra-tools | |
| rMATS version 4.2.2 | Shen et al, 2014 | |
| BSgenome R package version 1.66.3 | https://bioconductor.org/packages/Bsgenome | |
| Biostrings R package version 2.66.0 | https://bioconductor.org/packages/Biostrings | |
| GenomicRanges R package version 1.50.2 | Lawrence et al, 2013 | |
| UpSetR R package version 1.4.0 | Conway et al, 2017 | |
| qsmooth R package version 1.14.0 | Hicks et al, 2018 | |
| ggplot2 R package | Wickham, 2016 | |
| rmats2sashimiplot | https://github.com/Xinglab/rmats2sashimiplot | |
| FIMO version 5.5.5 | Grant et al, 2011 | |
| stringr R package version 1.5.0 | https://cran.r-project.org/web/packages/stringr/index.html | |
| WebGestalt | Wang et al, 2017 | |
| HOMER | Heinz et al, 2010 | |
| gprofiler R package version 0.2.2 or Web app | Reimand et al, 2016 | https://biit.cs.ut.ee/gprofiler/gost |
| shinyGO version 0.77 | Ge et al, 2020 | |
| msigdbr R package version 7.5.1 | Liberzon et al, 2015 | |
| fgsea R package version 1.24.0 | Korotkevich et al, 2021 | |
| ComplexHeatmap R package version 2.14.0 | Gu et al, 2016 | |
| **Other** | | |
| *Kits, instrumentation, laboratory equipment, lab ware etc. that are critical for the experimental procedure and do not fit in any of the above categories can be listed here.* | | |
| ViiA™ 7 quantitative PCR apparatus | Life technologies | |
| Covaris E220 sonicator | Covaris | |
| Seahorse XFe24 Analyzer | Agilent | |
| Odyssey CLx imaging system | Li-COR | |
| Nanodrop | Thermo Scientific | |
| Fortessa | BD | |
| FACSCalibur | BD | |
| NovaSeq 6000 | Illumina | |
| autoMACS | Miltenyi | |

(B6.129(Cg)- Gt(ROSA)26Sor[tm4(ACTB-tdTomato,-EGFP)Luo]/J) (Muzumdar et al, 2007) (RRID:IMSR_JAX:007676) and C57BL6/J CD45.1 congenics (B6.SJL-*Ptprc[a]* *Pepc[b]*/BoyJ) (RRID:IMSR_JAX:002014) mice were obtained from Jackson labs (Bar Harbour, MN). All mice were in the C57BL6/J background. Mice were housed and bred, as required to obtain the desired genotypes, at the IRCM specific pathogens-free facility in a 12 h dark/light cycles and food and drink at libitum. Male and female mice older than 6 weeks were used for experiments based on availability, without any major sex-based differences being observed, except for bone marrow chimera experiments in which only female mice were used. All mouse work was reviewed and approved by the animal protection committee at the IRCM (protocols # 2021-1087 and 2019-05), according to the guidelines of the Canadian Council of Animal Care.

## Bone marrow chimera

Mouse BM cells were isolated by flushing the femur and tibia bones with PBS using a syringe with 23G needle. BM from *CD21-cre Hnrnpl[F/F]* or *CD21-cre Hnrnpl[F/+]* mice (CD45.2[+]), were mixed with identical numbers of BM cells from CD45.1[+] WT mice. Lethally irradiated (9.5 Gy) C57BL6/J (CD45.2[+]) recipient mice were injected by the tail vein with 100 μl of PBS containing $5 \times 10^6$ mixed BM cells. All donor and recipient mice were females. Hematological reconstitution was confirmed in blood by flow cytometry 4 weeks post-injection. Mice were immunized (see below) 3 months after reconstitution. Serum was collected 11 days later, and the spleen was analyzed the next day. For mixed BM chimera experiments with *Rosa[mT/mG]* reporter, the proportion of CD45.1[+] and CD45.2[+], as well as GFP[+] and tdTomato[+] cells, were

determined for each B cell subset by flow cytometry. The CD45.2+/CD45.1+ ratios of all subsets were normalized to the ratio of the BM mix used to reconstitute the corresponding group of recipient mice. The proportional contribution of each genotype to each subpopulation was then calculated and plotted as a percentage.

## Immunization

Two to four months old mice were immunized intraperitoneally with 100 μg of NP$_{18}$-OVA (Biosearch technologies) in Imject Alum adjuvant (Thermo Scientific). Mice were age- and sex-matched whenever possible depending on their availability.

## ELISA

Antigen-specific antibodies were captured from immunized mice sera in EIA High Binding surface chemistry 96-well plates (Costar) coated with NP$_{26}$-BSA (Biosearch technologies) and incubated with dilutions of sera. Captured IgG1 was detected with Biotin-conjugated rat anti-mouse IgG1 (BD Pharmigen) followed by HRP-conjugated streptavidin (1:5000; Thermo Scientific) and developed using 2,2-azino-bis(3-ethylbenzothiazoline-6-sulfonic acid) substrate (Sigma). All antibodies are listed in Reagents and tools table.

## Western blotting

Cells were extracted in RIPA lysis buffer (1% NP-40, 10% glycerol, 20 mM Tris pH 8.0, 137 mM NaCl, 10% glycerol, 2 mM EDTA), containing protease and phosphatase inhibitors (Thermo Scientific) followed by sonication in a water bath for 10 min. Extracts separated by SDS-PAGE were transferred to nitrocellulose membranes (BIO-RAD). Membranes were blocked in TBS + 5% milk and probed with primary antibodies (1 h to overnight), washed 4 × 5 min in TBS + 0.1% Tween-20, and incubated with secondary antibodies conjugated to AlexaFluor680 or IRDye 800 for 1 h. A signal was measured using the Odyssey CLx imaging system (LI-COR) and quantified using ImageStudiolite software. Antibodies used for WB are listed in Reagents and tools table.

## Flow cytometry

Lymphocytes were obtained by mashing the mouse spleen through a 70 μm cell strainer with a syringe plunger. Cells suspensions were washed in PBS and resuspended in 1 mL of red blood cell lysis (155 mM NH$_4$Cl, 10 mM KHCO$_3$, 0.1 mM EDTA) for 5 min at room temperature. After washing, cells were resuspended in PBS + 1% BSA and stained with the different combinations of antibodies indicated in the results. Antibodies are listed in Reagents and tools table. Live cells were distinguished using DAPI or propidium iodide (PI) staining, as appropriate. Gating strategies for NF, FO, MZ and GC B cells are shown in Fig. EV1D.

## Primary mouse B cell cultures

Naïve primary B cells from mice were purified from splenocytes by CD43$^+$ cell depletion using anti-CD43 microbeads (Miltenyi, cat# 130-049-801) and an autoMACS (Miltenyi), or by using the EasySep™ Mouse B Cell Isolation Kit (Stem Cell, Cat. #19854) and

the column-free magnet EasyEights™ (Stem Cell, Cat. #18103), following manufacturer instructions. Resting B cells were cultured at 37 °C with 5% (vol vol$^{-1}$) CO$_2$ in RPMI 1640 media (Wisent), supplemented with 10% fetal bovine serum (Wisent), 1% penicillin/streptomycin (Wisent), 0.1 mM 2-mercaptoethanol (Bioshop), 10 mM HEPES, 1 mM sodium pyruvate and plated at 0.5 million cells/ml and stimulated with lipopolysaccharide (LPS) (5 μg/mL, Sigma) + IL-4 (5 ng/mL, PeproTech). Alternatively, naïve B cells were seeded onto 40LB feeder cells (Nojima et al, 2011) (a kind gift from Dr Daisuke Kitamura) to generate iGB cells. One day before B cell plating, 40LB cells were irradiated (120 Gy) and plated at 0.3 × 10$^6$ cells per well in 2 mL (six-well plate) or 0.13 × 10$^6$ cells per well (24-well plate) in 0.5 mL DMEM media supplemented with 10% fetal bovine serum (Wisent) and 1% penicillin/streptomycin (Wisent). Purified naïve B cells were plated on 40LB feeders at 10$^5$ cells per well in 4 mL of iGB media (six-well plate) or 2 × 10$^4$ cells per well in 1 mL (24-well plate), supplemented with 1 ng/mL IL-4. At day 3 post-plating, the same volume of fresh media was added to the wells, supplemented with 1 ng/mL IL-4 (PeproTech). On subsequent days, half of the volume per well was removed and replaced with media as above.

## *Hnrnpl* knockdown in CH12-F3 cells

CH12-F3 B cell lymphoma cells (a gift from Dr. Tasuku Honjo, Kyoto University) were cultured in RPMI 1640 medium (Wisent) supplemented with 10% fetal bovine serum (Wisent), 1% penicillin/streptomycin (Wisent), 0.1 mM 2-mercaptoethanol (Bioshop), and 10 mM HEPES. HEK293T cells were cultured in DMEM medium (Wisent) supplemented with 10% fetal bovine serum (Wisent) and 1% penicillin/streptomycin (Wisent). hnRNPL-targeting shRNA sequence (CCTGAACCATTACCAGATGAA) was obtained from the RNAi consortium database (Yang et al, 2011) (clone ID: TRCN0000112039). As a control, we used the sequence from MISSION® pLKO.1 non-mammalian shRNA control (CAACAA-GATGAAGAGCACCAA; Sigma Aldrich, cat# SHC002). Two complementary oligonucleotides encoding shRNA were designed (Frank et al, 2017), annealed, and ligated into the lentiviral EZ-tet-pLKO-blast (Addgene plasmid #85973) using NheI and EcoRI restriction enzymes (New England Biolabs). Oligonucleotide sequences are in Reagents and tools table. HEK293T cells (RRID:CVCL_0063) were co-transfected with EZ-tet-pLKO-blast, pMD2.G, and psPAX2 vectors (1:0.25:0.75 molar ratios, 2 μg total DNA) using Trans-IT LT-1 (Mirus Bio, Cat# MIR 2305). Supernatant with the lentivirus was harvested 48 h later. For lentiviral transduction, 0.5 million CH12-F3 cells in 0.5 ml were mixed with 1.5 ml viral supernatant in the presence of 8 μg/ml Hexadimethrine bromide (Sigma, Cat# H9268) in a 24-well plate and spun at 600 × g for 90 min at 30 °C. Media was replaced 4 h later and 2 days following transduction, cells were selected with 10 μg/ml Blasticidin (Thermo Fisher Scientific, cat# R21001) for 7 days. For inducing hnRNPL knockdown, CH12-F3 cells were plated at 50,000 cells/ml with 250 ng/ml doxycycline (Bioshop) and cultured for 3 days.

## Monitoring apoptosis, cell cycle, proliferation, and isotype switching

To assess apoptosis ex vivo, 3 × 10$^5$ cells were stained with 3 μL Annexin V-APC (cat #550474, BD Pharmigen) in 100 μL of the

provided Binding buffer (1x) for 15 min at RT. Then 400 μL of Binding buffer (1x) and 5 μL of propidium iodide (20 μg/mL) were added prior to flow cytometry acquisition. For cell cycle analysis, live cells were stained with 10 μg/ml Hoechst 33342 (Thermo) in iGB media for 1 h at 37 °C. The cell cycle stages were then calculated using FlowJo software. To assess cell division, naïve B cells were stained with 2.5 μM CellTrace™ Violet (Invitrogen, cat. no. C34557) in PBS for 20 min at 37 °C before quenching with media as per the manufacturer's protocol, just prior to their plating with cytokines or onto 40LB cells. CTV-loaded B cells were stimulated with LPS/IL-4 with conditions mentioned in the previous section to induce IgG1 switching and analyzed after 4 days by flow cytometry.

## Reverse transcription and quantitative PCR

RNA was isolated using TRIzol (Life Technologies) or TRI-reagent (Molecular Research Center, Inc), following the manufacturer's instruction, and quantified by NanoDrop (Thermo Fisher). cDNA was synthesized from 1 μg of RNA using the ProtoScript™ M-MuLV Taq RT-PCR kit and random primers (New England BioLabs). Quantitative PCR using SYBR select master mix (Applied Biosystems) was performed and analyzed in a ViiA™ 7 machine and software (Life technologies). All oligonucleotides are listed in Reagents and tools table.

To evaluate half-lives of Sirt1 isoforms, CH12 cells were treated with 100 μM DRB (Cayman Chemical, cat# 10010302-50) to inhibit transcription by RNA Pol II. The levels of each isoform at indicated times following DRB addition were assessed by RT-qPCR using isoform-specific primers. For each sample, isoform levels were normalized to time 0. In GraphPad Prism software, the decay rate of each isoform for each sample was fit by non-linear regression to "[Inhibitor] vs. response -- variable slope" equation using the robust regression method, with the constraint of Top=1. The IC50 value was taken as the half-life. This model and method were chosen because it offered the best goodness of fit measures.

## Chromatin immunoprecipitation

The detection of hnRNPL by ChIP was performed as described previously (Rashkovan et al, 2014; Helness et al, 2021) with the addition of Disuccinimidyl glutarate (DSG) crosslinker (Thermo). Briefly, $20 \times 10^6$ resting or activated B cells/ChIP freshly prepared from spleens were treated with 1.5 mM DSG for 17 min at RT. Cells were then cross-linked with 0.25% formaldehyde for resting B cells or 1% formaldehyde for activated B cells for 8 min and quenched with 125 mM glycine. Cells were lysed and chromatin was sonicated (Covaris E220 sonicator) to a size range of 200–500 bp. Samples were immunoprecipitated with 5 μg anti-hnRNPL antibody (Aviva ARP40368_P050. Lot: QC9464-42964) or control anti-IgG antibody (sc-2025, Santa Cruz Biotechnology) coupled with Dynabeads Protein G (Life Technologies). ChIP DNA was analyzed by qPCR. The relative enrichment over input was calculated using the ΔΔCt method and normalized to a negative control intergenic region per sample.

## Mitochondrial analysis

For Mitotracker stainings, 500,000 cells were stained with Mitotracker Green and Mitotracker Deep Red (Thermo) at a final concentration of 20 nM each for 30 min at 37 °C in 100 μl RPMI without phenol red (Thermo) + 1% FBS. For measuring ROS levels, 250,000 cells were stained with 1 μM CM-H2DCFDA (Thermo) in 250 μl RPMI without phenol red (Thermo) + 1% FBS for 25 min at 37 °C.

## Oxygen rate consumption analysis

0.5 million B cells were cultured in assay media (XF base medium (Agilent), 200 mg glucose, 0,5 mM sodium pyruvate, and 2 mM glutamine, pH 7.4) for 1 h without $CO_2$ prior to measurement of $O_2$ consumption and extracellular acidification rate (ECAR) by XF$^e$24 (Seahorse Bioscience) with sequential addition of 1 μM of oligomycin, 1 μM of FCCP, and 0,5 μM of rotenone.

## RNA-sequencing

GFP+ cells from *Rosa$^{mT/mG}$ CD21-cre Hnrnpl$^{F/F}$* (hnRNPL-deficient) and *Rosa$^{mT/mG}$ CD21-cre* (control) mice were FACS sorted from splenic B cell cultures, 1 day after activation with LPS/IL-4. RNA was extracted using a Qiagen RNeasy Mini kit. mRNA was enriched using NEBNext Poly(A) mRNA Magnetic Isolation Module (NEB), and the library was prepared using KAPA RNA HyperPrep kit (Roche Diagnostics), as per manufacturer's instructions. Equimolar libraries were sequenced on Illumina NovaSeq 6000 at the McGill University and Génome Québec Innovation Centre to generate 100 bp paired-end reads.

## RNA-seq analysis

Sequencing adapters were trimmed with fastp (0.23.1) (Chen et al, 2018) before alignment with STAR (2.7.9a) (Dobin et al, 2013) to mm10 reference genome (for mouse samples) or hg38 (for human samples). Gene expression counts were determined with feature-Counts (Rsubread 2.12.2) (Liao et al, 2014) in a strand-specific fashion using primary assembly annotations from GENCODE (vM23 for mouse and v43 for human). Differentially expressed genes were determined with DESeq2 (version 1.38.2) (Anders and Huber, 2010). Genes with |fold-change|≥1.5-fold and adjusted *p* value <0.1 were defined as differentially expressed. Mapped reads were filtered for mapping quality (MAPQ ≥1) with samtools (1.16.1) (Danecek et al, 2021) to generate bigwig files using bamCoverage and bamCompare from deepTools 3.5.0 with a bin size of 1 bp. Data were deposited at GEO under accession number GSE242069.

The following publicly available RNA-seq datasets were reanalyzed: primary human keratinocytes (GSE162546) (Li et al, 2021a, 2021b), HEK293T (GSE151296) (Bhattacharya et al, 2021a), HepG2 (GSE87985 and GSE88069) (ENCODE Project Consortium, 2012a, 2012b), K562 (GSE87973 and GSE88364) (ENCODE Project Consortium, 2012a, 2012b), LNCaP (GSE72844) (Fei et al, 2017a, 2017b), BJ fibroblasts (GSE154148) (McCarthy et al, 2021a, 2021b), mouse thymocytes (GSE33306) (Gaudreau et al, 2012a, 2012b) and mouse fetal liver cells (GSE57875) (Gaudreau et al, 2016a, 2016b). Raw FASTQ files of these datasets were downloaded with fasterq-dump tool from sra-toolkit (version 3.0.0) (https://github.com/ncbi/sra-tools) and RNA-seq and splicing analyses were performed using the bioinformatics pipeline described above and below, respectively.

## Splicing analysis

For optimal detection of splice variants, a multi-sample two-pass mapping strategy was employed during read mapping, following the instructions from the STAR manual (Dobin et al, 2013). Differential splicing analysis was performed from BAM files with rMATS (version 4.2.2) (Shen et al, 2014) with "--variable-read-length --novelSS" and other default options. To set a stringent RNA-seq coverage threshold, events were required to be supported by a minimum of 10 average junction reads (obtained from the ".JC" file). Then, junction counts plus the included exon counts were used for calculating the inclusion level and FDR. We defined differential splicing events as those with sufficient coverage at junctions (average junction read count ≥10), a mean inclusion level difference of ≥10% and ≤95% between genotypes and an FDR <0.1, as described by others (Phillips et al, 2020; Zhang et al, 2020).

To compare human and mouse splice variants, an algorithm was developed. First, a master table of all differential splicing events across all mouse and human samples with FDR <0.2 cutoff was made. Then for each homologous gene (as defined in the Ensembl BioMart database), each unique skipped exon (SE) splicing event (defined by the start and end positions of three exons: the spliced exon and its adjacent exons 3' and 5' of it, as identified in the rMATS output tables) in mouse was compared to all splicing events in human for its homologous gene. Briefly, the sequence of each type of exon ("target", "3'-adjacent" or "5'-adjacent") was obtained using the BSgenome R packages (version 1.66.3; https://bioconductor.org/packages/BSgenome; Human sequences from BSgenome.Hsapiens.UCSC.hg38 version 1.4.5 and mouse sequences from BSgenome.Mmusculus.UCSC.mm10 version 1.4.3). These sequences were compared to all unique exons of the same type ("target", "3'-adjacent" or "5'-adjacent") in their homologous counterpart, using the pairwiseAlignment function from Biostrings R package (version 2.66.0; https://bioconductor.org/packages/Biostrings). The mouse and human splicing events with a minimum percentage identity of 70% for all 3 exons and the highest pairwise alignment scores were considered equivalent splicing events. This process was repeated after removing this set of equivalent splicing events, to look for other equivalent splicing events in the same gene. We define conserved splicing events as those differential splicing events (with the same filters as above: average junction read count ≥10, FDR <0.1 and inclusion level difference of ≥10% and ≤95% between genotypes) observed in at least six of the nine datasets and being detected in at least one mouse dataset. The choice of ≥6 datasets was set as a compromise between sensitivity and accuracy given the different quality of the available datasets, as summarized in Fig. EV4. GenomicRanges package (version 1.50.2) was used to handle genomic coordinates information (Lawrence et al, 2013). Finally, the details of splicing events detailed in the comparative splicing analysis figure were manually checked using UCSC genome browser as well as the ExPasy translate tool (https://web.expasy.org/translate) to verify splice variants and stop codons. The UpSetR package (1.4.0) was used to compare between datasets and make upset plots (Conway et al, 2017). To normalize gene expression values across samples, qsmooth function from qsmooth package (1.14.0) (Hicks et al, 2018) was used. Other plots were made using ggplot2 R package (Wickham, 2016). The sashimi plot for the SIRT1 splicing event was generated using rmats2sashimiplot

(https://github.com/Xinglab/rmats2sashimiplot) with the option "--intron_s 15" to scale down intron lengths. hnRNPL consensus motifs were obtained from Ray et al (Ray et al, 2013). FIMO version 5.5.5 (Grant et al, 2011) was used to scan for the presence of the consensus motifs in the skipped exon and the immediately adjacent introns and exons of conserved (inter- and intraspecies) skipped exon changes. CA repeats within a stretch of 14 or 10 nucleotides were searched for using the R package stringr (version 1.5.0), with the regex patterns "CA.{0,2}CA.{0,2}CA.{0,2}CA" or "CA.{0,2}CA.{0,2}CA" respectively.

## ChIP-seq analysis

K562 ChIP-seq dataset is from GSE120104 (ENCODE Project Consortium, 2012a, 2012b). For hnRNPL and Myc ChIP-seq peaks, IDR thresholded peaks files (ENCFF854WAP and ENCFF608CXN, respectively) were obtained from ENCODE portal. For comparison with up- and downregulated genes, genes were considered Myc- and hnRNPL-bound, if a peak overlapped ±2 kb of the gene transcription start site (TSS). Transcription Factor motif enrichment analysis was performed using WebGestalt (Wang et al, 2017). Genome ontology and distance to nearest TSS were computed using HOMER (Heinz et al, 2010).

## Functional annotation analyses

Long non-coding RNAs were identified based on GENCODE annotations (vM23 for mouse and v43 for human). KEGG pathway enrichment analysis was performed with shinyGO (version 0.77) (Ge et al, 2020). Gene ontology (GO) term enrichment analysis was performed with the g:GOSt module of gprofiler (Reimand et al, 2016) using the web app (https://biit.cs.ut.ee/gprofiler/gost) or the R package (version 0.2.2) with the Ensembl 109, Ensembl Genomes 56 database (built on 2023-03-29; http://biit.cs.ut.ee/gprofiler_archive3/e109_eg56_p17). The background gene sets for GO analyses were the list of all genes that have annotations in the GO:BP source database (27,205 genes for Mus musculus, 21,110 genes for Homo sapiens). For both KEGG pathway and GO term enrichment analysis, the set size was limited from 20 to 500 elements and statistical cut-off was set to ≤0.05 (FDR or adjusted $p$ value, respectively). Hallmark GSEA analysis was performed using msigdbr (7.5.1) (Liberzon et al, 2015) and fgsea (1.24.0) (Korotkevich et al, 2021) R packages, with the FDR cut-off ≤0.1. The GSEA comparison heatmap was plotted with ComplexHeatmap (2.14.0) (Gu et al, 2016) R package.

## Statistics

The sample size used for each experiment was based on the magnitude of the effect, the availability of mice, and/or to ensure reproducibility. All data points plotted in the figures represent biological replicates. Male and female mice older than 6 weeks were chosen randomly for experiments and were distinguished only based on their genotype, unless specified above. Blinding was not performed, nor was it necessary since most measurements were performed objectively using automated software. No data was excluded from the analysis. The statistical tests were performed using bioinformatics packages and tools mentioned above or the GraphPad Prism software version 7 or 8. Groups were tested for

normality using the Shapiro-Wilk test (implemented in GraphPad Prism). If all groups passed the normality test, parametric tests were used (unpaired two-tailed *t*-test with Welch's correction for comparing two groups; one-way ANOVA with post hoc Tukey's test for comparing ≥2 groups). Otherwise, non-parametric tests were used (unpaired two-tailed Mann–Whitney test for comparing two groups). Statistical tests and significance cut-off used are indicated in the corresponding figure legends and text.

## Data availability

The datasets produced in this study are available in the following databases: -RNA-seq data: Gene Expression Omnibus GSE242069. The scripts used for different analyses are available at: https://github.com/LabDiNoia/RNAseq_pipeline1. https://github.com/LabDiNoia/hnRNPL_project_2024.

The source data of this paper are collected in the following database record: Accession number S-BSST1365, https://doi.org/10.6019/SBSST1365, biostudies:S-SCDT-10_1038-S44319-024-00152-3.

## Peer review information

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

## Acknowledgements

We thank Anne-Marie Patenaude for assistance with some of the mouse work. We thank the technical assistance of IRCM core facilities personnel: E Massicotte and J Lord with flow cytometry; O Neyret, S Boissel, M Rondeau, P Gingras-Gélinas, and F Couderc with Sanger and NGS experiments; V Calderon with bioinformatics; M Laprise, E-L Thivierge, S Demontigny, M-C Lavallee, C Dube, and P Bergeron with animal technical help. Computations were made on the supercomputer Narval from École de technologie supérieure, managed by Calcul Québec and Compute Canada. The operation of this supercomputer is funded by the Canada Foundation for Innovation (CFI), Ministère de l'Économie, des Sciences et de l'Innovation du Québec (MESI) and le Fonds de recherche du Québec – Nature et technologies (FRQ-NT). This work was supported by operating grants from the Canadian Institutes of Health Research (CIHR) PJ-155944 to JMDN and FDN-148372 to TM. PGS was partially supported by a fellowship from Fondation de recherche en Santé de Québec

(FRQ-S) and the Cole Foundation. TM was supported by a Canada Research Chair (Tier 1) and JMDN by a Bourse de mérite from FRQ-S.

## Author contributions

**Poorani Ganesh Subramani**: Data curation; Software; Formal analysis; Validation; Investigation; Visualization; Methodology; Writing—original draft. **Jennifer Fraszczak**: Investigation; Writing—review and editing. **Anne Helness**: Investigation; Methodology. **Jennifer L Estall**: Supervision; Validation; Writing —review and editing. **Tarik Möröy**: Supervision; Project administration; Writing —review and editing. **Javier M Di Noia**: Conceptualization; Supervision; Funding acquisition; Validation; Visualization; Writing—original draft; Project administration; Writing—review and editing.

Source data underlying figure panels in this paper may have individual authorship assigned. Where available, figure panel/source data authorship is listed in the following database record: biostudies:S-SCDT-10_1038-S44319-024-00152-3, https://doi.org/10.6019/S-BSST1365.

## Disclosure and competing interests statement

The authors declare no competing interests.

# Expanded View Figures

**Figure EV1.   Characterization of mice with *Hnrnpl* deletion in B cells (Related to Figs. 1 and 2).**

(A) Body and spleen weights in *CD21-cre* (+/+), *CD21-cre Hnrnpl*$^{F/+}$ (F/+) and *CD21-cre Hnrnpl*$^{F/F}$ (F/F) mice 11 days post-immunization with NP-OVA. Data compiled from multiple mice (*n* = as indicated) and 4 experiments. (B) Splenic B and T cell counts for the mice in (A). (C) GC B cell counts for the mice in (A). (D) Gating and representative flow cytometry plot and proportions of splenic NF, MZ, FO, and GC B cell subpopulations for the mice in (A). (E) Body and spleen weights and splenocyte counts, of irradiated mice that received BM cells from μ*MT* mice and either *CD21-cre Hnrnpl*$^{F/+}$ (F/+) or *CD21-cre Hnrnpl*$^{F/F}$ (F/F) mice, (*n* = 9 F/+, *n* = 8 F/F. Data compiled from two experiments. (F) Initial proportion of CD45.1:CD45.2 BM cell mixes from CD45.1 WT and CD45.2 either *Rosa*$^{mT/mG}$ *CD21-cre Hnrnpl*$^{F/+}$ (F/+) or *Rosa*$^{mT/mG}$ *CD21-cre Hnrnpl*$^{F/F}$ (F/F), used for reconstituting irradiated C57BL6/J CD45.2 mice. (G) Total B cell counts (symbols are individual mice) and proportion of splenic B cell subpopulations (with representative flow cytometry plots below) in mice reconstituted with BM mixes from (F), *n* = 8 mice per group, from two experiments. (H) Representative flow cytometry plots of the proportions of splenic B cell subpopulations newly formed (NF), marginal zone B (MZ), follicular B (Fo), and germinal center (GC) B cells in each group of mice reconstituted with BM mixes from (F). Data information: In (A–E, G leftmost panel), data were presented for individual mice with lines indicating mean values. In (G), data were presented as mean ± SD. Statistical tests were: (A–D) one-way ANOVA with post hoc Tukey's multiple comparison test, (E) unpaired two-tailed *t*-test with Welch's correction, (G) unpaired, two-tailed Mann–Whitney test, with *p* < 0.05 considered as significant differences in group means.

                                                                        

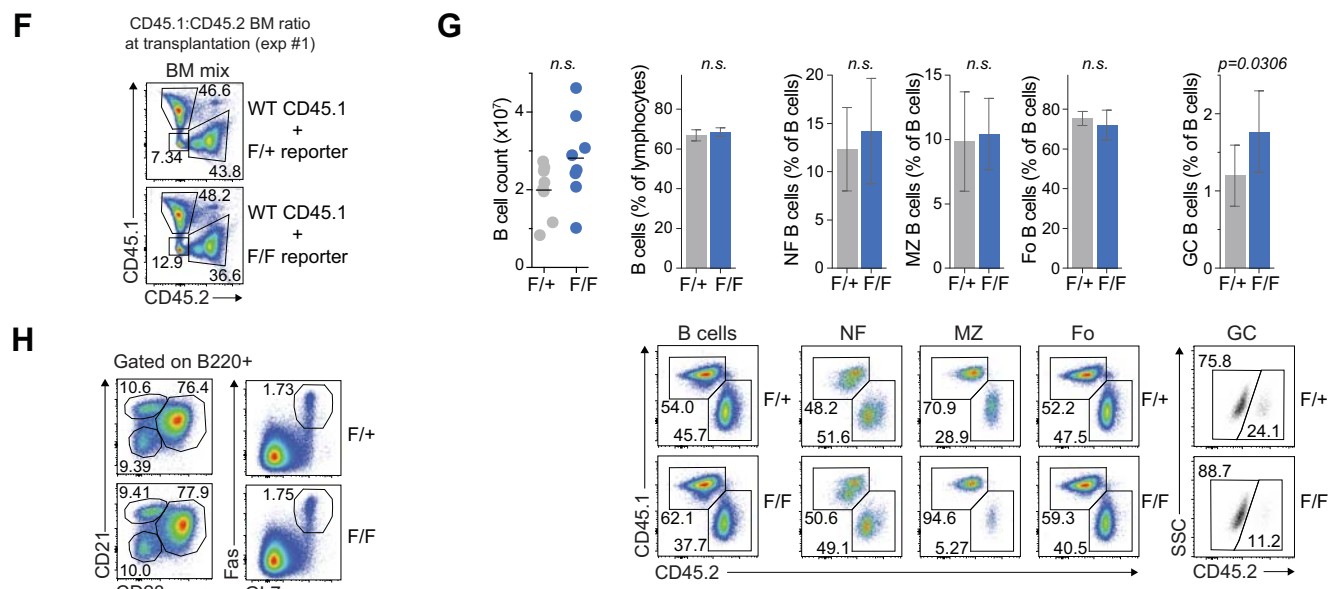

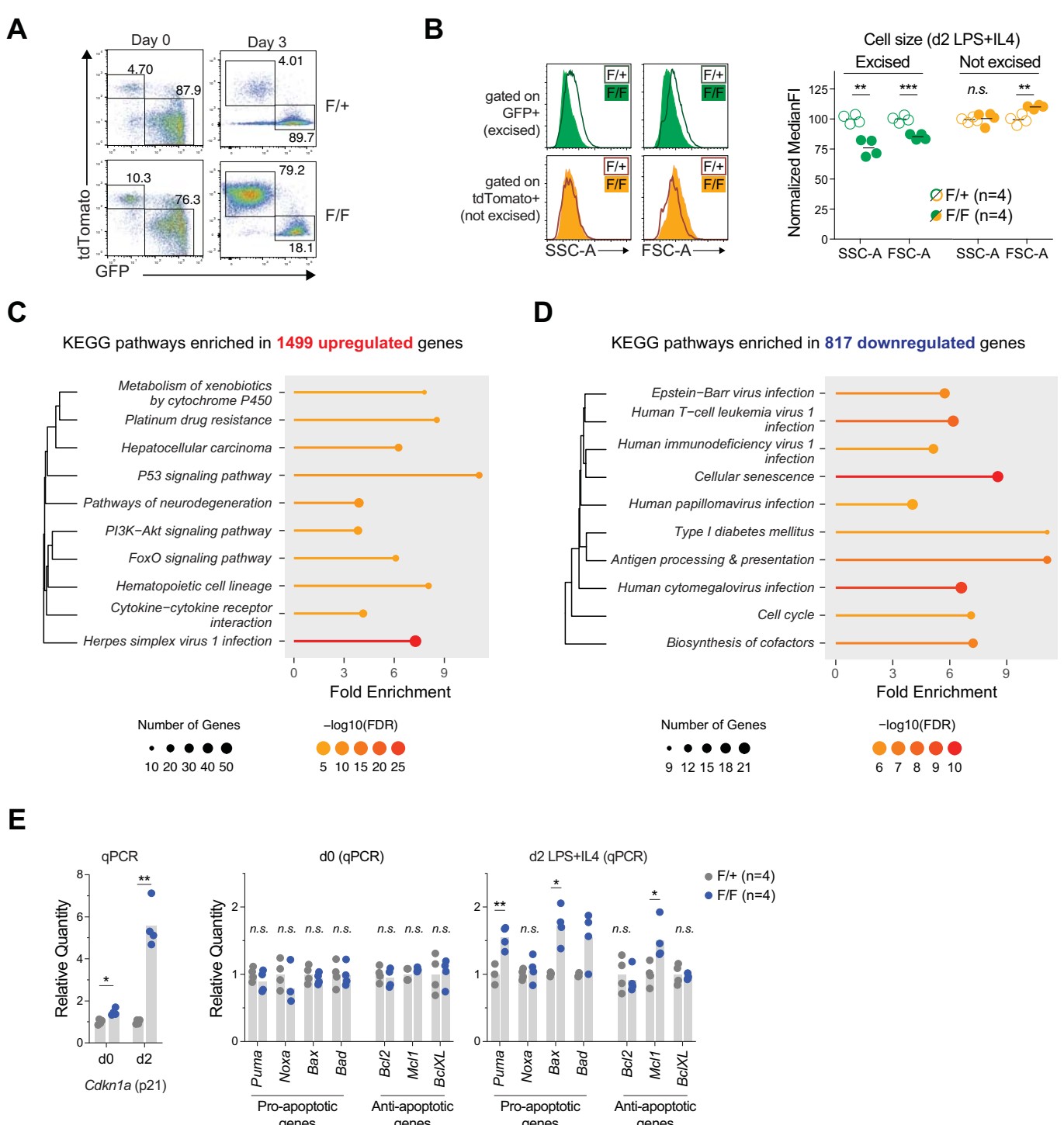

**Figure EV2. Cellular and transcriptional effects of hnRNPL loss in B cells (Related to Fig. 3).**

(A) Representative flow cytometry plots showing the proportions of hnRNPL-excised (GFP+; green) and non-excised (tdTomato+; orange) splenic B cells from *Rosa^{mT/mG} CD21-cre Hnrnpl^{F/+}* (F/+) or *Rosa^{mT/mG} CD21-cre Hnrnpl^{F/F}* (F/F), before and after (day 3) ex vivo activation with LPS/IL-4. (B) Representative flow cytometry histograms of parameters indicating cell size (FSC) and granularity (SSC) of activated B cells from (A), (n = 4 mice per group). (C) Top ten KEGG pathways enriched in genes upregulated in hnRNPL-deficient versus WT cells. The dendrograms indicate the degree of similarity (shared genes) between pathways. (D) Top ten KEGG pathways enriched in genes downregulated in hnRNPL-deficient versus WT cells. Legends, as in (C). (E) Quantification of gene expression by RT-qPCR for p21 (*Cdkn1a*), pro- and antiapoptotic factors in B cells, resting (d0) or activated (48 h post-LPS/IL-4) (d2), from *CD21-cre Hnrnpl^{F/+}* (F/+) or *CD21-cre Hnrnpl^{F/F}* (F/F) mice, (n = 4 mice each group). Data information: In (B, E), data were presented for individual mice with lines or bars indicating mean values. Data compiled from two experiments. Statistical significance ($p < 0.05$) by unpaired, two-tailed $t$-test with Welch's correction (*$p < 0.05$; **$p < 0.01$; ***$p < 0.001$).

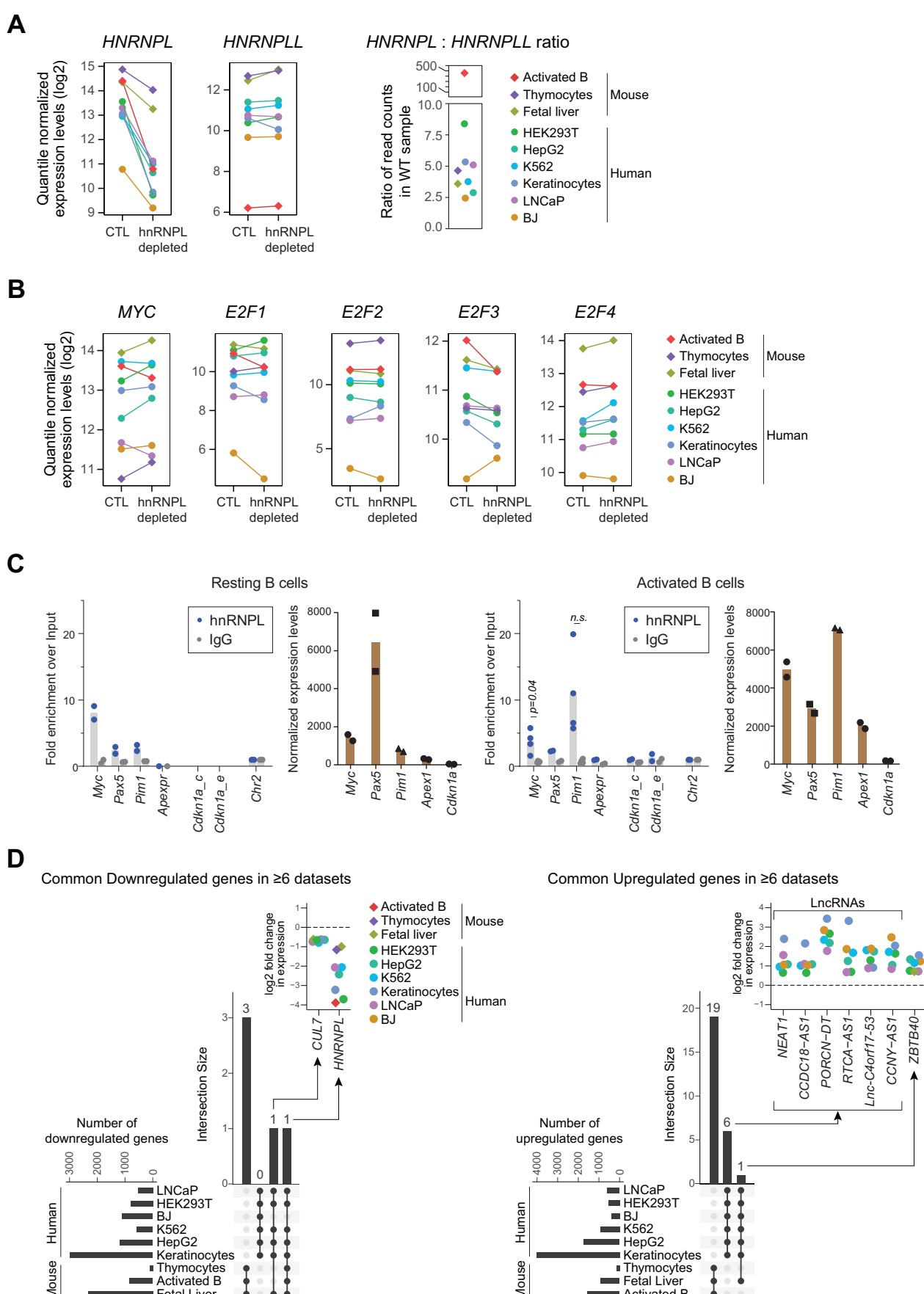

**Figure EV3. Conserved roles of hnRNPL – (Related to Fig. 5).**

(A) Quantile-normalized expression levels of *HNRNPL* and *HNRNPLL* in the indicated control (CTL) or hnRNPL-depleted cells, and ratio of *HNRNPL* over *HNRNPLL* read counts in each control cell type. (B) Quantile-normalized expression levels of the indicated genes in the various control or hnRNPL-depleted cells. (C) hnRNPL occupancy by ChIP-qPCR at the indicated loci in WT splenic B cells resting or activated (LPS/IL-4 for 2 days) and normalized expression levels of the corresponding genes in the same cells (GSE90094). (D) Comparison of significantly (adjusted *p* value <0.1 and fold-change ≥1.5) up- and downregulated genes upon hnRNPL depletion shared by the indicated cell types. All non-zero intersections of ≥6 datasets are shown. The insets show the relative expression (log$_2$ fold-change compared to Control) of selected genes in the same cell types. Data information: In (A, B) data points are individual datasets, in (C) data points are individual mice with bars indicating means ($n = 2$ or 4 biological replicates, from two experiments). Statistical testing was done for amplicons with four replicates by two-tailed paired *t*-test.

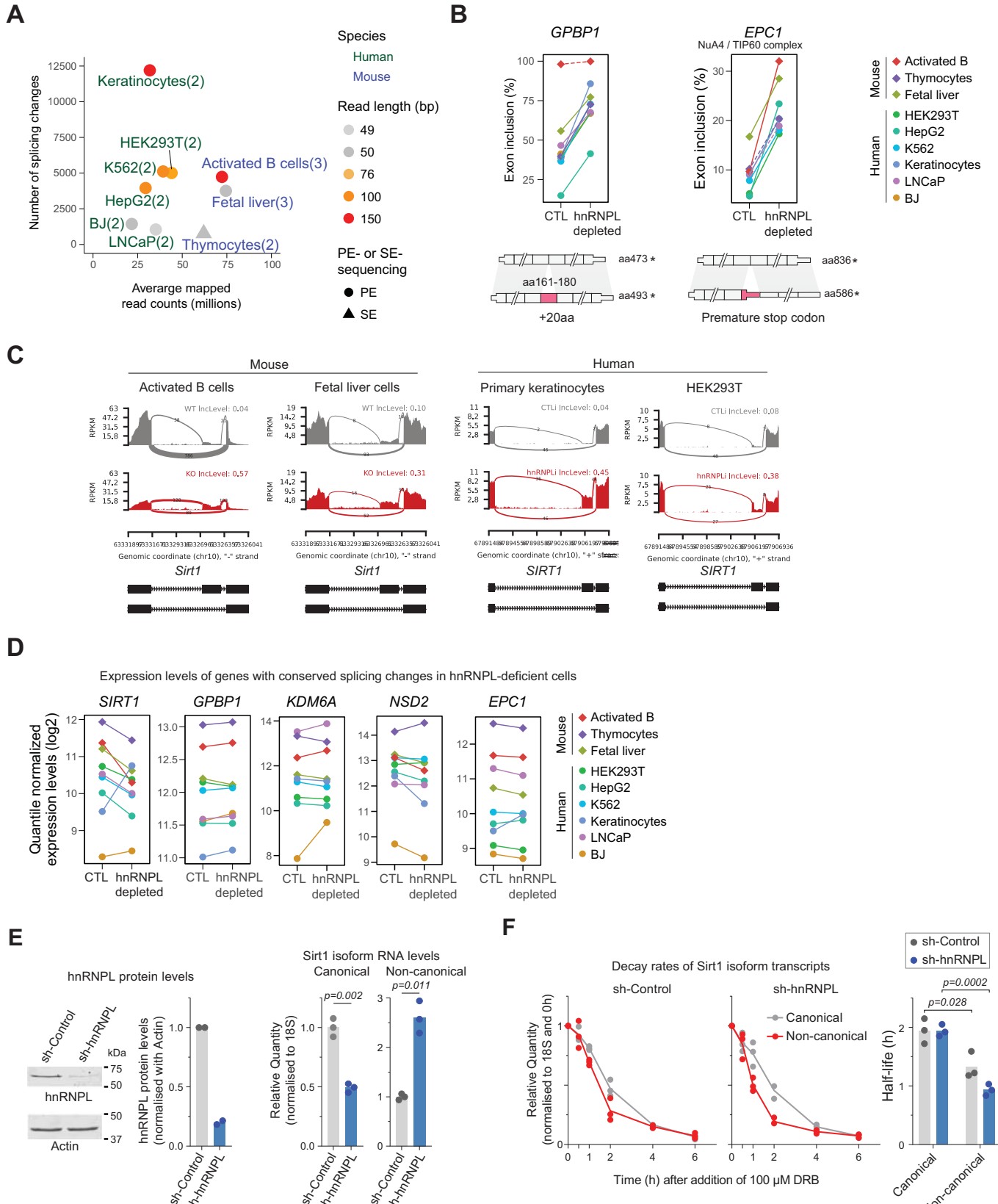

**Figure EV4. Comparison of hnRNPL depletion effects among cell types (Related to Fig. 6).**

(A) Number of splicing changes detected in different hnRNPL-deficient cell types as a function of read depth. Read length and species of the datasets are indicated by codes (legends), as well as number of replicates per condition (in parenthesis). (B) Mean exon inclusion levels for the indicated splicing events in control (CTL) and hnRNPL-depleted cell types. The transcript schemes indicate the exons (in red, the exon more included in hnRNPL-deficient cells), positions of amino acids coded by the included exon and stop codons in the respective human transcripts. (C) Sashimi plot showing the inclusion of an intermediate exon in *SIRT1* regulated by hnRNPL status in selected cell types. (D) Quantile-normalized expression levels of indicated genes in the various CTL or hnRNPL-depleted cell types. (E) Representative western blot and quantification of hnRNPL protein levels in CH12-F3 cells expressing shRNA control or targeting hnRNPL ($n = 2$ biological replicates). Steady-state levels of canonical and non-canonical Sirt1 transcript isoforms measured by RT-qPCR normalized to the sh-Control ($n = 3$ biological replicates). (F) Decay rates and half-lives of canonical and non-canonical Sirt1 isoforms following transcription inhibition with 100 μM DRB in sh-Control and sh-hnRNPL CH12-F3 cells measured by RT-qPCR ($n = 3$ biological replicates). Data information: In (A, B, D), data points represent individual datasets. In (E, F), data points are biological replicates with bars indicating means. Statistical significance ($p < 0.05$) tested by unpaired two-tailed *t*-tests with Welch's correction.

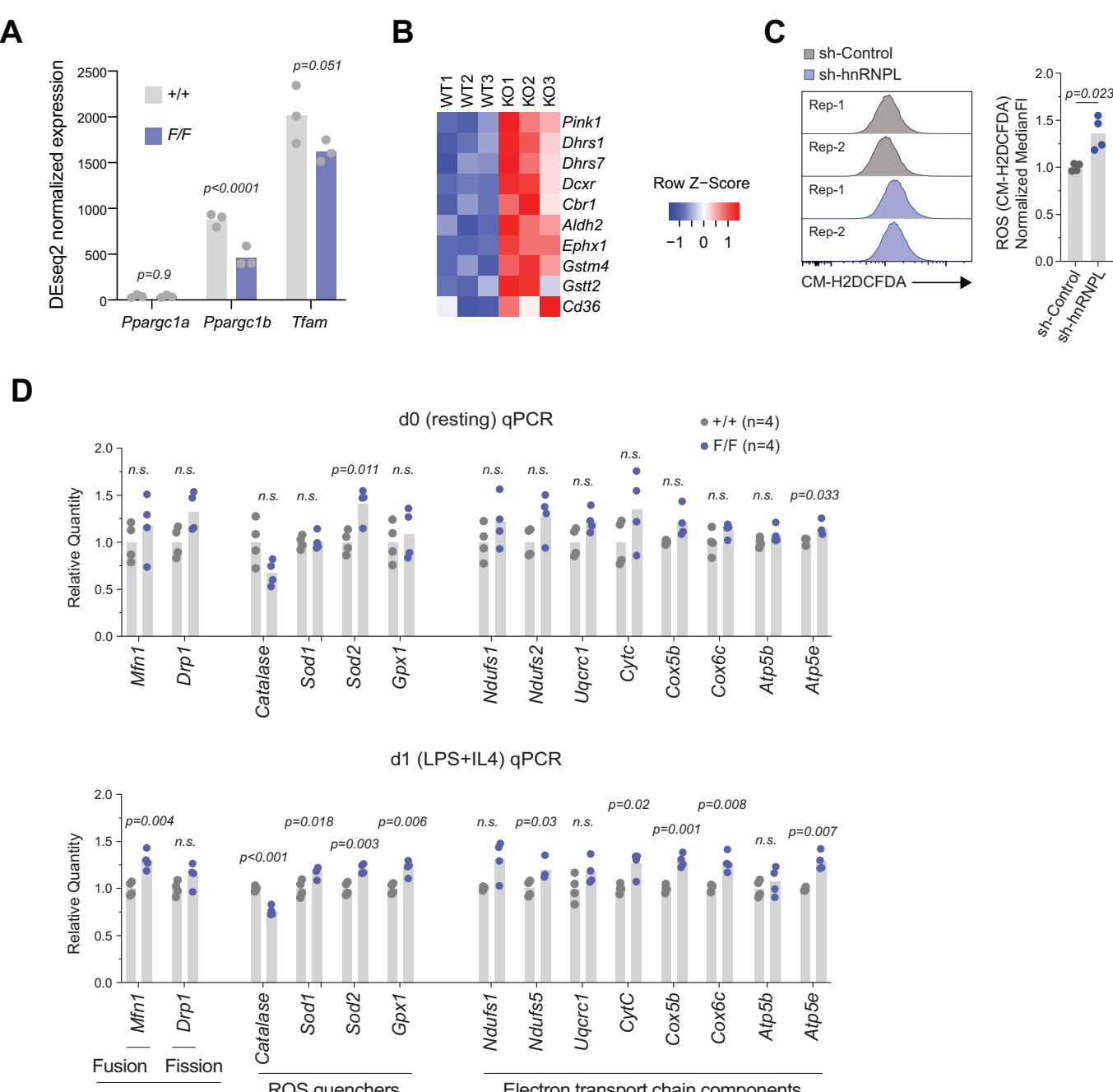

**Figure EV5. Mitochondrial function-related gene expression (Related to Fig. 7).**

(A) Expression of the indicated genes from RNA-seq of splenic B cells activated ex vivo for 1 day with LPS/IL-4 from hnRNPL-deficient (GFP+ cells from $Rosa^{mT/mG}$ $CD21$-$cre$ $Hnrnpl^{F/F}$) and WT (GFP+ cells from $Rosa^{mT/mG}$ $CD21$-$cre$) mice. (B) Heatmap of selected genes in individual biological replicates from RNA-seq from data in Dataset EV1. (C) Representative flow cytometry histograms of ROS levels measured by the CM-H2DCFDA probe in CH12-F3 cells expressing shRNA targeting hnRNPL (sh-hnRNPL) or control (sh-Control). ($n = 4$ biological replicates per genotype from two experiments). (D) Relative expression level of selected genes measured by RT-qPCR in resting (d0) and ex vivo LPS/IL-4-activated splenic B cells from $CD21$-$cre$ (+/+) and $CD21$-$cre$ $Hnrnpl^{F/F}$ (F/F) mice. ($n = 4$ biological replicates per genotype from two experiments). Data information: In (A, C, D), data points are individual biological replicates with bars indicating means; in (B), row normalized data, both using values from Dataset EV1. Statistical significance ($P$ adj <0.1 or $p$ < 0.05) was tested (A) by Deseq2, (C, D)) by unpaired two-tailed $t$-tests with Welch's correction.

