## [Peer Review File · EMBO Reports]

Conserved role of hnRNPL in alternative splicing of epigenetic modifiers enables B cell activation

Poorani Ganesh Subramani, Jennifer Fraszczak, Anne Helness, Jennifer Estall, Tarik Moroy, and Javier Di Noia

Corresponding author(s): Javier Di Noia (Javier.Di.Noia@ircm.qc.ca)

Review Timeline:

Submission Date:	5th Oct 23
Editorial Decision:	27th Nov 23
Revision Received:	20th Mar 24
Editorial Decision:	5th Apr 24
Revision Received:	15th Apr 24
Accepted:	24th Apr 24

Editor: Achim Breiling

Transaction Report:

Dear Dr. Di Noia,

Thank you for the transfer of your research manuscript to EMBO reports. I have now received the reports from the three referees that were asked to evaluate your study, which can be found at the end of this email.

As you will see, the referees think that the findings are of interest. However, they have several comments, concerns, and suggestions, indicating that a revision of the manuscript is necessary to allow publication of the study in EMBO reports. As the reports are below, and all the referee concerns need to be addressed, I will not detail them here.

Given the constructive referee comments, I would like to invite you to revise your manuscript with the understanding that all referee concerns must be addressed in the revised manuscript or in a detailed point-by-point response. Acceptance of your manuscript will depend on a positive outcome of a second round of review. It is EMBO reports policy to allow a single round of revision only and acceptance of the manuscript will therefore depend on the completeness of your responses included in the next, final version of the manuscript.

- 1) a .docx formatted version of the final manuscript text (including legends for main figures, EV figures and tables), but without the figures included. Figure legends should be compiled at the end of the manuscript text.
- 2) individual production quality figure files as .eps, .tif, .jpg (one file per figure), of main figures (up to 8) and EV figures. Please upload these as separate, individual files upon re-submission.

- 4) a complete author checklist, which you can download from our author guidelines

(<https://www.embopress.org/page/journal/14693178/authorguide>). Please insert page numbers in the checklist to indicate where the requested information can be found in the manuscript. The completed author checklist will also be part of the RPF.

- 5) that primary datasets produced in this study (e.g. RNA-seq, ChIP-seq, structural and array data) are deposited in an

appropriate public database. If no primary datasets have been deposited, please also state this in a dedicated section (e.g. 'No primary datasets have been generated and deposited'), see below.

The accession numbers and database should be listed in a formal "Data Availability" section (placed after Materials & Methods) that follows the model below. This is now mandatory (like the COI statement). Please note that the Data Availability Section is restricted to new primary data that are part of this study. This section is mandatory. As indicated above, if no primary datasets have been deposited, please state this in this section

Data availability

8) Regarding data quantification and statistics, please make sure that the number "n" for how many independent experiments were performed, their nature (biological versus technical replicates), the bars and error bars (e.g. SEM, SD) and the test used to calculate p-values is indicated in the respective figure legends (also for potential EV figures and all those in the final Appendix). Please also check that all the p-values are explained in the legend, and that these fit to those shown in the figure. Please provide statistical testing where applicable. Please avoid the phrase 'independent experiment', but clearly state if these were biological or technical replicates. Please also indicate (e.g. with n.s.) if testing was performed, but the differences are not significant. In case n=2, please show the data as separate datapoints without error bars and statistics. See also: <http://www.embopress.org/page/journal/14693178/authorguide#statisticalanalysis>

9) Please also note our reference format:

10) We updated our journal's competing interests policy in January 2022 and request authors to consider both actual and perceived competing interests. Please review the policy <https://www.embopress.org/competing-interests> and update your competing interests if necessary. Please name this section 'Disclosure and Competing Interests Statement' and put it after the Acknowledgements section.

11) We now use CRediT to specify the contributions of each author in the journal submission system. CRediT replaces the author contribution section. Please use the free text box to provide more detailed descriptions and do not provide your final manuscript text file with an author contributions section. See also our guide to authors:

<https://www.embopress.org/page/journal/14693178/authorguide#authorshippinguidelines>

12) We would encourage you to use 'Structured Methods', our new Materials and Methods format. According to this format, the Materials and Methods section should include a Reagents and Tools Table (listing key reagents, experimental models, software and relevant equipment and including their sources and relevant identifiers) followed by a Methods and Protocols section in which we encourage the authors to describe their methods using a step-by-step protocol format with bullet points, to facilitate the adoption of the methodologies across labs. More information on how to adhere to this format as well as downloadable

templates (.doc or .xls) for the Reagents and Tools Table can be found in our author guidelines (section 'Structured Methods'):

Please order the manuscript sections like this, using these names:

Title page - Abstract - Keywords - Introduction - Results - Discussion - Materials and Methods - Data availability section - Acknowledgements - Disclosure and Competing Interests Statement - References - Figure legends - Expanded View Figure legends

I look forward to seeing a revised version of your manuscript when it is ready. Please let me know if you have questions or comments regarding the revision.

Yours sincerely,

Referee #1:

The manuscript reports data using sophisticated mouse models and molecular biology approaches, that hnRNPL is essential for the activation of B cells (particularly for the transition from resting to activation) via alternative splicing of histone modifiers which changes the transcriptional program and metabolism. Their analysis of 8 hnRNPL depleted cell types and RNA seq data from activated cells show that MYC and E2F transcriptional programs are required for proliferation. An interesting finding in this paper is that they found AS events affect the histone modifiers such as KDM6A, NSD2 and SIRT1. The study provides mechanistic insight into a very complex and understudied aspect of gene regulation in a physiological context. This is good work and should be published following some achievable revisions.

A couple of points that could be clarified to make the manuscript clearer and more contemporary:

Are TRP53 protein amounts known to be increased in the HNRNPL knockout B cells? Could this contribute to the phenotype?

The manuscript will benefit from consideration (perhaps best placed in the discussion) of other recent work on splicing regulators in B cell activation and the GC response.

<https://pubmed.ncbi.nlm.nih.gov/37474714/>

<https://pubmed.ncbi.nlm.nih.gov/34772950/>

<https://pubmed.ncbi.nlm.nih.gov/36997512/>

Minor revisions:

On line 54 cryptic exon inclusion is mentioned, but it is not mentioned later on. If this is an important mechanism by which hnRNPL acts, what does the RNAseq data have to say about it. Some clarification will be helpful here.

For the text relating to figure 1E-F- it would be helpful to clarify the choice of day 12 for analysis of the response and in figure 1H- why 11 days post-immunisation?

Figure 1J- Are they sure about the specificity of Ab for hnRNPL? If it is possible it reacts with the paralog it would be helpful to clarify that in the text.

Figure 3A is empty and filled circles can be more clearly identified.

Figure 6- Clarify why these are conserved splicing events?

Line 297- they indicated they "turned to analyze pre-mRNA splicing". It may be best to change it to "alternative mRNA splicing".

Figure 6-A- What is the inclusion level cut off.

Line 303- The most common AS pattern is exon skipping and it is worth noting that here.

Line 312- How do they know that it cannot be compensated by other paralogs?

Figure 7B- it is unclear as it is written as Rpl35a in the figure but Rps35a in the text.

Referee #2:

This manuscript from Di Noia and colleagues investigates roles of the multifunctional RNA-binding protein hnRNPL in B cell development and activation. The authors used CD21-cre to conditionally delete hnRNP L from peripheral B cells, finding that B cells lacking hnRNPL have activation defects and undergo elevated rates of apoptosis. Class switching is also found to be impaired in hnRNPL-depleted cells. The authors perform RNA-seq from hnRNPL-depleted and control activated B-cells and compare their own data to public datasets of hnRNPL depletion studies from several additional cell types. Consistent with impaired activation and increased apoptosis, the authors find that p53 pathway targets are upregulated and Myc and E2F targets are downregulated in hnRNPL-deficient activated B cells. Of these pathways, Myc and E2F pathways appear to be downregulated in several hnRNPL-depleted cell types.

The authors further investigate the effects of hnRNPL deletion on alternative splicing, identifying a small set of chromatin regulators that are consistently alternatively spliced across the panel of hnRNPL depletion datasets. One such target is SIRT1, prompting examination of mitochondrial defects in the knockout cells. This is an interesting system, and understanding cell type-specific functions of common splicing factors is an important goal. Overall, the experiments appear to be well designed and executed.

The major drawback of the paper is that there are no clear connections between the effects of hnRNPL depletion on RNA biogenesis and the observed cellular phenotypes. Please see below for detailed comments.

Major points:

1. As noted above, the paper is lacking a clear link between the mis-spliced or differentially expressed genes identified and the B cell phenotypes observed. Two areas stand out: 1) the molecular cause(s) of Myc/E2F pathway downregulation are not identified and 2) the mitochondrial dysfunction investigated in Figure 7 is not directly attributed to SIRT1 regulation. More information in either of these areas would significantly strengthen the manuscript.
2. The rMATS and associated downstream analyses should be more completely described in the methods section, and the approach and interpretation should be modified to take into account significant shortcomings of the rMATS algorithm. In particular, the rMATS algorithm performs very poorly on retained introns and mutually exclusive exons. The mutually exclusive exon identifications are likely almost entirely spurious (the number of validated MXE events in the human genome is less than or similar to the ~800 claimed to be regulated by hnRNPL in figure 6A; see for example PMID 29242366), complicating the comparison of alternative splicing and differential gene expression in Figure 6B. Further, rMATS generates many spurious calls if the default coverage filters are used. In addition, no information is given about the background set of genes used for the GO analysis in Figure 6C, which is critical for interpretation.

Minor points:

1. SD should in most instances be used rather than SE.
2. The authors should describe what lncRNA annotations were used in Figure 5B.

Referee #3:

Class switch recombination (CSR) is a DNA recombination-coupled with deletion mechanism that occurs at the immunoglobulin heavy chain locus (IgH) for determination of effector function of expressed antibodies in B cells. CSR is driven by the activity of the enzyme Activation Induced Deaminase (AID), whose enzymatic activity has been widely debated. In this regard, AID has been suggested to edit mRNAs (for their activation) that eventually catalyze CSR, or has been shown to deaminate DNA to induce mutations and DNA breaks directly. Previous studies by Hu, Honjo et al (PMID: 25902538) have suggested that HnRNPL functions as a co-factor of AID that regulates CSR. Hu et al's observations pitched AID's function as a RNA editing enzyme, and contrasts with the more widely accepted function of AID as a DNA deaminase in CSR.

In this study, Subramani et al, very elegantly establishes using mouse models that HnRNPL does play a role in CSR but this is more likely due to its function in cellular metabolism. Using a combination of conditional allele mouse model and various cell lines, Subramani et al demonstrate that hnRNP-L is important for promoting proliferation of activated B cells and preventing apoptosis. hnRNP-L depleted cells demonstrate wide range of gene expression changes (relatively, enriched with Myc and E2F transcriptional network) including expression of important histone modifiers like KDM6A, NSD2, and SIRT1. As would be expected following reduced SIRT1 expression, hnRNPL-deficient B cells had defective mitochondria leading to ROS overproduction. Mitochondrial dysregulation is aligned with B cell activation defects. The experiments are well done with proper

controls. The statistical significance of the experiments are well described.

Overall, this is an important study and will be well received in the fields of B Cell biology and CSR. I do not have any major concerns and suggest publication.

If there is anything I can suggest, the authors could rationalize the previous conclusions regarding CSR by Hu et al slightly better. Primary B cells requires the B cell activation pathways to proliferate and survive. Hu et al, performed their experiments in B cell lines where cell survival may not need these pro-survival mechanisms. Thus, role of hnRNP may also be important for CSR (unlikely via RNA editing though) and also be important for mitochondrial biology. The authors could elaborate on this point since their study will be likely be followed and cited based on their judgement on the role of these hnRNP proteins in CSR.

Second, the role of hnRNP-L in alternative splicing control is quite interesting and relevant. However, the evidence provided here does not complete address how this works. The discussion could be tuned to address how exactly hnRNP-L works for alternative splicing or the conclusions could be tempered.

Overall, this is a terrific study and deserves publication.

Montréal, March 20, 2024

Dear Dr Breiling,

Thank you for handling our manuscript EMBOR-2023-58272-T. We also thank the Referees for the positive evaluation and constructive suggestions. We have addressed all their comments in the point-by-point response below. The changes introduced in response to the comments have greatly improved the revised manuscript. Major text changes or those addressing Referees concerns are highlighted in the word document of the manuscript. Small changes introduced to improve grammar or text clarity were not highlighted.

The major data modifications to the revised version are as follows:

- 1 - Improved splicing analysis, resulting in modified figure panels 6A-F, EV4A,B,D, plus additional description in the Methods section, to address a major concern of Referee 2 and specific comments of the other Referees. New Table S3 lists the differential splicing events in hnRNPL deficient versus control B cells.
- 2 – We generated new data to address the other major comment from Referee 2 and a comment from Referee 3, which asked for a more direct link between splicing events regulated by hnRNPL and cellular phenotypes (new figure panels 4K, 6F, 6I, EV4E, EV4F).

We discuss in the point-by-point the limitations we face when trying to link specific splicing events to a given phenotype because of the large number of splicing and gene expression changes upon hnRNPL deletion. We have made this limitation clear in the Discussion. However, we now better highlight our finding that hnRNPL has a conserved role in regulating the splicing of transcriptional regulators including histone modifiers, and that this regulation has consequences for the expression or activity of these factors. In the revised manuscript we provide experimental data to support this for selected examples. We also show evidence of global epigenetic alterations in hnRNPL deficient B cells. One of these alterations can be linked to the effect of a conserved splicing change in KDM6A. This further supports our tenet that hnRNPL regulates gene expression indirectly, at least in part via regulating splicing of histone modifiers that can affect the epigenetic landscape. We think this is a novel finding from our analysis, which can explain the phenotypes we observe in hnRNPL deficient B cells and compensates for the limitations that any paper has.

Additional changes were made to meet journal format and editorial requests, which included:

- Supplementary figures were converted into expanded view figures. Figures S1 and S2 were merged into figure EV1, the subsequent supplementary figures were renamed as EV2-EV5.
- Datapoints are shown in all figure panels with $n < 5$ and Statistics section in Methods was expanded.
- We have no Appendix. All supplementary data are Excel tables, some of which are large and/or have multiple tabs. This does not adapt well to a single pdf format as indicated for Supplementary data, so they are provided as Excel data files.
- Source data was deposited in BioStudies and can be accessed at <https://www.ebi.ac.uk/biostudies/studies/S-BSST1365?key=6e7fc71-882d-4b74-a15c-dd2de83ad553> and, as before, original RNA-seq data is at GSE242069 using the token: cfibkkkwhlknrcx.

Thank you very much for considering our work; we look forward to a positive response,

s

Javier M Di Noia

Point by Point responses.

Referee #1:

The manuscript reports data using sophisticated mouse models and molecular biology approaches, that hnRNPL is essential for the activation of B cells (particularly for the transition from resting to activation) via alternative splicing of histone modifiers which changes the transcriptional program and metabolism. Their analysis of 8 hnRNPL depleted cell types and RNA seq data from activated cells show that MYC and E2F transcriptional programs are required for proliferation. An interesting finding in this paper is that they found AS events affect the histone modifiers such as KDM6A, NSD2 and SIRT1. The study provides mechanistic insight into a very complex and understudied aspect of gene regulation in a physiological context. This is good work and should be published following some achievable revisions.

Thank you for the positive comments and appreciating the quality of our work.

A couple of points that could be clarified to make the manuscript clearer and more contemporary:

Are TRP53 protein amounts known to be increased in the HNRNPL knockout B cells? Could this contribute to the phenotype?

A few publications have analyzed the link of hnRNPL with p53 in other systems. We summarized them below in more detail but, collectively, they show that, while hnRNPL depletion consistently leads to apoptosis, its effect on p53 itself is cell type dependent.

Trp53 protein levels were increased in hnRNPL deficient mouse fetal liver cells (Gaudreau et al, 2016). Similarly, in mouse ESC, hnRNPL depletion stabilized p53 protein (Li et al, 2015). In contrast, others have reported a role for hnRNPL in p53 translation following exogenous DNA damage, with hnRNPL depletion preventing p53 accumulation in two mouse cell lines (Seo et al, 2017). In the same vein, in human renal Wilms tumors and prostate cancer cell lines, hnRNPL was found to bind the Trp53 transcript, yet hnRNPL depletion did not change, and even decreased, p53 protein levels (Luo et al, 2019; Zhou et al, 2017). In the latter systems, increased apoptosis was ascribed to a decrease in antiapoptotic BCL2 (Luo et al, 2019; Zhou et al, 2017). Finally, an siRNA screen in a human lung cancer cell line identified hnRNPL as a negative regulator of the p53 pathway by transcriptomics, but hnRNPL depletion did not change the p53 protein levels (Siebring-Van Olst et al, 2017).

These references are now briefly summarized in the introduction of the manuscript.

*The status of p53 protein was unknown in hnRNPL-deficient B cells. We have now probed this by WB in hnRNPL-proficient versus -deficient naïve and activated B cells and indeed there is an increase in p53 levels (**new Figure 4K**), consistent with the activation signature observed by RNA-seq, and the increased apoptosis phenotype.*

The manuscript will benefit from consideration (perhaps best placed in the discussion) of other recent work on splicing regulators in B cell activation and the GC response.

<https://pubmed.ncbi.nlm.nih.gov/37474714/>

<https://pubmed.ncbi.nlm.nih.gov/34772950/>

<https://pubmed.ncbi.nlm.nih.gov/36997512/>

We agree. The article by Huang et al was referenced in the introduction but we have added the others and modified the wording to emphasize the relevance for splicing in activated and GC B cells. We also added the a sentence to the Discussion, highlighting the common link to Myc activity.

Minor revisions:

On line 54 cryptic exon inclusion is mentioned, but it is not mentioned later on. If this is an important mechanism by which hnRNPL acts, what does the RNAseq data have to say about it. Some clarification will be helpful here.

This mention was influenced by the paper by McClory et al 2018 (PMID: 29581412). Several splicing factors, including hnRNPL, can prevent the inclusion of intronic sequences that resemble but are not considered real exons, based on their lack of interspecies conservation. Since we did not analyze this specific aspect, we have emphasized its role on alternative splicing, and we focus on discussing events conserved across systems.

For the text relating to figure 1E-F- it would be helpful to clarify the choice of day 12 for analysis of the response and in figure 1H- why 11 days post-immunisation?

This was just a matter of convenience since both data are from the same mice. We bled the mice at day 11 for the ELISA and euthanized them for flow cytometry analyses the next day.

Figure 1J- Are they sure about the specificity of Ab for hnRNPL? If it is possible it reacts with the paralog it would be helpful to clarify that in the text.

The decrease in signal specifically in the KO cells indicates the Ab recognizes hnRNPL. The residual signal in the WB in Fig 1J most likely reflects that a fraction of the naive CD21-cre Hnrnp1^{F/F} B cells do not completely delete the gene. This is consistent with results in Figure 2D, which suggest that hnRNPL deletion has a small effect on resting B cells when competing with WT cells.

A cross-reaction with paralogs is highly unlikely. As shown in the screenshot below, hnRNP-LL is the closest paralog and has many identities to hnRNP-L over the region used as immunogen (Ab peptide), but we show that hnRNP-LL is barely expressed (Fig 1A) and not upregulated upon loss of hnRNP-LL (Fig EV3A) in B cells. The similarity with Ptbp1 is very low.

heterogeneous nuclear ribonucleoprotein L-like [Mus musculus]

Sequence ID: NP_659051.3 Length: 591 Number of Matches: 1
See 1 more title(s) See all Identical Proteins(IPG)

Range 1: 114 to 154 GenPept Graphics Next Match Pn

	Score	Expect	Method	Identities	Positives	Gaps
	57.0 bits(136)	6e-11	Composition-based stats.	28/45(62%)	33/45(73%)	4/45(8%)
Ab peptide	Query	6	GGGENYDDPHKTPASPVVHIRGLIDGVVEADLVEALQEFGPISYV	50		
			GGG + HK SPVWH+RGL + VVEADLVEAL+FG I YV			
hnRNPL	Sbjct	114	GGGSH---HKVSVSPVHVHVRGLCESVVEADLVEALEKFGTICVY	154		

Ptbp1

Sequence ID: Query_108581 Length: 531 Number of Matches: 1
Range 1: 37 to 86 Graphics Next Match Pn

	Score	Expect	Method	Identities	Positives	Gaps
	21.6 bits(44)	0.002	Composition-based stats.	12/50(24%)	23/50(46%)	2/50(4%)
Ab peptide	Query	1	AAGGGGGGENYDDPHKTPASP--VVHIRGLIDGVVEADLVEALQEFGPIS	48		
			+A G + + ++ P V+HIR L V E +++ FG ++			
Ptbp1	Sbjct	37	SAANGDSKKFKGDSRSAGVPSRVIHIRKLPIDVTEGEVLSLGLPFGKVT	86		

Since hnRNPL is essential in most cell lines (DepMap) we cannot produce a knockout cell line to perform gold standard validation of the antibody, but the signal is also strongly reduced in CH12 B cell line by an inducible shRNA targeting hnRNPL (new Figure EV4E), which also expresses the paralogs. Thus, we are confident that the remaining signal observed in WB largely reflects residual hnRNPL in CH12 and the background of unexcised cells in the case of CD21-cre Hnrnp1^{F/F} mouse B cells.

Figure 3A is empty and filled circles can be more clearly identified.

Legends were added to the plots to identify the hnRNPL status of each set.

Figure 6- Clarify why these are conserved splicing events?

Splicing was reanalyzed following the advice from Referee 2. As is now stated in methods and the results, we define as "conserved" any specific splicing event that was present in ≥ 6 of the 9 datasets, which is a compromise between sensitivity and accuracy given the different quality of the available datasets. Please refer to answer to major point 2 from Referee 2 for full details.

Line 297- they indicated they "turned to analyze pre-mRNA splicing". It may be best to change it to "alternative mRNA splicing"

Done, thank you.

Figure 6-A- What is the inclusion level cut off.

This is 10%. The inclusion level, plus additional filters used according to the suggestions of Referee 2, are detailed in the revised results, figure legends, and methods for the splicing analysis. We did not change the cut off but implemented a more stringent minimal coverage filter. The conclusions did not change. Please refer to answer to major point 2 from Referee 2 for full details.

Line 303- The most common AS pattern is exon skipping and it is worth noting that here.

This is now noted in the text in the corresponding results section.

Line 312- How do they know that it cannot be compensated by other paralogs?

This can be inferred by the fact that eliminating hnRNPL leads to clear phenotypes. We cannot rule out that hnRNPLL could compensate for hnRNPL if it was expressed. But since we show that Hnrnp11 is not expressed or induced upon Hnrnp1 deletion in B cells, it does not. Any other potential compensation by other hnRNPs that are expressed in B cells is ruled out by the clear phenotypic effect. We have added a brief mention of these considerations to the sentence, for clarity.

Figure 7B- it is unclear as it is written as Rpl35a in the figure but Rps35a in the text.

Thank you for noticing this typo. We used Rpl35a, as is now consistently indicated.

Referee #2:

This manuscript from Di Noia and colleagues investigates roles of the multifunctional RNA-binding protein hnRNPL in B cell development and activation. The authors used CD21-cre to conditionally delete hnRNPL from peripheral B cells, finding that B cells lacking hnRNPL have activation defects and undergo elevated rates of apoptosis. Class switching is also found to be impaired in hnRNPL-depleted cells. The authors perform RNA-seq from hnRNPL-depleted and control activated B-cells and compare their own data to public datasets of hnRNPL depletion studies from several additional cell types. Consistent with impaired activation and increased apoptosis, the authors find that p53 pathway targets are upregulated and Myc and E2F targets are downregulated in hnRNPL-deficient activated B cells. Of these pathways, Myc and E2F pathways appear to be downregulated in several hnRNPL-depleted cell types.

The authors further investigate the effects of hnRNPL deletion on alternative splicing, identifying a small set of chromatin regulators that are consistently alternatively spliced across the panel of hnRNPL depletion datasets.

One such target is SIRT1, prompting examination of mitochondrial defects in the knockout cells. This is an interesting system, and understanding cell type-specific functions of common splicing factors is an important goal. Overall, the experiments appear to be well designed and executed.

The major drawback of the paper is that there are no clear connections between the effects of hnRNPL depletion on RNA biogenesis and the observed cellular phenotypes. Please see below for detailed comments.

We thank the Referee for a fair assessment and for recognizing the interest of our work, and our contribution to uncovering common and cell type specific functions of hnRNPL.

Major points:

1. As noted above, the paper is lacking a clear link between the mis-spliced or differentially expressed genes identified and the B cell phenotypes observed. Two areas stand out: 1) the molecular cause(s) of Myc/E2F pathway downregulation are not identified and 2) the mitochondrial dysfunction investigated in Figure 7 is not directly attributed to SIRT1 regulation. More information in either of these areas would significantly strengthen the manuscript.

We acknowledge this limitation of the paper. As we state in the Discussion, “The number of genes affected in either their expression or splicing by hnRNPL deficiency imply that the cause of B cell activation and antibody response defects is likely pleiotropic”. For this reason, we opted for a comparative analysis approach to identify conserved points of regulation by hnRNPL. We found that hnRNPL deficiency affects common programs and even specific splicing events across cell types and species. Furthermore, many conserved hnRNPL-regulated alternative splicing events affect transcriptional and epigenetic modifiers. This is a main message of our work and implies that hnRNPL can have indirect effects on transcription, including the MYC/E2F programs and mitochondrial function, which can explain the B cell activation and proliferation defects that we report. Predicted changes in the epigenetic landscape upon hnRNPL loss could also explain context dependent changes, such as down or up regulation of MYC/E2F depending on the cell type (see below).

*We now provide evidence that indeed there are global epigenetic alterations in hnRNPL deficient B cells (**new Fig 6I**). Notably, we find significantly reduced global H3K27me3. We focused on H3K27me3 because after the splicing reanalysis (see answer to next point), a splicing change in the H3K27me3 demethylase KDM6A among the two most conserved events (**revised Fig 6E**). Reduced H3K27me3 in hnRNPL deficient B cells is fully consistent with the increase in the KDM6A transcript isoform that contains the NLS and leads to a gain-of-function phenotype (i.e., increased H3K27me3 demethylation), as shown by others (Fotouhi et al, 2023). Reduced H3K27me3 can explain at least some of the genes upregulated in the hnRNPL-deficient B cells, including lncRNAs, which can further contribute to gene expression changes.*

*We have also experimentally demonstrated our prediction that the splicing change affecting Sirt1 would lead to Sirt1 downregulation by favoring the production of a transcript subjected to non-sense mediated decay (**Fig EV4E, F**, we did these experiments in a B cell line because we had a limited number of animals that we preferred to use for WB analysis of epigenetic marks). Because Sirt1 can deacetylate some H3 lysine residues, we also probed H3K9ac levels and found a global reduction in resting hnRNPL-deficient B cells. While Sirt1 downregulation might be expected to increase H3K9ac, this is far from a straightforward prediction. First, unlike the change in KDM6A, reduced Sirt1 would not have a dominant phenotype. Second, there are at least 7 other histone deacetylases and 5 histone acetyl transferases that could affect this mark. Nonetheless, we report this because the global H3K9ac decrease provides further evidence of epigenetic alteration in the absence of hnRNPL.*

*In this context, it is unlikely that any single change in alternative splicing can provide the molecular link between hnRNPL and the MYC/E2F transcriptional programs. The splicing of E2Fs or MYC transcripts was not affected in any cell type (**Tables S3, S4**), ruling out a direct effect. Gene expression of these TFs was largely unaffected too (**Fig EV3B**). In hnRNPL deficient B cells, E2f1 and E2f3 were significantly reduced, but other E2fs or Myc were not (**Fig 4J, Table S1**). Moreover, while the signatures are affected across cell lines, the specific genes do not overlap*

much between different cells (Fig EV3). Furthermore, the MYC/E2F programs were downregulated in 6 cell types but upregulated in HEK293 cells (Fig 5A). Collectively, these observations suggest that the effect on MYC/E2F programs is conserved but the mechanisms behind it are more context dependent. We posit that it could be explained as the indirect consequence of changes in the epigenetic landscape. The conserved upregulation of lncRNAs in hnRNPL-deficient cells, adds another layer of complexity, since lncRNA are well known regulators of gene expression. We have thus modified the Results and Discussion to avoid defining the effect of hnRNPL on MYC/E2F as “regulation”, which can suggest a more direct molecular link than we can prove. We focus on the epigenetic changes as a likely explanation, since we can rule out direct effects on MYC/E2F.

We could not find an easy way to test if Sirt1 was causing the mitochondrial defects. Note that we mentioned in the original manuscript that SIRT1 made us investigate mitochondria, but we were careful to not claim a direct link and to point out additional defects in Ppargc1b and Tfam. In the revised manuscript, we have reduced the discussion on Sirt1 and clearly stated that we cannot directly link its downregulation to mitochondrial defects. We make it clear that other key gene changes like Tfam and Ppargc1b, as well as the global epigenetic changes or lncRNAs, can all contribute to the mitochondrial dysfunction in hnRNPL-deficient B cells.

In summary, we provide additional support to one of our main findings: that hnRNPL regulates the epigenetic landscape of B cells, at least in part by regulating conserved splicing events in chromatin modifiers. This is expected to have broad effects on gene expression, with a conserved effect on MYC and E2F programs, which underlie proliferation (and activation in the case of B cells), both of which are affected. We would argue that these original findings are novel and provide insight into hnRNPL function, compensating for the limitation we acknowledge.

2. The rMATS and associated downstream analyses should be more completely described in the methods section, and the approach and interpretation should be modified to take into account significant shortcomings of the rMATS algorithm. In particular, the rMATS algorithm performs very poorly on retained introns and mutually exclusive exons. The mutually exclusive exon identifications are likely almost entirely spurious (the number of validated MXE events in the human genome is less than or similar to the ~800 claimed to be regulated by hnRNPL in figure 6A; see for example PMID 29242366), complicating the comparison of alternative splicing and differential gene expression in Figure 6B. Further, rMATS generates many spurious calls if the default coverage filters are used. In addition, no information is given about the background set of genes used for the GO analysis in Figure 6C, which is critical for interpretation.

We thank the Referee for these important technical considerations that we had not fully considered. We have now completely reanalyzed splicing in B cells and the other 8 datasets using more stringent parameters, as detailed below. Our main conclusions about conserved effects and conserved spliced events did not change after this revision.

We indeed used the default rMATS coverage filter, as surmised by the Referee. We have now corrected this oversight (borne from inexperience). After a literature survey, we used a coverage filter of ≥ 10 average reads per event, in addition to inclusion level differences of $\geq 10\%$ versus control, and $FDR < 0.1$. As expected, these stringent conditions reduced the number of altered splicing events in all datasets but showed similar findings.

Using the new filtering, rMATS still detected 587 MXE events between WT and hnRNPL-deficient B cells (revised Fig. 6A), substantially less than the 800 in the original analysis. These 587 events might still be considered high when compared to the 855 total MXEs reported for the human exome by the paper mentioned by the Referee (Hatje et al, 2017). However, Hatje et al did not only use another algorithm (WebScipio), but also focused the discovery and validation of new MXEs on mutually exclusive exons that had similar lengths and a high degree of sequence homology. This method is highly accurate but could have underestimated the MXE repertoire. Aberrantly spliced MXE events in the absence of hnRNPL analyzed by rMATS do not have to adhere to these conditions, which can partly explain a higher number. Thus, the revised number of MXEs we report might be overestimated or could reflect the large effect of hnRNPL on splicing. We cannot discriminate between these alternatives without undertaking a validation effort that is beyond the scope of this paper. While the literature

suggests that tools like Limma or IRFinder might be better suited to specifically detect RI, rMATS is an accepted and extensively used general purpose tool to study differential alternative splicing, including RI and MXE events (Mehmood et al, 2020; David et al, 2022). We trust the expertise of the Referee on the matter but side by side comparisons show that all splicing analysis tools have limitations, and none seems clearly superior to the others (Liu et al, 2014; Mehmood et al, 2020). rMATS has limitations like all others but performs well for skipped exon events, which is the type of event we focus on and compare across datasets. Because of these considerations, we think it is fair to report the revised numbers. This answer would be published by EMBO Reports, so this discussion will be available for the consideration and critical assessment of interested researchers.

We note that the use of a more stringent coverage filter had implications for the comparison of multiple datasets because of their heterogeneous quality. The performance of rMATS (and all tools), improves with higher sequencing depth (> 40 M reads/sample), more replicates, longer (>100 bp) and paired-end reads (Mehmood et al, 2020). As summarized in **figure R1**, the BJ and LNCaP cell lines datasets showed the worst combinations (lower coverage + shorter reads) among the datasets. Not surprisingly they were the ones we lost in finding conserved processes among the genes with altered splicing (**revised Figure 6D**). Because of this, we revised our

definition of conserved biological process or splicing event from being in ≥ 8 to ≥ 6 datasets, as a good compromise between sensitivity and specificity. Importantly, **the reanalysis did not change any of the major findings of the original work (revised Figures 6A-D):** 1) differential skipped exon events predominated in hnRNPL deficient cells at least 5-fold to one, 2) there was little overlap between genes with differential expression and differential splicing, 3) histone modification was the most enriched GO in the splicing changes in B cells and the most conserved one across datasets, 4) the conserved SE splicing events identified were still enriched in transcripts of transcriptional regulators and histone modifiers like KDM6A, NSD2 and SIRT1.

As requested, we have added the information about the background gene set used for GO analyses in Methods. We used the list of all genes that have annotations in the GO:BP source database (27,205 genes for *Mus musculus*, 21,110 genes for *Homo sapiens*). This choice allowed equivalent criteria for comparisons across the multiple datasets in two species. The total number of expressed genes detected (≥ 1 read) in the RNA-seq datasets was at least $\sim 32,000$ (except for the Thymocyte dataset in which $\sim 21,000$ genes were detected). Therefore, in almost all cases, limiting the background gene set to only those found in the GO:BP source databases gives us a more conservative estimate of the enrichment than using the larger set of all annotated genes for background.

*We have modified the Methods to reflect all the considerations described above with full description of the procedures and parameters used for the analyses. We also included one of the panels above as **new Figure EV4** summarizing the relevant features of all RNA-seq dataset.*

Minor points:

1. SD should in most instances be used rather than SE.

Thank you for noting this. The revised version displays either individual data point values (when $n < 5$) or SD in all panels, except for the SeaHorse curves (First panel of Figs 7C and D) in which SD complicates the visualization. Still, the individual data points for each calculated parameter are shown in the adjacent panels, which shows the actual data dispersion and their statistical assessment.

2. The authors should describe what lncRNA annotations were used in Figure 5B.

We used GENCODE annotations (vM23 for mouse and v43 for human) for annotating RNA-seq data and identifying lncRNAs. This is now indicated in the methods.

Referee #3:

Class switch recombination (CSR) is a DNA recombination-coupled with deletion mechanism that occurs at the immunoglobulin heavy chain locus (IgH) for determination of effector function of expressed antibodies in B cells. CSR is driven by the activity of the enzyme Activation Induced Deaminase (AID), whose enzymatic activity has been widely debated. In this regard, AID has been suggested to edit mRNAs (for their activation) that eventually catalyze CSR, or has been shown to deaminate DNA to induce mutations and DNA breaks directly. Previous studies by Hu, Honjo et al (PMID: 25902538) have suggested that HnRNPL functions as a co-factor of AID that regulates CSR. Hu et al's observations pitched AID's function as a RNA editing enzyme, and contrasts with the more widely accepted function of AID as a DNA deaminase in CSR.

In this study, Subramani et al, very elegantly establishes using mouse models that HnRNPL does play a role in CSR but this is more likely due to its function in cellular metabolism. Using a combination of conditional allele mouse model and various cell lines, Subramani et al demonstrate that hnRNP-L is important for promoting proliferation of activated B cells and preventing apoptosis. hnRNP-L depleted cells demonstrate wide range of gene expression changes (relatively, enriched with Myc and E2F transcriptional network) including expression of important histone modifiers like KDM6A, NSD2, and SIRT1. As would be expected following reduced SIRT1 expression, hnRNPL-deficient B cells had defective mitochondria leading to ROS overproduction. Mitochondrial dysregulation is aligned with B cell activation defects The experiments are well done with proper controls. The statistical significance of the experiments are well described.

Overall, this is an important study and will be well received in the fields of B Cell biology and CSR. I do not have any major concerns and suggest publication.

Thank you for the positive comments and assessment.

If there is anything I can suggest, the authors could rationalize the previous conclusions regarding CSR by Hu et al slightly better. Primary B cells requires the B cell activation pathways to proliferate and survive. Hu et al, performed their experiments in B cell lines where cell survival may not need these pro-survival mechanisms. Thus, role of hnRNP may also be important for CSR (unlikely via RNA editing though) and also be important for

mitochondrial biology. The authors could elaborate on this point since their study will be likely be followed and cited based on their judgement on the role of these hnRNP proteins in CSR.

Unfortunately, neither our nor Hu et al's results shine much light on whether AID deaminates DNA (as most of the field and us think proven), or RNA. However, we consider our data to be complementary to the data of Hu et al regarding the role of hnRNPL in CSR.

Hu et al performed their work in CH12 cells, a cell line that switches to IgA and in which the Sa is constitutively expressed. Unlike us, they did not report any proliferation or germline transcript defects, but rather found a defect at the DNA repair stage of CSR. Interestingly, for this revision, we knocked down hnRNPL in CH12 cells to measure half-life of Sirt1 isoforms. Despite >75% reduction in hnRNPL protein (new Fig EV4E), we did not observe any CSR defect. However, Hu et al first deleted one Hnrnpl allele and then further reduced expression by siRNA on the het CH12 cells. Theirs and our results could be compatible if hnRNPL needed to be reduced below a certain (quite low) threshold for it to affect CSR. We are providing the result here (Fig R2), but we think is not worth adding to the manuscript to avoid the appearance of a discrepancy when there might not one.

In primary B cells, hnRNPL deletion causes severe proliferation and survival defects but we were able to see a CSR defect in live hnRNPL deficient B cells that had undergone the same number of cell divisions as the WT controls. This cell intrinsic defect in CSR to IgG1 (Fig 3A) is consistent with the result from Hu et al. Unlike Hu et al, we found decreased Sg1 germline transcript, suggesting a transcriptional defect that could at least in part explain the defect in CSR. This does not rule out that hnRNPL has additional downstream roles during DNA end joining, as proposed by Hu et al. In fact we now highlight in the analysis of splicing section that one category affected is DNA end-joining, including the splicing of several enzymes from NHEJ. We did not go in depth here because do not have a way to properly test it in the context of our paper (shRNA in Ch12 is insufficient, and

primary B cells have upstream issues). But we have briefly discussed all these considerations, mentioning that a role in regulating GLT transcription and another one in NHEJ (perhaps by altering splicing of key components) are not mutually exclusive.

Second, the role of hnRNP-L in alternative splicing control is quite interesting and relevant. However, the evidence provided here does not complete address how this works. The discussion could be tuned to address how exactly hnRNP-L works for alternative splicing or the conclusions could be tempered.

This is a fair point. Our data does not discriminate between primary and potentially secondary effects of hnRNPL loss in splicing in B cells. Because our novelty is not in demonstrating that hnRNPL regulates splicing, but on the finding of conserved events, we have done additional analysis of the conserved events. We now show: “As indirect evidence that these conserved events were hnRNPL targets, we scored the presence of hnRNPL-binding motifs (Ray et al, 2013; Smith et al, 2013) within the skipped exon or the adjacent introns. All 70 displayed at least one CA dinucleotide-rich that can be recognized by hnRNPL, and 45 out of the 70 events had the canonical ACACACA or ACACAAA recognized motif” (new Fig. 6F).

Overall, this is a terrific study and deserves publication.

We are very thankful for the appreciation.

References

David JK, Maden SK, Wood MA, Thompson RF & Nellore A (2022) Retained introns in long RNA-seq reads are not reliably detected in sample-matched short reads. *Genome Biol* 23: 240

- Fotouhi O, Nizamuddin S, Falk S, Schilling O, Knüchel-Clarke R, Biniossek ML & Timmers HTM (2023) Alternative mRNA Splicing Controls the Functions of the Histone H3K27 Demethylase UTX/KDM6A. *Cancers (Basel)* 15: 3117
- Gaudreau M-C, Grapton D, Helness A, Vadnais C, Fraszczak J, Shooshtarizadeh P, Wilhelm B, Robert F, Heyd F & Möröy T (2016) Heterogeneous Nuclear Ribonucleoprotein L is required for the survival and functional integrity of murine hematopoietic stem cells. *Sci Rep* 6: 27379
- Hatje K, Rahman R, Vidal RO, Simm D, Hammesfahr B, Bansal V, Rajput A, Mickael ME, Sun T, Bonn S, *et al* (2017) The landscape of human mutually exclusive splicing. *Mol Syst Biol* 13: 1–19
- Huang H, Li Y, Zhang G, Ruan G-X, Zhu Z, Chen W, Zou J, Zhang R, Wang J, Ouyang Y, *et al* (2023) The RNA-binding protein hnRNP F is required for the germinal center B cell response. *Nat Commun* 14: 1731
- Li M, Gou H, Tripathi BKK, Huang J, Jiang S, Dubois W, Waybright T, Lei M, Shi J, Zhou M, *et al* (2015) An Apela RNA-Containing Negative Feedback Loop Regulates p53-Mediated Apoptosis in Embryonic Stem Cells. *Cell Stem Cell* 16: 669–683
- Liu R, Loraine AE & Dickerson JA (2014) Comparisons of computational methods for differential alternative splicing detection using RNA-seq in plant systems. *BMC Bioinformatics* 15: 1–16
- Luo X, Deng C, Liu F, Liu X, Lin T, He D & Wei G (2019) HnRNPL promotes wilms tumor progression by regulating the p53 and Bcl2 pathways. *Oncotargets Ther* 12: 4269–4279
- Mehmood A, Laiho A, Venäläinen MS, McGlinchey AJ, Wang N & Elo LL (2020) Systematic evaluation of differential splicing tools for RNA-seq studies. *Brief Bioinform* 21: 2052–2065
- Monzón-Casanova E, Screen M, Díaz-Muñoz MD, Coulson RMR, Bell SE, Lamers G, Solimena M, Smith CWJ & Turner M (2018) The RNA-binding protein PTBP1 is necessary for B cell selection in germinal centers article. *Nat Immunol* 19: 267–278
- Osma-Garcia IC, Capitan-Sobrino D, Mouysset M, Bell SE, Lebourrier M, Turner M & Diaz-Muñoz MD (2021) The RNA-binding protein HuR is required for maintenance of the germinal centre response. *Nat Commun* 12: 6556
- Osma-Garcia IC, Mouysset M, Capitan-Sobrino D, Aubert Y, Turner M & Diaz-Muñoz MD (2023) The RNA binding proteins TIA1 and TIAL1 promote Mcl1 mRNA translation to protect germinal center responses from apoptosis. *Cell Mol Immunol* 20: 1063–1076
- Ray D, Kazan H, Cook KB, Weirauch MT, Najafabadi HS, Li X, Gueroussov S, Albu M, Zheng H, Yang A, *et al* (2013) A compendium of RNA-binding motifs for decoding gene regulation. *Nature* 499: 172–177
- Seo J-Y, Kim D-Y, Kim S-H, Kim H-J, Ryu HG, Lee J, Lee K-H & Kim K-T (2017) Heterogeneous nuclear ribonucleoprotein (hnRNP) L promotes DNA damage-induced cell apoptosis by enhancing the translation of p53. *Oncotarget* 8: 51108–51122
- Siebring-Van Olst E, Blijlevens M, de Menezes RX, van der Meulen-Muileman IH, Smit EF & van Beusechem VW (2017) A genome-wide siRNA screen for regulators of tumor suppressor p53 activity in human non-small cell lung cancer cells identifies components of the RNA splicing machinery as targets for anticancer treatment. *Mol Oncol* 11: 534–551
- Smith SA, Ray D, Cook KB, Mallory MJ, Hughes TR & Lynch KW (2013) Paralogs hnRNP L and hnRNP LL exhibit overlapping but distinct RNA binding constraints. *PLoS One* 8: e80701
- Zhou X, Li Q, He J, Zhong L, Shu F, Xing R, Lv D, Lei B, Wan B, Yang Y, *et al* (2017) HnRNP-L promotes prostate cancer progression by enhancing cell cycling and inhibiting apoptosis. *Oncotarget* 8: 19342–19353

Dear Dr. Di Noia,

Thank you for the submission of your revised manuscript to our editorial offices. I have now received the reports from the two referees that I asked to re-evaluate the study, you will find below. As you will see, the referees now supports the publication of the study in EMBO reports. Referee #2 has some remaining concerns and suggestions to improve the manuscript, I ask you to address in a final revised manuscript. Please also provide a final p-b-p-response regarding the remaining points of the referee.

- Please provide a final title with not more than 100 characters (including spaces).
- Please provide the abstract written in present tense throughout.
- Please indicate on the title page the corresponding author (and add his/her e-mail address) and order the manuscript sections like this, only using these names:
Title page - Abstract - Keywords - Introduction - Results - Discussion - Methods - Data availability section - Acknowledgements - Disclosure and Competing Interests Statement - References - Figure legends - Expanded View Figure legends
- Please remove the referee tokens from the 'Data availability' and make sure that the datasets are public latest on the publication date of the manuscript.
- Please make sure that the number "n" for how many independent experiments were performed, their nature (biological versus technical replicates), the bars and error bars (e.g. SEM, SD) and the test used to calculate p-values is indicated in the respective figure legends (for main and EV figures) of the final revised manuscript. Please also check that all the p-values are explained in the legend, and that these fit to those shown in the figure. Please provide statistical testing where applicable. Please avoid the phrase 'independent experiment', but clearly state if these were biological or technical replicates. Please also indicate (e.g. with n.s.) if testing was performed, but the differences are not significant. In case n=2, please show the data as separate datapoints without error bars and statistics. See also:

<http://www.embopress.org/page/journal/14693178/authorguide#statisticalanalysis>

If n<5, please show single datapoints for diagrams. It seems that presently some diagrams show only partial statistics or miss the 'n.s.'. Moreover:

- Please indicate the statistical test used for data analysis in the legends of figures 1h-j; 6c-d, i; EV 1d.
- Please note that information related to n is missing in the legends of figure 1j; EV 1a-e.
- Please add to each legend a 'Data Information' section explaining the statistics used or providing information regarding replicates and scales.

- Please make sure that all figure panels (main and EV figures) are called out separately and sequentially. Presently, there seems to be no separate callouts for panels 1B and 6I. Please check.
- The tables S1-S5 are Datasets. Please upload these as dataset files, named Dataset EV1-EV5 (also in the files themselves) and change their callouts accordingly. Please put a title and a legend on the first TAB of each of the excel files.
- Please define the asterisks shown in panel 4K. See also the comment of referee #2.
- Please include Tables S7 and S8 in one reagents and tools table. I have attached templates for that in word or excel format. Please upload the filled in table to the manuscript tracking system as 'Reagent Table' file. Please also adjust any callouts to this table. The example linked below shows how the table will display in the published article and includes examples of the type of information that should be provided for the different categories of reagents and tools. Please list your reagents/tools using the categories provided in the template and do not add additional subheadings to the table. Reagents/tools that do not fit in any of the specific categories can be listed under "Other":

https://www.embopress.org/pb%2Dassets/embo-site/msb_177951_sample_FINAL.pdf

In addition, I would need from you:

- a short, two-sentence summary of the manuscript (not more than 35 words).
- two to four short (!) bullet points highlighting the key findings of your study (two lines each).
- a schematic summary figure as separate file that provides a sketch of the major findings (not a data image) in jpeg or tiff format (with the exact width of 550 pixels and a height of not more than 400 pixels) that can be used as a visual synopsis on our

website.

Best,

Referee #1:

This study expands our understanding of how splicing is regulated and impacts in B cell activation. its a well conducted and thorough study which will have a broader interest.

I appreciate the effort made by the authors to revise the manuscript and support its publication.

Referee #2:

The revised manuscript from Di Nioia and colleagues is improved, and the reviewers' comments have largely been addressed. There remain a couple of issues that should be resolved; please see below for detailed comments.

1. Page 15, the text discussing H3K9ac and H3K27me3 refers of Fig EV4E, but the correct figure call-out is 6I. Also the text, says that "H3K9ac was increased in resting Hnrnp1-/- B cells, but Figure 6I shows a decrease. This, as the text indicates, would be the opposite of the effect expected from Sirt1 downregulation.
2. Figure 4K is not clearly explained. What does the asterisk indicate? Also, why is there a difference in band mobility between resting and d1 activated B cells?
3. The authors should again re-consider their inclusion of rMATS-derived MXE data, as these analyses are almost certainly incorrect. As they note, the important information for the paper is the SE class, for which rMATS performs well.

Point by point responses

Referee #1:

This study expands our understanding of how splicing is regulated and impacts in B cell activation. its a well conducted and thorough study which will have a broader interest.

I appreciate the effort made by the authors to revise the manuscript and support its publication.

Thank you!

Referee #2:

The revised manuscript from Di Nioia and colleagues is improved, and the reviewers' comments have largely been addressed. There remain a couple of issues that should be resolved; please see below for detailed comments.

Thank you for the suggestions and advice on splicing.

1. Page 15, the text discussing H3K9ac and H3K27me3 refers of Fig EV4E, but the correct figure call-out is 6I. Also the text, says that "H3K9ac was increased in resting Hnrnp1-/- B cells, but Figure 6I shows a decrease. This, as the text indicates, would be the opposite of the effect expected from Sirt1 downregulation.

Thank you for catching this, we have corrected these mistakes.

2. Figure 4K is not clearly explained. What does the asterisk indicate? Also, why is there a difference in band mobility between resting and d1 activated B cells?

*Thanks again. The * got misplaced in the figure, we moved it to its intended place on the side of the blot now and we indicate in the legend that "The asterisk indicates the p53 band, assigned based on controls run in parallel."*

We believe the migration difference is due to the different ratio of extraction buffer / protein. Resting B cells have much less protein than activated B cells. As it is apparent from the WB, the p53 antibody gave non-specific bands in B cells. In the absence of a p53 ko control we used a positive control to validate the position of the expected band.

3. The authors should again re-consider their inclusion of rMATS-derived MXE data, as these analyses are almost certainly incorrect. As they note, the important information for the paper is the SE class, for which rMATS performs well.

We understand the concern, but we prefer to leave the data in the paper. Most papers using rMATS, including from labs that commonly analyze splicing, report all types of events including MXEs. This includes the papers studying B cells that we reference in the manuscript. Thus, the exclusion may seem strange or arbitrary to that audience, and we could not find appropriate references to justify this exclusion based exclusively on poor rMATS performance. We do acknowledge that other algorithms may perform better, as we discussed in our rebuttal. That discussion is available to the readers. We have now also added the sentence "We note that MXEs are rare events, and rMATS may overestimate them compared to previous estimates done by a different method in human cells (Hatje et al, 2017)". This plus the discussion in the revision file should allow the readers to judge the data by themselves and raise the point made by the Referee. We hope this is acceptable.

Dr. Javier Di Noia
Institut de Recherches Cliniques de Montréal
Center for immunity, inflammation and infectious diseases
110 Av des Pins Ouest
Montréal, Québec H2W 1R7
Canada

Dear Dr. Di Noia,

I am very pleased to accept your manuscript for publication in the next available issue of EMBO reports. Thank you for your contribution to our journal.

Yours sincerely,
